# On the non-universality of deep learning: quantifying the cost of symmetry

**Emmanuel Abbe**
EPFL

**Enric Boix-Adserà**
MIT

## Abstract

We prove limitations on what neural networks trained by noisy gradient descent (GD) can efficiently learn. Our results apply whenever GD training is equivariant, which holds for many standard architectures and initializations. As applications, (i) we characterize the functions that fully-connected networks can weak-learn on the binary hypercube and unit sphere, demonstrating that depth-2 is as powerful as any other depth for this task; (ii) we extend the merged-staircase necessity result for learning with latent low-dimensional structure [ABM22] to beyond the mean-field regime. Under cryptographic assumptions, we also show hardness results for learning with fully-connected networks trained by stochastic gradient descent (SGD).

## 1 Introduction

Over the last decade, deep learning has made advances in areas as diverse as image classification [KSH12], language translation [BCB14], classical board games [SHS+18], and programming [LCC+22]. Neural networks trained with gradient-based optimizers have surpassed classical methods for these tasks, raising the question: can we hope for deep learning methods to eventually replace all other learning algorithms? In other words, is deep learning a universal learning paradigm? Recently, [AS20, AKM+21] proved that in a certain sense the answer is yes: any PAC-learning algorithm [Val84] can be efficiently implemented as a neural network trained by stochastic gradient descent; analogously, any Statistical Query algorithm [Kea98] can be efficiently implemented as a neural network trained by noisy gradient descent.

However, there is a catch: the result of [AS20] relies on a carefully crafted network architecture with memory and computation modules, which is capable of emulating an arbitrary learning algorithm. This is far from the architectures which have been shown to be successful in practice. Neural networks in practice do incorporate domain knowledge, but they have more "regularity" than the architectures of [AS20], in the sense that they do not rely on heterogeneous and carefully assigned initial weights (e.g., convolutional networks and transformers for image recognition and language processing [LB+95, LKF10, VSP+17], graph neural networks for analyzing graph data [GMS05, BZSL13, VCC+17], and networks specialized for particle physics [BAO+20]). We therefore refine our question:

*Is deep learning with "regular" architectures and initializations a universal learning paradigm? If not, can we quantify its limitations when architectures and data are not well aligned?*

We would like an answer applicable to a wide range of architectures. In order to formalize the problem and develop a general theory, we take an approach similar to [Ng04, Sha18, LZA21] of understanding deep learning through the *equivariance group $G$* (a.k.a., symmetry group) of the learning algorithm.

36th Conference on Neural Information Processing Systems (NeurIPS 2022).

**Definition 1.1** (*G*-equivariant algorithm)**.** *A randomized algorithm $\mathcal{A}$ that takes in a data distribution $\mathcal{D} \in \mathcal{P}(\mathcal{X} \times \mathcal{Y})$[1] and outputs a function $\mathcal{A}(\mathcal{D}) : \mathcal{X} \to \mathcal{Y}$ is said to be G-equivariant if for all $g \in G$*

$$\mathcal{A}(\mathcal{D}) \stackrel{d}{=} \mathcal{A}(g(\mathcal{D})) \circ g. \qquad \text{(G-equivariance)}$$

*Here $g$ is a group element that acts on the data space $\mathcal{X}$, and so is viewed as a function $g : \mathcal{X} \to \mathcal{X}$, and $g(\mathcal{D})$ is the distribution of $(g(\boldsymbol{x}), y)$, where $(\boldsymbol{x}, y) \sim \mathcal{D}$.*

In the case that the algorithm $\mathcal{A}$ is deep learning on the distribution $\mathcal{D}$, the equivariance group depends on the optimizer, the architecture, and the network initialization [Ng04, LZA21].[2]

**Examples of $G$-equivariant algorithms in deep learning**   In many deep learning settings, the equivariance group of the learning algorithm is large. Thus, in this paper, we call an algorithm "regular" if it has a large equivariance group. For example, SGD training of fully-connected networks with Gaussian initialization is orthogonally-equivariant [Ng04]; and is permutation-equivariant if we add skip connections [HZRS16]. SGD training of convolutional networks is translationally-equivariant if circular convolutions are used [SNPP19], and SGD training of i.i.d.-initialized transformers without positional embeddings is equivariant to permutations of tokens [VSP+17]. Furthermore, [LZA21, Theorem C.1] provides general conditions under which a deep learning algorithm is equivariant. See also the preliminaries in Section 2.

**Summary of this work**   Based off of $G$-equivariance, we prove limitations on what "regular" neural networks trained by noisy gradient descent (GD) or stochastic gradient descent (SGD) can efficiently learn, implying a separation with the initializations and architectures considered in [AS20]. For GD, we prove a master theorem that enables two novel applications: (a) characterizing which functions can be efficiently weak-learned by fully-connected (FC) networks on both the hypercube and the unit sphere; and (b) a necessity result for which functions on the hypercube with latent low-dimensional structure can be efficiently learned. See Sections 1.2 and 1.3 for more details.

## 1.1   Related work

Most prior work on computational lower bounds for deep learning has focused on proving limitations of kernel methods (a.k.a. linear methods). Starting with [Bar93] and more recently with [WLLM19, AL19, KMS20, AL20, Hsu, HSSV21, ABM22] it is known that there are problems on which kernel methods provably fail. These results apply to training neural networks in the Neural Tangent Kernel (NTK) regime [JGH18], but do not apply to more general nonlinear training. Furthermore, for specific architectures such as FC architectures [GMMM21, Mis22] and convolutional architectures [MM21], the kernel and random features models at initialization are well understood, yielding stronger lower bounds for training in the NTK regime.

For nonlinear training, which is the setting of this paper, considerably less is known. In the context of sample complexity, [Ng04] introduced the study of the equivariance group of SGD, and constructed a distribution on $d$ dimensions with a $\Omega(d)$ versus $O(1)$ sample complexity separation for learning with an SGD-trained FC architecture versus an arbitrary algorithm. More recently, [LZA21] built on [Ng04] to show a $O(1)$ versus $\Omega(d^2)$ sample-complexity separation between SGD-trained convolutional and FC architectures. In this paper, we also analyze the equivariance group of the training algorithm, but with the goal of proving superpolynomial computational lower bounds.

In the context of computational lower bounds, it is known that networks trained with noisy[3] gradient descent (GD) fall under the Statistical Query (SQ) framework [Kea98], which allows showing computational limitations for GD training based on SQ lower bounds. This has been combined in [AS20, SSS17, MS20, ACHM22] with the permutation symmetry of GD-training of i.i.d. FC networks to prove impossibility of efficiently learning high-degree parities and polynomials. In

---

[1]The set of probability distributions on $\Omega$ is denoted by $\mathcal{P}(\Omega)$. You should think of $\mathcal{D} \in \mathcal{P}(\mathcal{X} \times \mathcal{Y})$ as a distribution of pairs $(\boldsymbol{x}, y)$ of covariates and labels.

[2]Note that the equivariance group of a *training algorithm* should not be confused with the equivariance group of an *architecture* in the context of geometric deep learning [BBCV21]. In that context, $G$-equivariance refers to the property of a neural network architecture $f_{\mathsf{NN}}(\cdot; \boldsymbol{\theta}) : \mathcal{X} \to \mathcal{Y}$ that $f_{\mathsf{NN}}(g(\boldsymbol{x}); \boldsymbol{\theta}) = g(f_{\mathsf{NN}}(\boldsymbol{x}; \boldsymbol{\theta}))$ for all $\boldsymbol{x} \in \mathcal{X}$ and all group elements $g \in G$. In that case, $G$ acts on both the input in $\mathcal{X}$ and output in $\mathcal{Y}$.

[3]Here the noise is used to control the gradients' precision as in [AS20, AKM+21].

our work, we show that these arguments can be viewed in the broader context of more general group symmetries, yielding stronger lower bounds than previously known. For stochastic gradient descent (SGD) training, [ABM22] proves a computational limitation for training of two-layer mean-field networks, but their result applies only when SGD converges to the mean-field limit, and does not apply to more general architectures beyond two-layer networks. Finally, most related to our SGD hardness result is [Sha18], which shows limitations of SGD-trained FC networks under a cryptographic assumption. However, the argument of [Sha18] relies on training being equivariant to linear transformations of the data, and therefore requires that data be whitened or preconditioned. Instead, our result for SGD does not require any preprocessing steps.

There is also recent work showing sample complexity benefits of invariant/equivariant neural network *architectures* [MMM21, EZ21, Ele21, BVB21, Ele22]. In contrast, we study equivariant training *algorithms*. These are distinct concepts: a deep learning algorithm can be $G$-equivariant, while the neural network architecture is neither $G$-invariant nor $G$-equivariant. For example, a FC network is not invariant to orthogonal transformations of the input. However, if we initialize it with Gaussian weights and train with SGD, then the learning algorithm is equivariant to orthogonal transformations of the input (see Proposition 2.5 below).

## 1.2   Contribution 1: Lower bounds for noisy gradient descent (GD)

Consider the supervised learning setup where we train a neural network $f_{\mathsf{NN}}(\cdot; \boldsymbol{\theta}) : \mathcal{X} \to \mathbb{R}$ parametrized by $\boldsymbol{\theta} \in \mathbb{R}^p$ to minimize the mean-squared error on a data distribution $\mathcal{D} \in \mathcal{P}(\mathcal{X} \times \mathbb{R})$,

$$\ell_{\mathcal{D}}(\boldsymbol{\theta}) = \mathbb{E}_{(\boldsymbol{x}, y) \sim \mathcal{D}}[(y - f_{\mathsf{NN}}(\boldsymbol{x}; \boldsymbol{\theta}))^2]. \tag{1}$$

The noisy Gradient Descent (GD) training algorithm randomly initializes $\boldsymbol{\theta}^0 \sim \mu_{\boldsymbol{\theta}}$ for some initialization distribution $\mu_{\boldsymbol{\theta}} \in \mathcal{P}(\mathbb{R}^p)$, and then iteratively updates the parameters with step size $\eta > 0$ in a direction $\boldsymbol{g}_{\mathcal{D}}(\boldsymbol{\theta}^k)$ approximating the population loss gradient, plus Gaussian noise $\boldsymbol{\xi}^k \sim \mathcal{N}(0, \tau^2 \boldsymbol{I})$,

$$\boldsymbol{\theta}^{k+1} = \boldsymbol{\theta}^k - \eta \boldsymbol{g}_{\mathcal{D}}(\boldsymbol{\theta}^k) + \boldsymbol{\xi}^k. \tag{GD}$$

Up to a constant factor, $\boldsymbol{g}_{\mathcal{D}}(\boldsymbol{\theta})$ is the population loss gradient, except we have clipped the gradients of the network with the projection operator $\Pi_{B(0,R)}$ to lie in the ball $B(0, R) = \{\boldsymbol{z} : \|\boldsymbol{z}\|_2 \le R\} \subset \mathbb{R}^p$,[4]

$$\boldsymbol{g}_{\mathcal{D}}(\boldsymbol{\theta}) = -\mathbb{E}_{(\boldsymbol{x}, y) \sim \mathcal{D}}[(y - f_{\mathsf{NN}}(\boldsymbol{x}; \boldsymbol{\theta}))(\Pi_{B(0,R)} \nabla_{\boldsymbol{\theta}} f_{\mathsf{NN}}(\boldsymbol{x}; \boldsymbol{\theta}))].$$

Clipping the gradients is often used in practice to avoid instability from exploding gradients (see, e.g., [ZHSJ19] and references within). In our context, clipping ensures that the injected noise $\boldsymbol{\xi}^k$ is on the same scale as the gradient $\nabla_{\boldsymbol{\theta}} f_{\mathsf{NN}}$ of the network and so it controls the gradients' precision. Similarly to the works [AS20, AKM$^+$21, ACHM22], we consider noisy gradient descent training to be efficient if the following conditions are met.

**Definition 1.2** (Efficiency of GD, informal)**.** *GD training is efficient if the clipping radius $R$, step size $\eta$, and inverse noise magnitude $1/\tau$ are all polynomially-bounded in $d$, since then* (GD) *can be efficiently implemented using noisy minibatch SGD*[5].

We prove that some data distributions cannot be efficiently learned by $G$-equivariant GD training. For this, we introduce the $G$-alignment:

**Definition 1.3** ($G$-alignment)**.** *Let $G$ be a compact group, let $\mu_{\mathcal{X}} \in \mathcal{P}(\mathcal{X})$ be a distribution over data points, and let $f \in L^2(\mu_{\mathcal{X}})$ be a labeling function. The $G$-alignment of $(\mu_{\mathcal{X}}, f)$ is:*

$$\mathcal{C}((\mu_{\mathcal{X}}, f); G) = \sup_h \mathbb{E}_{g \sim \mu_G}[\mathbb{E}_{\boldsymbol{x} \sim \mu_{\mathcal{X}}}[f(g(\boldsymbol{x}))h(\boldsymbol{x})]^2],$$

*where $\mu_G$ is the Haar measure of $G$ and the supremum is over $h \in L^2(\mu_{\mathcal{X}})$ such that $\|h\|^2 = 1$.*

In our applications, we use tools from representation theory (see e.g., [Kna96]) to evaluate the $G$-alignment. Using the $G$-alignment, we can prove a master theorem for lower bounds:

**Theorem 1.4** (GD lower bound, informal statement of Theorem 3.1)**.** *Let $\mathcal{D}_f \in \mathcal{P}(\mathcal{X} \times \mathbb{R})$ be the distribution of $(\boldsymbol{x}, f(\boldsymbol{x}))$ for $\boldsymbol{x} \sim \mu_{\mathcal{X}}$. If $\mu_{\mathcal{X}}$ is $G$-invariant[6] and the $G$-alignment of $(\mu_{\mathcal{X}}, f)$ is small, then $f$ cannot be efficiently learned by a $G$-equivariant GD algorithm.*

---

[4]Note that if $f_{\mathsf{NN}}$ is an $R$-Lipschitz model, then $\boldsymbol{g}_{\mathcal{D}}(\boldsymbol{\theta})$ will simply be the population gradient of the loss.

[5]Efficient implementability by minibatch SGD assumes bounded residual errors.

[6]Meaning that if $\boldsymbol{x} \sim \mu_{\mathcal{X}}$, then for any $g \in G$, we also have $g(\boldsymbol{x}) \sim \mu_{\mathcal{X}}$.

**Proof ideas**  We first make an observation of [Ng04]: if a $G$-equivariant algorithm can learn the function $f$ by training on the distribution $\mathcal{D}_f$, then, for any group element $g \in G$, it can learn $f \circ g$ by training on the distribution $\mathcal{D}_{f \circ g}$. In other words, the algorithm can learn the class of functions $\mathcal{F} = \{f \circ g : g \in g\}$, which can potentially be much larger than just the singleton set $\{f\}$. We conclude by showing that the class of functions $\mathcal{F}$ cannot be efficiently learned by GD training. The intuition is that the $G$-alignment measures the diversity of the functions in $\mathcal{F}$. If the $G$-alignment is small, then there is no function $h$ that correlates with most of the functions in $\mathcal{F}$, which can be used to show $\mathcal{F}$ is hard to learn by gradient descent.

This type of argument appears in [AS20, ACHM22] in the specific case of Boolean functions and for permutation equivariance; our proof both applies to a more general setting (beyond Boolean functions and permutations) and yields sharper bounds; see Appendix A.3. Our bound can also be interpreted in terms of the Statistical Query framework, as we discuss in Appendix A.4. While Theorem 1.4 is intuitively simple, we demonstrate its power and ease-of-use by deriving two new applications.

**Application: Characterization of weak-learnability by fully-connected (FC) networks**  In our first application, we consider weak-learnability: when can a function be learned non-negligibly better than just outputting the estimate $f_{\mathsf{NN}} \equiv 0$? Using Theorem 1.4, we characterize which functions over the binary hypercube $f : \{+1, -1\}^d \to \mathbb{R}$ and over the sphere $f : \mathbb{S}^{d-1} \to \mathbb{R}$ are efficiently weak-learnable by GD-trained FC networks with i.i.d. symmetric and i.i.d. Gaussian initialization, respectively. The takeaway is that a function $f : \{+1, -1\}^d \to \mathbb{R}$ is weak-learnable if and only if it has a nonnegligible Fourier coefficient of order $O(1)$ or $d - O(1)$. Similarly, a function $f : \mathbb{S}^{d-1} \to \mathbb{R}$ is weak-learnable if and only if it has nonnegligible projection onto the degree-$O(1)$ spherical harmonics. Perhaps surprisingly, such functions can be efficiently weak-learned by 2-layer fully-connected networks, which shows that adding more depth does not help. This application is presented in Section 3.1.

**Application: Evidence for the staircase property**  In our second application, we consider learning a target function $f : \{+1, -1\}^d \to \mathbb{R}$ that only depends on the first $P$ coordinates, $f(\boldsymbol{x}) = h(x_1, \ldots, x_P)$. Our regime of interest here is when the function $hand : \{+1, -1\}^P \to \mathbb{R}$ remains fixed and the dimension $d$ grows, since this models the situation where a latent low-dimensional space determines the labels in a high-dimensional dataset. Recently, [ABM22] studied SGD-training of mean-field two-layer networks, and gave a near-characterization of which functions can be learned to arbitrary accuracy $\epsilon$ in $O_{h,\epsilon}(d)$ samples, in terms of the *merged-staircase property* (MSP). Using Theorem 1.4, we prove that the MSP is necessary for GD-learnability whenever training is permutation-equivariant (which applies beyond the 2-layer mean-field regime) and we also generalize it beyond leaps of size 1. Details are in Section 3.2.

## 1.3  Contribution 2: Hardness for stochastic gradient descent (SGD)

The second part of this paper concerns Stochastic Gradient Descent (SGD) training, which randomly initializes the weights $\boldsymbol{\theta}^0 \sim \mu_{\boldsymbol{\theta}}$, and then iteratively trains the parameters with the following update rule to try to minimize the loss (1):

$$\boldsymbol{\theta}^{k+1} = \boldsymbol{\theta}^k - \eta \nabla_{\boldsymbol{\theta}} (y - f_{\mathsf{NN}}(\boldsymbol{x}_{k+1}; \boldsymbol{\theta}))^2 \mid_{\boldsymbol{\theta} = \boldsymbol{\theta}^k}, \tag{SGD}$$

where $(y_{k+1}, \boldsymbol{x}_{k+1}) \sim \mathcal{D}$ is a fresh sample on each iteration, and $\eta > 0$ is the learning rate.[7]

Proving computational lower bounds for SGD is a notoriously difficult problem [AKM$^+$21], exacerbated by the fact that for general architectures SGD can be used to simulate any polynomial-time learning algorithm [AS20]. However, we demonstrate that one can prove hardness results for SGD training based off of cryptographic assumptions when the training algorithm has a large equivariance group. We demonstrate the non-universality of SGD on a standard FC architecture.

**Theorem 1.5** (Hardness for SGD, informal statement of Theorem 4.4)**.** *Under the assumption that the Learning Parities with Noise (LPN) problem*[8] *is hard, FC neural networks with Gaussian initialization*

---

[7]For brevity, we focus on one-pass SGD with a single fresh sample per iteration. Our results extend to empirical risk minimization (ERM) setting and to mini-batch SGD, see Remark E.1.

[8]See Section 4 and Appendix D.3 for definitions and discussion on LPN.

*trained by SGD cannot learn* $f_{\mathrm{mod8}} : \{+1, -1\}^d \to \{0, \dots, 7\}$,

$$f_{\mathrm{mod8}}(\boldsymbol{x}) \equiv \sum_{i=1}^{d} x_i \pmod 8,$$

*in polynomial time from noisy samples* $(\boldsymbol{x}, f_{\mathrm{mod8}}(\boldsymbol{x}) + \xi)$ *where* $\boldsymbol{x} \sim \{+1, -1\}^d$ *and* $\xi \sim \mathcal{N}(0, 1)$.

This result shows a limitation of SGD training based on an average-case reduction from a cryptographic problem. The closest prior result is in [Sha18], which proved hardness results for learning with SGD on FC networks, but required preprocessing the data with a whitening transformation.

**Proof idea** The FC architecture and Gaussian initialization are necessary: an architecture that outputted $f_{\mathrm{mod8}}(\boldsymbol{x})$ at initialization would trivially achieve zero loss. However, SGD on Gaussian-initialized FC networks is *sign-flip* equivariant, and this symmetry makes $f_{\mathrm{mod8}}$ hard to learn. If a sign-flip equivariant algorithm can learn the function $f_{\mathrm{mod8}}(\boldsymbol{x})$ from noisy samples, then it can learn the function $f_{\mathrm{mod8}}(\boldsymbol{x} \odot \boldsymbol{s})$ from noisy samples, where $\boldsymbol{s} \in \{+1, -1\}^d$ is an unknown sign-flip vector, and $\odot$ denotes elementwise product. However, this latter problem is hard under standard cryptographic assumptions. More details in Section 4.

## 2 Preliminaries

**Notation** Let $\mathcal{H}_d = \{+1, -1\}^d$ be the binary hypercube, and $\mathbb{S}^{d-1} = \{\boldsymbol{x} \in \mathbb{R}^d : \|\boldsymbol{x}\|_2 = 1\}$ be the unit sphere. The law of a random variable $X$ is $\mathcal{L}(X)$. If $S$ is a finite set, then $X \sim S$ stands for $X \sim \mathrm{Unif}[S]$. Also let $\boldsymbol{x} \sim \mathbb{S}^{d-1}$ denote $\boldsymbol{x}$ drawn from the uniform Haar measure on $\mathbb{S}^{d-1}$. For a set $\Omega$, let $\mathcal{P}(\Omega)$ be the set of distributions on $\Omega$. Let $\odot$ be the elementwise product. For any $\mu_{\mathcal{X}} \in \mathcal{P}(\mathcal{X})$, and group $G$ acting on $\mathcal{X}$, we say $\mu_{\mathcal{X}}$ is $G$-invariant if $g(\boldsymbol{x}) \overset{d}{=} \boldsymbol{x}$ for $\boldsymbol{x} \sim \mu_{\mathcal{X}}$ and any $g \in G$.

### 2.1 Equivariance of GD and SGD

We define GD and SGD equivariance separately.

**Definition 2.1.** *Let* $\mathcal{A}^{\mathrm{GD}}$ *be the algorithm that takes in data distribution* $\mathcal{D} \in \mathcal{P}(\mathcal{X} \times \mathbb{R})$, *runs* (GD) *on initialization* $\boldsymbol{\theta}^0 \sim \mu_{\boldsymbol{\theta}}$ *for* $k$ *steps, and outputs the function* $\mathcal{A}^{\mathrm{GD}}(\mathcal{D}) = f_{\mathsf{NN}}(\cdot; \boldsymbol{\theta}^k)$

*We say "*$(f_{\mathsf{NN}}, \mu_{\boldsymbol{\theta}})$*-GD is* $G$*-equivariant" if* $\mathcal{A}^{\mathrm{GD}}$ *is* $G$*-equivariant in the sense of Definition 1.1.*

**Definition 2.2.** *Let* $\mathcal{A}^{\mathrm{SGD}}$ *be the algorithm that takes in samples* $(\boldsymbol{x}_i, y_i)_{i \in [n]}$, *runs* (SGD) *on initialization* $\boldsymbol{\theta}^0 \sim \mu_{\boldsymbol{\theta}}$ *for* $n$ *steps, and outputs* $\mathcal{A}^{\mathrm{SGD}}((\boldsymbol{x}_i, y_i)_{i \in [n]}) = f_{\mathsf{NN}}(\cdot; \boldsymbol{\theta}^k)$.

*We say "*$(f_{\mathsf{NN}}, \mu_{\boldsymbol{\theta}})$*-SGD is* $G$*-equivariant" if* $\mathcal{A}^{\mathrm{SGD}}((\boldsymbol{x}_i, y_i)_{i \in [n]}) \overset{d}{=} \mathcal{A}^{\mathrm{SGD}}((g(\boldsymbol{x}_i), y_i)_{i \in [n]}) \circ g$ *for any* $g \in G$, *and any samples* $(\boldsymbol{x}_i, y_i)_{i \in [n]}$.

### 2.2 Regularity conditions on networks imply equivariances of GD and SGD

We take a data space $\mathcal{X} \subseteq \mathbb{R}^d$, and consider the following groups that act on $\mathbb{R}^d$.

**Definition 2.3.** *Define the following groups and actions:*

- *Let* $G_{perm} = S_d$ *denote the group of permutations on* $[d]$. *An element* $\sigma \in G_{perm}$ *acts on* $\boldsymbol{x} \in \mathbb{R}^d$ *in the standard way:* $\sigma(\boldsymbol{x}) = (x_{\sigma(1)}, \dots, x_{\sigma(d)})$.

- *Let* $G_{\mathrm{sign,perm}}$ *denote the group of signed permutations, an element* $g = (\boldsymbol{s}, \sigma) \in G_{\mathrm{sign,perm}}$ *is given by a sign-flip vector* $\boldsymbol{s} \in \mathcal{H}_d$ *and a permutation* $\sigma \in G_{perm}$. *It acts on* $\boldsymbol{x} \in \mathbb{R}^d$ *by* $g(\boldsymbol{x}) = \boldsymbol{s} \odot \sigma(\boldsymbol{x}) = (s_1 x_{\sigma(1)}, \dots, s_d x_{\sigma(d)})$.[9]

- *Let* $G_{\mathrm{rot}} = SO(d) \subseteq GL(d, \mathbb{R})$ *denote the rotation group. An element* $g \in G_{\mathrm{rot}}$ *is a rotation matrix that acts on* $\boldsymbol{x} \in \mathbb{R}^d$ *by matrix multiplication.*

---

[9]The group product is $g_1 g_2 = (\boldsymbol{s}_1, \sigma_1)(\boldsymbol{s}_2, \sigma_2) = (\boldsymbol{s}_1 \odot \sigma_1(\boldsymbol{s}_2), \sigma_1 \circ \sigma_2)$.

Under mild conditions on the neural network architecture and initialization, GD and SGD training are known to be $G_{perm}$-, $G_{\text{sign,perm}}$-, or $G_{\text{rot}}$-equivariant [Ng04, LZA21].

**Assumption 2.4** (Fully-connected i.i.d. first layer and no skip connections from the input). *We can decompose the parameters as $\boldsymbol{\theta} = (\boldsymbol{W}, \boldsymbol{\psi})$, where $\boldsymbol{W} \in \mathbb{R}^{m \times d}$ is the matrix of the first-layer weights, and there is a function $g_{\text{NN}}(\cdot; \boldsymbol{\psi}) : \mathbb{R}^m \to \mathbb{R}$ such that $f_{\text{NN}}(\boldsymbol{x}; \boldsymbol{\theta}) = g_{\text{NN}}(\boldsymbol{W}\boldsymbol{x}; \boldsymbol{\psi})$. Furthermore, the initialization distribution is $\mu_{\boldsymbol{\theta}} = \mu_{\boldsymbol{W}} \times \mu_{\boldsymbol{\psi}}$, where $\mu_{\boldsymbol{W}} = \mu_w^{\otimes(m \times d)}$ for $\mu_w \in \mathcal{P}(\mathbb{R})$.*

Notice that Assumption 2.4 is satisfied by FC networks with i.i.d. initialization. Under assumptions on $\mu_w$, we obtain equivariances of GD and SGD (see Appendix E for proofs.)

**Proposition 2.5** ([Ng04, LZA21]). *Under Assumption 2.4, GD and SGD are $G_{perm}$-equivariant. If $\mu_w$ is sign-flip symmetric, then GD and SGD are $G_{\text{sign,perm}}$-equivariant. If $\mu_w = \mathcal{N}(0, \sigma^2)$ for some $\sigma$, then GD and SGD are $G_{\text{rot}}$-equivariant.*

## 3 Lower bounds for learning with GD

In this section, let $\mathcal{D}(f, \mu_{\mathcal{X}}) \in \mathcal{P}(\mathcal{X} \times \mathbb{R})$ denote the distribution of $(\boldsymbol{x}, f(\boldsymbol{x}))$ where $\boldsymbol{x} \sim \mu_{\mathcal{X}}$.

We give a master theorem for computational lower bounds for learning with $G$-equivariant GD.

**Theorem 3.1** (GD lower bound using $G$-alignment). *Let $G$ be a compact group, and let $f_{\text{NN}}(\cdot; \boldsymbol{\theta}) : \mathcal{X} \to \mathbb{R}$ be an architecture and $\mu_{\boldsymbol{\theta}} \in \mathcal{P}(\mathbb{R}^p)$ be an initialization such that GD is $G$-equivariant.*

*Fix any $G$-invariant distribution $\mu_{\mathcal{X}} \in \mathcal{P}(\mathcal{X})$, any label function $f_* \in L^2(\mu_{\mathcal{X}})$, and any baseline function $\alpha \in L^2(\mu_{\mathcal{X}})$ satisfying $\alpha \circ g = \alpha$ for all $g \in G$. Let $\boldsymbol{\theta}^k$ be the random weights after $k$ time-steps of GD training with noise parameter $\tau > 0$, step size $\eta > 0$, and clipping radius $R > 0$ on the distribution $\mathcal{D} = \mathcal{D}(f_*, \mu_{\mathcal{X}})$. Then, for any $\epsilon > 0$,*

$$\mathbb{P}_{\boldsymbol{\theta}^k}[\ell_{\mathcal{D}}(\boldsymbol{\theta}^k) \le \|f_* - \alpha\|_{L^2(\mu_{\mathcal{X}})}^2 - \epsilon] \le \frac{\eta R \sqrt{k\mathcal{C}}}{2\tau} + \frac{\mathcal{C}}{\epsilon},$$

*where $\mathcal{C} = \mathcal{C}((f_* - \alpha, \mu_{\mathcal{X}}); G)$ is the $G$-alignment of Definition 1.3.*

As discussed in Section 1.2, the theorem states that if the $G$-alignment $\mathcal{C}$ is very small, then GD training cannot efficiently improve on the trivial loss from outputting $\alpha$: either the number of steps $k$, the gradient precision $R/\tau$, or the step size $\eta$ have to be very large in order to learn. Appendix A shows a generalization of the theorem for learning a class of functions $\mathcal{F} = \{f_1, \dots, f_m\}$ instead of just a single function $f_*$. This result goes beyond the lower bound of [AS20] even when $G$ is the trivial group with one element: the main improvement is that Theorem 3.1 proves hardness for learning real-valued functions beyond just Boolean-valued functions. We demonstrate the usefulness of the theorem through two new applications in Sections 3.1 and 3.2.

### 3.1 Application: Characterizing weak-learnability by FC networks

In our first application of Theorem 3.1, we consider FC architectures with i.i.d. initialization, and show how to use their training equivariances to characterize what functions they can weak-learn: i.e., for what target functions $f_*$ they can efficiently achieve a non-negligible correlation after training.

**Definition 3.2** (Weak learnability). *Let $\{\mu_d\}_{d \in \mathbb{N}}$ be a family of distributions $\mu_d \in \mathcal{P}(\mathcal{X}_d)$, and let $\{f_d\}_{d \in \mathbb{N}}$ be a family of functions $f_d \in L^2(\mu_d)$. Finally, let $\{\tilde{f}_d\}_{d \in \mathbb{N}}$ be a family of estimators, where $\tilde{f}_d$ is a random function in $L^2(\mu_d)$. We say that $\{f_d, \mu_d\}_{d \in \mathbb{N}}$ is "weak-learned" by the family of estimators $\{\tilde{f}_d\}_{d \in \mathbb{N}}$ if there are constants $d_0, C > 0$ such that for all $d > d_0$,*

$$\mathbb{P}_{\tilde{f}_d}[\|f_d - \tilde{f}_d\|_{L^2(\mu_d)}^2 \le \|f_d\|_{L^2(\mu_d)}^2 - d^{-C}] \ge 9/10. \tag{2}$$

The constant $9/10$ in the definition is arbitrary. In words, weak-learning measures whether the family of estimators $\{\tilde{f}_d\}$ has a non-negligible edge over simply estimating with the identically zero functions $\tilde{f}_d \equiv 0$. We study weak-learnability by GD-trained FC networks.

**Definition 3.3.** *We say that $\{f_d, \mu_d\}_{d \in \mathbb{N}}$ is efficiently weak-learnable by GD-trained FC networks if there are FC networks and initializations $\{f_{\text{NN},d}, \mu_{\boldsymbol{\theta},d}\}$, and hyperparameters $\{\eta_d, k_d, R_d, \tau_d\}$ such that for some constant $c > 0$,*

- *Hyperparameters are polynomial size: $0 \leq \eta_d, k_d, R_d, 1/\tau_d \leq O(d^c)$;*

- *$\{\tilde{f}_d\}$ weak-learns $\{f_d, \mu_d\}$ in the sense of Definition 3.2, where $\tilde{f}_d = f_{\mathsf{NN}}(\cdot; \boldsymbol{\theta}_d)$ for weights $\boldsymbol{\theta}_d$ that are GD-trained on $\mathcal{D}(f_d, \mu_d)$ for $k_d$ steps with step size $\eta_d$, clipping radius $R_d$, and noise $\tau_d$, starting from initialization $\mu_{\boldsymbol{\theta}, d}$.*

*If $\mu_{\boldsymbol{\theta}, d}$ is i.i.d copies of a symmetric distribution, we say that the FC networks are symmetrically-initialized, and Gaussian-initialized if $\mu_{\boldsymbol{\theta}, d}$ is i.i.d. copies of a Gaussian distribution.*

### 3.1.1 Functions on hypercube, FC networks with i.i.d. symmetric initialization

Let us first consider functions on the Boolean hypercube $f : \mathcal{H}_d \to \mathbb{R}$. These can be uniquely written as a multilinear polynomial

$$f(\boldsymbol{x}) = \sum_{S \subseteq [d]} \hat{f}(S) \prod_{i \in S} x_i,$$

where $\hat{f}(S)$ are the Fourier coefficients of $f$ [O'D14]. We characterize weak learnability of functions on the hypercube in terms of their Fourier coefficients. The full proof is deferred to Appendix B.1.

**Theorem 3.4.** *Let $\{f_d\}_{d \in \mathbb{N}}$ be a family of functions $f_d : \mathcal{H}_d \to \mathbb{R}$ with $\|f_d\|_{L^2(\mathcal{H}_d)} \leq 1$. Then $\{f_d, \mathcal{H}_d\}$ is efficiently weak-learnable by GD-trained symmetrically-initialized FC networks if and only if there is a constant $C > 0$ such that for each $d \in \mathbb{N}$ there is $S_d \subseteq [d]$ with $|S_d| \leq C$ or $|S_d| \geq d - C$, and $|\hat{f}_d(S_d)| \geq \Omega(d^{-C})$.*

The algorithmic result can be achieved by two-layer FC networks, and relies on random features analysis where each network weight is initialized to $0$ with probability $1 - p$, and $+1$ or $-1$ with equal probability $p/2$.[10] Therefore, for weak learning on the hypercube, two-layer networks are as good as networks of any depth. For the converse impossibility result, we apply Theorem 3.1, recalling that GD is $G_{\mathrm{sign,perm}}$-equivariant by Proposition 2.5, and noting that $G_{\mathrm{sign,perm}}$-alignment is:

**Lemma 3.5.** *Let $f : \mathcal{H}_d \to \mathbb{R}$. Then $\mathcal{C}((f, \mathcal{H}_d); G_{\mathrm{sign,perm}}) = \max_{k \in [d]} \binom{d}{k}^{-1} \sum_{\substack{S \subseteq [d] \\ |S| = k}} \hat{f}(S)^2$.*

*Proof.* In the following, let $\boldsymbol{s} \sim \mathcal{H}_d$ and $\sigma \sim G_{perm}$, so that $g = (\boldsymbol{s}, \sigma) \sim G_{\mathrm{sign,perm}}$. Also let $\boldsymbol{x}, \boldsymbol{x}' \sim \mathcal{H}_d$ be independent. For any $h : \mathcal{H}_d \to \mathbb{R}$, by (a) tensorizing, (b) expanding $f$ in the Fourier basis, (c) the orthogonality relation $\mathbb{E}_{\boldsymbol{s}}[\chi_S(\boldsymbol{s})\chi_{S'}(\boldsymbol{s})] = \delta_{S,S'}$, and (d) tensorizing,

$$\mathbb{E}_g[\mathbb{E}_{\boldsymbol{x}}[f(g(\boldsymbol{x}))h(\boldsymbol{x})]^2] = \mathbb{E}_{\sigma, \boldsymbol{s}}[\mathbb{E}_{\boldsymbol{x}}[f(\boldsymbol{s} \odot \sigma(\boldsymbol{x}))h(\boldsymbol{x})]^2]$$

$$\overset{(a)}{=} \mathbb{E}_{\sigma, \boldsymbol{s}, \boldsymbol{x}, \boldsymbol{x}'}[f(\boldsymbol{s} \odot \sigma(\boldsymbol{x}))f(\boldsymbol{s} \odot \sigma(\boldsymbol{x}'))h(\boldsymbol{x})h(\boldsymbol{x}')]$$

$$\overset{(b)}{=} \mathbb{E}_{\boldsymbol{x}, \boldsymbol{x}', \sigma}\Big[\sum_{S, S' \subseteq [d]} \hat{f}(S)\hat{f}(S')h(\boldsymbol{x})h(\boldsymbol{x}')\chi_S(\sigma(\boldsymbol{x}))\chi_{S'}(\sigma(\boldsymbol{x}')) \, \mathbb{E}_{\boldsymbol{s}}[\chi_S(\boldsymbol{s})\chi_{S'}(\boldsymbol{s})]\Big]$$

$$\overset{(c)}{=} \mathbb{E}_{\boldsymbol{x}, \boldsymbol{x}', \sigma}\Big[\sum_{S \subseteq [d]} \hat{f}(S)^2 h(\boldsymbol{x})h(\boldsymbol{x}')\chi_S(\sigma(\boldsymbol{x}))\chi_S(\sigma(\boldsymbol{x}'))\Big]$$

$$\overset{(d)}{=} \mathbb{E}_\sigma\Big[\sum_{S \subseteq [d]} \hat{f}(S)^2 \, \mathbb{E}_{\boldsymbol{x}}[h(\boldsymbol{x})\chi_S(\sigma(\boldsymbol{x}))]^2\Big]$$

$$= \sum_{S \subseteq [d]} \hat{f}(S)^2 \, \mathbb{E}_\sigma[\hat{h}(\sigma^{-1}(S))^2]$$

$$= \sum_{S \subseteq [d]} \hat{f}(S)^2 \binom{d}{|S|}^{-1} \sum_{S', |S'| = |S|} \hat{h}(S')^2.$$

And since $\sum_{S', |S'| = |S|} \hat{h}(S')^2 \leq \|h\|_{L^2(\mathcal{H}_d)}^2$, the supremum over $h$ such that $\|h\|_{L^2(\mathcal{H}_d)} = 1$ is achieved by taking $h(\boldsymbol{x}) = \chi_S(\boldsymbol{x})$ for some $S$. $\qquad\square$

---

[10]Surprisingly, this means that the full parity function $f_*(\boldsymbol{x}) = \prod_{i=1}^d x_i$ can be efficiently learned with such initializations. See Appendix B.

So if the Fourier coefficients of $f$ are negligible for all $S$ s.t. $\min(|S|, d - |S|) \leq O(1)$, then the $G_{\text{sign,perm}}$-alignment of $f$ is negligible. By Theorem 3.1, this means $f$ cannot be learned efficiently. In Appendix B.1.2 we give a concrete example of a hard function, that was not previously known.

### 3.1.2 Functions on sphere, FC networks with i.i.d. Gaussian initialization

We now study learning a target function on the unit sphere, $f \in L^2(\mathbb{S}^{d-1})$, where we take the standard Lebesgue measure on $\mathbb{S}^{d-1}$. A key fact in harmonic analysis is that $L^2(\mathbb{S}^{d-1})$ can be written as the direct sum of subspaces spanned by spherical harmonics of each degree (see, e.g., [Hoc12]).

$$L^2(\mathbb{S}^{d-1}) = \bigoplus_{l=0}^{\infty} \mathcal{V}_{d,l},$$

where $\mathcal{V}_{d,l} \subseteq L^2(\mathbb{S}^{d-1})$ is the space of degree-$l$ spherical harmonics, which is of dimension

$$\dim(\mathcal{V}_{d,l}) = \frac{2l + d - 2}{l} \binom{l + d - 3}{l - 1}.$$

Let $\Pi_{\mathcal{V}_{d,l}} : L^2(\mathbb{S}^{d-1}) \to \mathcal{V}_{d,l}$ be the projection operator to the space of degree-$l$ spherical harmonics. In Appendix B.2, we prove this characterization of weak-learnability for functions on the sphere:

**Theorem 3.6.** *Let $\{f_d\}_{d \in \mathbb{N}}$ be a family of functions $f_d : \mathbb{S}^{d-1} \to \mathbb{R}$ with $\|f_d\|_{L^2(\mathbb{S}^{d-1})} \leq 1$. Then $\{f_d, \mathbb{S}^{d-1}\}$ is efficiently weak-learnable by GD-trained Gaussian-initialized FC networks if and only if there is a constant $C > 0$ such that $\sum_{l=0}^{C} \|\Pi_{\mathcal{V}_{d,l}} f_d\|^2 \geq d^{-C}$.*

The algorithmic result can again be achieved by two-layer FC networks, and is a consequence of the analysis of the random feature kernel in [GMMM21], which shows that the projection of $f_d$ onto the low-degree spherical harmonics can be efficiently learned. For the impossibility result, we apply Theorem 3.1, noting that GD is $G_{\text{rot}}$-equivariant by Proposition 2.5, and the $G_{\text{rot}}$-alignment is:

**Lemma 3.7.** *Let $f \in L^2(\mathbb{S}^{d-1})$. Then $\mathcal{C}((f, \mathbb{S}^{d-1}); G_{\text{rot}}) = \max_{l \in \mathbb{Z}_{\geq 0}} \|\Pi_{\mathcal{V}_{d,l}} f\|^2 / \dim(\mathcal{V}_{d,l})$.*

*Proof.* The $G_{\text{rot}}$-alignment is computed using the representation theory of $G_{\text{rot}}$, specifically the Schur orthogonality theorem (see, e.g., [Ser77, Kna96]). For any $l$, the subspace $\mathcal{V}_{d,l}$ is invariant to action by $G_{\text{rot}}$, meaning that we may define the representation $\Phi_l$ of $G_{\text{rot}}$, which for any $g \in G_{\text{rot}}, f \in \mathcal{V}_{d,l}$ is given by $\Phi_l(g) : \mathcal{V}_{d,l} \to \mathcal{V}_{d,l}$ and $\Phi_l(g)f = f \circ g^{-1}$. Furthermore, $\Phi_l$ is a unitary, irreducible representation, and $\Phi_l$ is not equivalent to $\Phi_{l'}$, for any $l \neq l'$ (see e.g., [Sta90, Theorem 1]). Therefore, by the Schur orthogonality relations [Kna96, Corollary 4.10], for any $v_1, w_1 \in \mathcal{V}_{d,l_1}$ and $v_2, w_2 \in \mathcal{V}_{d,l_2}$, we have

$$\mathbb{E}_{g \sim G_{\text{rot}}}[\langle \phi_{l_1}(g) v_1, w_1 \rangle_{L^2(\mathbb{S}^{d-1})} \langle \phi_{l_2}(g) v_2, w_2 \rangle_{L^2(\mathbb{S}^{d-1})}]$$
$$= \delta_{l_1 l_2} \langle v_1, v_2 \rangle_{L^2(\mathbb{S}^{d-1})} \langle w_1, w_2 \rangle_{L^2(\mathbb{S}^{d-1})} / \dim(\mathcal{V}_{d,l_1}). \quad (3)$$

Let $g \sim G_{\text{rot}}$, drawn from the Haar probability measure. For any $h \in L^2(\mathbb{S}^{d-1})$ such that $\|h\|_{L^2(\mathbb{S}^{d-1})}^2 = 1$, by (a) the decomposition of $L^2(\mathbb{S}^{d-1})$ into subspaces of spherical harmonics, (b) the $G_{\text{rot}}$-invariance of each subspace $\mathcal{V}_{d,l}$, and (c) the Schur orthogonality relations in (3),

$$\mathbb{E}_g[\langle f \circ g, h \rangle_{L^2(\mathbb{S}^{d-1})}^2] \overset{(a)}{=} \sum_{l_1, l_2 = 0}^{\infty} \mathbb{E}_g[\langle \Pi_{\mathcal{V}_{d,l_1}}(f \circ g), \Pi_{\mathcal{V}_{d,l_1}} h \rangle_{L^2(\mathbb{S}^{d-1})} \langle \Pi_{\mathcal{V}_{d,l_2}}(f \circ g), \Pi_{\mathcal{V}_{d,l_2}} h \rangle_{L^2(\mathbb{S}^{d-1})}]$$

$$\overset{(b)}{=} \sum_{l_1, l_2 = 0}^{\infty} \mathbb{E}_g[\langle (\Pi_{\mathcal{V}_{d,l_1}} f) \circ g, \Pi_{\mathcal{V}_{d,l_1}} h \rangle_{L^2(\mathbb{S}^{d-1})} \langle (\Pi_{\mathcal{V}_{d,l_2}} f) \circ g, \Pi_{\mathcal{V}_{d,l_2}} h \rangle_{L^2(\mathbb{S}^{d-1})}]$$

$$\overset{(c)}{=} \sum_{l=0}^{\infty} \frac{1}{\dim(\mathcal{V}_{d,l})} \|\Pi_{\mathcal{V}_{d,l}} f\|_{L^2(\mathbb{S}^{d-1})}^2 \|\Pi_{\mathcal{V}_{d,l}} h\|_{L^2(\mathbb{S}^{d-1})}^2$$

$$\leq \left( \sum_{l=0}^{\infty} \|\Pi_{\mathcal{V}_{d,l}} h\|_{L^2(\mathbb{S}^{d-1})}^2 \right) \max_{l \in \mathbb{Z}_{\geq 0}} \frac{1}{\dim(\mathcal{V}_{d,l})} \|\Pi_{\mathcal{V}_{d,l}} f\|_{L^2(\mathbb{S}^{d-1})}^2$$

$$= \max_{l \in \mathbb{Z}_{\geq 0}} \frac{1}{\dim(\mathcal{V}_{d,l})} \|\Pi_{\mathcal{V}_{d,l}} f\|_{L^2(\mathbb{S}^{d-1})}^2.$$

Let $l^*$ be the optimal value of $l$ in the last line, which is known to exist by the fact that $\|\Pi_{\mathcal{V}_{d,l}} f\|^2 \leq \|f\|^2$ and $\dim(\mathcal{V}_{d,l}) \to \infty$ as $l \to \infty$. The inequality is achieved by $h = \Pi_{\mathcal{V}_{d,l^*}} f / \|\Pi_{\mathcal{V}_{d,l^*}} f\|$. $\qquad\square$

This implies that the $G_{\text{rot}}$-alignment of $f$ is negligible if and only if its projection to the low-order spherical harmonics is negligible. By Theorem 3.1, this implies the necessity result of Theorem 3.6.

### 3.2 Application: Extending the merged-staircase property necessity result

In our second application, we study the setting of learning a sparse function on the binary hypercube (a.k.a. a junta) that depends on only $P \leq d$ coordinates of the input $\boldsymbol{x}$, i.e.,

$$f_*(\boldsymbol{x}) = h_*(x_1, \dots, x_P),$$

where $h_* : \mathcal{H}_P \to \mathbb{R}$. The regime of interest to us is when $h_*$ is fixed and $d \to \infty$, representing a hidden signal in a high-dimensional dataset. This setting was studied by [ABM22], who identified the "merged-staircase property" (MSP) as an extension of [ABB+21]. We generalize the MSP below.

**Definition 3.8** (*l-MSP*)**.** *For $l \in \mathbb{Z}_+$ and $h_* : \mathcal{H}_P \to \mathbb{R}$, we say that $h_*$ satisfies the merged staircase property with leap $l$ (i.e., l-MSP) if its set of nonzero Fourier coefficients $\mathcal{S} = \{S : \hat{h}_*(S) \neq \emptyset\}$ can be ordered as $\mathcal{S} = \{S_1, \dots, S_m\}$ such that for all $i \in [m]$, $|S_i \setminus \cup_{j<i} S_j| \leq l$.*

For example, $h_*(\boldsymbol{x}) = x_1 + x_1 x_2 + x_1 x_2 x_3$ satisfies 1-MSP; $h_*(\boldsymbol{x}) = x_1 x_2 + x_1 x_2 x_3$ satisfies 2-MSP, but not 1-MSP because of the leap required to learn $x_1 x_2$; similarly $h_*(\boldsymbol{x}) = x_1 x_2 x_3 + x_4$ satisfies 3-MSP but not 2-MSP. If $h_*$ satisfies $l$-MSP for some small $l$, then the function $f_*$ can be learned greedily in an efficient manner, by iteratively discovering the coordinates on which it depends. In [ABM22] it was proved that the 1-MSP property nearly characterized which sparse functions could be $\epsilon$-learned in $O_{\epsilon, h_*}(d)$ samples by one-pass SGD training in the mean-field regime.

We prove the MSP necessity result for GD training. On the one hand, our necessity result is for a different training algorithm, GD, which injects noise during training. On the other, our result is much more general since it applies whenever GD is permutation-equivariant, which includes training of FC networks and ResNets of any depth (whereas the necessity result of [ABM22] applies only to two-layer architectures in the mean-field regime). We also generalize the result to any leap $l$.

**Theorem 3.9** (*l-MSP necessity*)**.** *Let $f_{\text{NN}}(\cdot; \boldsymbol{\theta}) : \mathcal{H}_d \to \mathbb{R}$ be an architecture and $\mu_{\boldsymbol{\theta}} \in \mathcal{P}(\mathbb{R}^p)$ be an initialization such that GD is $G_{perm}$-equivariant. Let $\boldsymbol{\theta}^k$ be the random weights after $k$ steps of GD training with noise parameter $\tau > 0$, step size $\eta$, and clipping radius $R$ on the distribution $\mathcal{D} = \mathcal{D}(f_*, \mathcal{H}_d)$. Suppose that $f_*(\boldsymbol{x}) = h_*(\boldsymbol{z})$ where $h_* : \mathcal{H}_P \to \mathbb{R}$ does not satisfy l-MSP for some $l \in \mathbb{Z}_+$. Then there are constants $C, \epsilon_0 > 0$ depending on $h_*$ such that*

$$\mathbb{P}_{\boldsymbol{\theta}^k}[\ell_{\mathcal{D}}(\boldsymbol{\theta}^k) \leq \epsilon_0] \leq \frac{C\eta R}{2\tau} \sqrt{\frac{k}{d^{l+1}}} + \frac{C}{d^{l+1}}.$$

The interpretation is that if $h_*$ does not satisfy $l$-MSP, then to learn $f_*$ to better than $\epsilon_0$ error with constant probability, we need at least $\Omega_{h_*,\epsilon}(d^{l+1})$ steps of (GD) on a network with step size $\eta = O_{h_*,\epsilon}(1)$, clipping radius $R = O_{h_*,\epsilon}(1)$, and noise level $\tau = \Omega_{h_*,\epsilon}(1)$. The proof is deferred to Appendix C. It proceeds by first isolating the "easily-reachable" coordinates $T \subseteq [P]$, and subtracting their contribution from $f_*$. We then bound $G$-alignment of the resulting function, where $G$ is the permutation group on $[d] \setminus T$.

## 4 Hardness for learning with SGD

In this section, for $\gamma > 0$, we let $\mathcal{D}(f, \mu_{\mathcal{X}}, \gamma) \in \mathcal{P}(\mathcal{X} \times \mathbb{R})$ denote the distribution of $(\boldsymbol{x}, f(\boldsymbol{x}) + \xi)$ where $\boldsymbol{x} \sim \mu_{\mathcal{X}}$ and $\xi \sim \mathcal{N}(0, \gamma^2)$ is independent noise.

We show that the equivariance of SGD on certain architectures implies that the function $f_{\text{mod8}} : \mathcal{H}_d \to \{0, \dots, 7\}$ given by

$$f_{\text{mod8}}(\boldsymbol{x}) \equiv \sum_i x_i \pmod 8 \tag{4}$$

is hard for SGD-trained, i.i.d. symmetrically-initialized FC networks. Our hardness result relies on a cryptographic assumption to prove superpolynomial lower bounds for SGD learning. For any $S \subseteq [d]$, let $\chi_S : \mathcal{H}_d \to \{+1, -1\}$ be the parity function $\chi_S(\boldsymbol{x}) = \prod_{i \in S} x_i$.

**Definition 4.1.** *The learning parities with Gaussian noise, $(d, n, \gamma)$-LPGN, problem is parametrized by $d, n \in \mathbb{Z}_{>0}$ and $\gamma \in \mathbb{R}_{>0}$. An instance $(S, \boldsymbol{q}, (\boldsymbol{x}_i, y_i)_{i \in [n]})$ consists of (i) an unknown subset $S \subseteq [d]$ of size $|S| = \lfloor d/2 \rfloor$, and (ii) a known query vector $\boldsymbol{q} \sim \mathcal{H}_d$, and i.i.d. samples $(\boldsymbol{x}_i, y_i)_{i \in [n]} \sim \mathcal{D}(\chi_S, \mathcal{H}_d, \gamma)$. The task is to return $\chi_S(\boldsymbol{q}) \in \{+1, -1\}$.*[11]

Our cryptographic assumption is that $\mathrm{poly}(d)$-size circuits cannot succeed on LPGN.

**Definition 4.2.** *Let $\gamma > 0$. We say $\gamma$-LPGN is $\mathrm{poly}(d)$-time solvable if there is a sequence of sample sizes $\{n_d\}_{d \in \mathbb{N}}$ and circuits $\{\mathcal{A}_d\}_{d \in \mathbb{N}}$ such that $n_d, \mathrm{size}(\mathcal{A}_d) \leq \mathrm{poly}(d)$, and $\mathcal{A}_d$ solves $(d, n_d, \gamma)$-LPGN with success probability at least $9/10$, when inputs are rounded to $\mathrm{poly}(d)$ bits.*

**Assumption 4.3.** *Fix $\gamma$. The $\gamma$-LPGN-hardness assumption is: $\gamma$-LPGN is not $\mathrm{poly}(d)$-time solvable.*

The LPGN problem is the simply standard Learning Parities with Noise problem (LPN) [BKW03], except with Gaussian noise instead of binary classification noise, and we are also promised that $|S| = \lfloor d/2 \rfloor$. In Appendix D.3, we derive Assumption 4.3 from the standard hardness of LPN. We now state our SGD hardness result.

**Theorem 4.4.** *Let $\{f_{\mathrm{NN},d}, \mu_{\boldsymbol{\theta},d}\}_{d \in \mathbb{N}}$ be a family of networks and initializations satisfying Assumption 2.4 (fully-connected) with i.i.d. symmetric initialization. Let $\gamma > 0$, and let $\{n_d\}$ be sample sizes such that $(f_{\mathrm{NN},d}, \mu_{\boldsymbol{\theta},d})$-SGD training on $n_d$ samples from $\mathcal{D}(f_{\mathrm{mod8}}, \mathcal{H}_d, \gamma)$ rounded to $\mathrm{poly}(d)$ bits yields parameters $\boldsymbol{\theta}_d$ with*

$$\mathbb{E}_{\boldsymbol{\theta}_d}[\|f_{\mathrm{mod8}} - f_{\mathrm{NN}}(\cdot; \boldsymbol{\theta}_d)\|^2] \leq 0.0001.$$

*Then, under $(\gamma/2)$-LPGN hardness, $(f_{\mathrm{NN},d}, \mu_{\boldsymbol{\theta},d})$-SGD on $n_d$ samples cannot run in $\mathrm{poly}(d)$ time.*

In order to prove Theorem 4.4, we use the sign-flip equivariance of gradient descent guaranteed by the symmetry in the initialization. A sign-flip equivariant network that learns $f_{\mathrm{mod8}}(\boldsymbol{x})$ from $\gamma$-noisy samples, is capable of solving the harder problem of learning $f_{\mathrm{mod8}}(\boldsymbol{x} \odot \boldsymbol{s})$ from $\gamma$-noisy samples, where $\boldsymbol{s} \in \mathcal{H}_d$ is an unknown sign-flip vector. However, through an average-case reduction we show that this problem is $(\gamma/2)$-LPGN-hard. Therefore the theorem follows by contradiction.

## 5 Discussion

The general GD lower bound in Theorem 3.1 and the approach for basing hardness of SGD training on cryptographic assumptions in Theorem 4.4 could be further developed to other settings.

There are limitations of the results to address in future work. First, the GD lower bound requires adding noise to the gradients, which can hinder training. Second, real-world data distributions are typically not invariant to a group of transformations, so the results obtained by this work may not apply. It is open to develop results for distributions that are approximately invariant.

Finally, it is open whether computational lower bounds for SGD/GD training can be shown beyond those implied by equivariance. For example, consider the function $f : \mathcal{H}_d \to \{+1, -1\}$ that computes the "full parity", i.e., the parity of all of the inputs $f(\boldsymbol{x}) = \prod_{i=1}^{d} x_i$. Past work has empirically shown that SGD on FC networks with Gaussian initialization [SSS17, AS20, NY21] fails to learn this function. Proving this would represent a significant advance, since there is no obvious equivariance that implies that the full parity is hard to learn — in fact we have shown weak-learnability with symmetric $\mathrm{Rad}(1/2)$ initialization, in which case training is $G_{\mathrm{sign,perm}}$-equivariant.

## Acknowledgements

We thank Jason Altschuler, Guy Bresler, Elisabetta Cornacchia, Sonia Hashim, Jan Hazla, Hannah Lawrence, Theodor Misiakiewicz, Dheeraj Nagaraj, and Philippe Rigollet for stimulating discussions. We thank the Simons Foundation and the NSF for supporting us through the Collaboration on the Theoretical Foundations of Deep Learning (deepfoundations.ai). This work was done in part while E.B. was visiting the Simons Institute for the Theory of Computing and the Bernoulli Center at EPFL, and was generously supported by Apple with an AI/ML fellowship.

---

[11]More formally, one would express this as a probabilistic promise problem [Ale03].

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
