# Contents

## A   Lower bound for GD, Proof of Theorem 3.1 and generalization

We prove a generalization of Theorem 3.1, which applies to learning a random function $f_* \sim \mu_{\mathcal{F}}$, where $\mu_{\mathcal{F}}$ is a distribution over functions in $L^2(\mu_{\mathcal{X}})$. In order to state the theorem, let us first define the $G$-alignment for a distribution over functions:

**Definition A.1.** *Let $G$ be a compact group that acts on $\mathcal{X}$, and let $\mu_\mathcal{X} \in \mathcal{P}(\mathcal{X})$ be a $G$-invariant distribution. For any distribution of functions $\mu_\mathcal{F} \in \mathcal{P}(L^2(\mu_\mathcal{X}))$, we define the $G$-alignment as:*

$$\mathcal{C}((\mu_\mathcal{F}, \mu_\mathcal{X}); G) = \sup_h \mathbb{E}_{g \sim \mu_G}[\mathbb{E}_{f \sim \mu_\mathcal{F}}[\langle f \circ g, h \rangle^2_{L^2(\mu_\mathcal{X})}]],$$

*where $\mu_G$ is the Haar probability measure over $G$, and the supremum is over $h \in L^2(\mu_\mathcal{X})$ such that $\|h\|^2 = 1$.*

This is a generalization of the $G$-alignment of Definition 1.3, since we can take $\mu_\mathcal{F}$ to be the probability distribution that has all mass on a deterministic function $f_*$. We use it to prove the following generalization of Theorem 3.1:

**Theorem A.2** (GD lower bound for distribution of functions, using $G$-alignment). *Let $G$ be a compact group that acts on $\mathcal{X}$, and let $f_{\mathsf{NN}}(\cdot; \boldsymbol{\theta}) : \mathcal{X} \to \mathbb{R}$ be an architecture and $\mu_{\boldsymbol{\theta}} \in \mathcal{P}(\mathbb{R}^p)$ be an initialization distribution, such that GD is $G$-equivariant.*

*Fix any $G$-invariant distribution $\mu_\mathcal{X} \in \mathcal{P}(\mathcal{X})$. For any $f \in L^2(\mu_\mathcal{X})$, let $\boldsymbol{\theta}^k_f$ be the random weights after $k$ time-steps of* (GD) *training with noise parameter $\tau > 0$, step size $\eta > 0$, and clipping radius $R > 0$ on the distribution $\mathcal{D}_f = \mathcal{D}(f, \mu_\mathcal{X})$. Fix any baseline function $\alpha \in L^2(\mu_\mathcal{X})$ such that $\alpha \circ g = \alpha$ for all $g \in G$, and let $\mu_\mathcal{F} \in \mathcal{P}(L^2(\mu_\mathcal{X}))$ be any distribution of target functions. Then, for any $\epsilon > 0$,*

$$\mathbb{P}_{f \sim \mu_\mathcal{F}, \boldsymbol{\theta}^k_f}[\ell_{\mathcal{D}_f}(\boldsymbol{\theta}^k_f) \le \|f - \alpha\|^2_{L^2(\mu_\mathcal{X})} - \epsilon] \le \frac{\eta R \sqrt{k\mathcal{C}}}{2\tau} + \frac{\mathcal{C}}{\epsilon},$$

*where $\mathcal{C} = \mathcal{C}((\bar{\mu}_\mathcal{F}, \mu_\mathcal{X}); G)$, and $\bar{\mu}_\mathcal{F}$ is the distribution of $f - \alpha$ for $f \sim \mu_\mathcal{F}$.*

Theorem 3.1 is the special case in which $\mu_\mathcal{F}$ has all probability mass on one atom $f_*$.

## A.1 Proof of Theorem A.2

We derive Theorem A.2 from the following theorem, which is the same bound but without the $G$-equivariance assumption (think of $G$ as being the trivial group). We first define the alignment of a distribution of functions:

**Definition A.3.** *Let $\mu_\mathcal{X} \in \mathcal{P}(\mathcal{X})$. For any distribution of functions $\mu_\mathcal{F} \in \mathcal{P}(L^2(\mu_\mathcal{X}))$, we define:*

$$\mathcal{C}(\mu_\mathcal{F}, \mu_\mathcal{X}) = \sup_h \mathbb{E}_{f \sim \mu_\mathcal{F}}[\langle f, h \rangle^2],$$

*where the supremum is over $h \in L^2(\mu_\mathcal{X})$ such that $\|h\|^2 = 1$.*

**Theorem A.4** (GD lower bound for distribution of functions). *Let $f_{\mathsf{NN}}(\cdot; \boldsymbol{\theta}) : \mathcal{X} \to \mathbb{R}$ be an architecture, and let $\mu_{\boldsymbol{\theta}} \in \mathcal{P}(\mathbb{R}^p)$ be an initialization distribution.*

*Fix $\mu_\mathcal{X} \in \mathcal{P}(\mathcal{X})$. For any $f \in L^2(\mu_\mathcal{X})$, let $\boldsymbol{\theta}^k_f$ be the random weights after $k$ time-steps of* (GD) *training with noise parameter $\tau > 0$, step size $\eta > 0$, and clipping radius $R > 0$ on the distribution $\mathcal{D}_f = \mathcal{D}(f, \mu_\mathcal{X})$. Then, for any $\alpha \in L^2(\mu_\mathcal{X})$ any $\mu_\mathcal{F} \in \mathcal{P}(L^2(\mu_\mathcal{X}))$, and any $\epsilon > 0$,*

$$\mathbb{P}_{f \sim \mu_\mathcal{F}, \boldsymbol{\theta}^k_f}[\ell_{\mathcal{D}_f}(\boldsymbol{\theta}^k_f) \le \|f - \alpha\|^2_{L^2(\mu_\mathcal{X})} - \epsilon] \le \frac{\eta R \sqrt{k\mathcal{C}}}{2\tau} + \frac{\mathcal{C}}{\epsilon},$$

*where $\mathcal{C} = \mathcal{C}(\bar{\mu}_\mathcal{F}, \mu_\mathcal{X})$, and $\bar{\mu}_\mathcal{F}$ is the distribution of $f - \alpha$ for $f \sim \mu_\mathcal{F}$.*

We defer the proof of this theorem, and first use it to prove Theorem A.2.

*Proof of Theorem A.2.* Let $\mu_G$ be the Haar probability measure on $G$. Define the distribution of functions $\nu_\mathcal{F} \in \mathcal{P}(L^2(\mu_\mathcal{X}))$ as the distribution of $f \circ g \in L^2(\mu_\mathcal{X})$, where $f \sim \mu_\mathcal{F}$ and $g \sim \mu_G$, independently. Notice that $f \circ g$ is in $L^2(\mu_\mathcal{X})$ since $\mu_\mathcal{X}$ is $G$-invariant and $f \in L^2(\mu_\mathcal{X})$, so this is well defined.

Now define $\bar{\mu}_\mathcal{F}$ and $\bar{\nu}_\mathcal{F}$ to be the distribution of $f - \alpha$ for $f \sim \mu_\mathcal{F}$ and $f \sim \nu_\mathcal{F}$, respectively. Since $\alpha \circ g = \alpha$ for all $g \in G$, we have

$$\mathcal{C}((\bar{\mu}_\mathcal{F}, \mu_\mathcal{X}); G) = \sup_h \mathbb{E}_{g \sim \mu_G}[\mathbb{E}_{f \sim \mu_\mathcal{F}}[\langle (f - \alpha) \circ g, h \rangle^2]] = \sup_h \mathbb{E}_{g \sim \mu_G}[\mathbb{E}_{f \sim \mu_\mathcal{F}}[\langle (f \circ g) - \alpha, h \rangle^2]]$$

$$= \sup_h \mathbb{E}_{f \sim \bar{\nu}_\mathcal{F}}[\langle f, h \rangle^2] = \mathcal{C}(\bar{\nu}_\mathcal{F}, \mu_\mathcal{X}).$$

We conclude by Theorem A.4 that

$$\mathbb{P}_{f\sim\mu_{\mathcal{F}},g\sim\mu_G,\boldsymbol{\theta}^k_{f\circ g}}[\ell_{\mathcal{D}_{f\circ g}}(\boldsymbol{\theta}^k_{f\circ g}) \leq \|f\circ g - \alpha\|^2 - \epsilon] \leq \frac{\eta R\sqrt{k\mathcal{C}}}{2\tau} + \frac{\mathcal{C}}{\epsilon},$$

for $\mathcal{C} = \mathcal{C}(\bar{\mu}_{\mathcal{F}}, \mu_{\mathcal{X}}, G)$. To derive Theorem A.2, note for any $f \in L^2(\mu_{\mathcal{X}})$ and $g \in G$, by the $G$-equivariance of GD training we have $\ell_{\mathcal{D}_{f\circ g}}(\boldsymbol{\theta}^k_{f\circ g}) \stackrel{d}{=} \ell_{\mathcal{D}_f}(\boldsymbol{\theta}^k_f)$. Finally, conclude by noting $\|f \circ g - \alpha\|^2_{L^2(\mu_{\mathcal{X}})} = \|(f - \alpha) \circ g\|^2_{L^2(\mu_{\mathcal{X}})} = \|f - \alpha\|^2_{L^2(\mu_{\mathcal{X}})}$, where we first use the $G$-invariance of $\alpha$ and then that of $\mu_{\mathcal{X}}$. $\qquad\square$

## A.2 Proof of Theorem A.4

The proof of Theorem A.4 is a variation on the junk flow argument of [AS20]. There are two significant differences. First, the junk data distribution is not chosen so that labels are independent of data, but rather chosen so that the labels are given by the function $\alpha$. This allows us to prove lower bounds beyond weak learning (cf. allowing the merged-staircase property necessity result of Section 3.2). Second, instead of using cross-predictability, we use a tighter bound based on the quantity $\mathcal{C}(\bar{\mu}_{\mathcal{F}}, \mu_{\mathcal{X}})$. This allows a $\Omega(d^{l+1})$-lower bound instead of a $\Omega(d^{(l+1)/2})$-lower bound for functions that are not $l$-MSP in Section 3.2. However, for this tighter bound during training we need to clip the gradients of the neural network instead of the gradients of the loss as in [AS20]. In Appendix A.3, we show how to recover the bound of [AS20] based on cross-predictability if we instead clip the gradients of the loss.

For the analysis, we define the following gradient descent trajectories:

- *GD trajectory on $\mathcal{D}_f$*: for any $f : \mathcal{X} \to \mathbb{R}$, we let $\boldsymbol{\theta}^0_f, \ldots, \boldsymbol{\theta}^k_f$ be the trajectory of (GD) on the data distribution $\mathcal{D}_f = \mathcal{D}(f, \mu_{\mathcal{X}}) \in \mathcal{P}(\mathcal{X} \times \mathbb{R})$. I.e., we initialize $\boldsymbol{\theta}^0 \sim \mu_{\boldsymbol{\theta}}$, and update

$$\boldsymbol{\theta}^{k+1}_f = \boldsymbol{\theta}^k_f - \eta\boldsymbol{g}_{\mathcal{D}_f}(\boldsymbol{\theta}^k_f) + \boldsymbol{\xi}^k, \qquad \text{where } \boldsymbol{\xi}^k \sim \mathcal{N}(0, \tau^2\boldsymbol{I}).$$

- *Junk GD trajectory*: define $\mathcal{D}_{\text{junk}} = \mathcal{D}(\alpha, \mu_{\mathcal{X}})$. We let $\boldsymbol{\theta}^0_\alpha, \ldots, \boldsymbol{\theta}^k_\alpha$ be the trajectory of (GD) on $\mathcal{D}_{\text{junk}}$, which we call the *junk trajectory*. I.e., we initialize $\boldsymbol{\theta}^0_\alpha \sim \mu_{\boldsymbol{\theta}}$, and update

$$\boldsymbol{\theta}^{k+1}_\alpha = \boldsymbol{\theta}^k_\alpha - \eta\boldsymbol{g}_{\mathcal{D}_{\text{junk}}}(\boldsymbol{\theta}^k_\alpha) + \tilde{\boldsymbol{\xi}}^k, \qquad \text{where } \tilde{\boldsymbol{\xi}}^k \sim \mathcal{N}(0, \tau^2\boldsymbol{I}).$$

We now prove that the junk trajectory $\boldsymbol{\theta}^k_\alpha$ stays close to the trajectory $\boldsymbol{\theta}^k_f$, for most functions $f \sim \mu_{\mathcal{F}}$. The bound depends on the quantity $\mathcal{C}(\bar{\mu}_{\mathcal{F}}, \mu_{\mathcal{X}})$.

**Lemma A.5** (Junk trajectory close to most GD trajectories). *Under the assumptions of Theorem A.4,*

$$\mathbb{E}_{f\sim\mu_{\mathcal{F}}}[\text{TV}(\mathcal{L}(\boldsymbol{\theta}^k_\alpha), \mathcal{L}(\boldsymbol{\theta}^k_f))] \leq \frac{\eta R}{2\tau}\sqrt{k\mathcal{C}(\bar{\mu}_{\mathcal{F}}, \mu_{\mathcal{X}})}.$$

Finally, we show that the junk GD trajectory is not correlated with most random $f \sim \mu_{\mathcal{F}}$. This bound again depends on the quantity $\mathcal{C}(\bar{\mu}_{\mathcal{F}}, \mu_{\mathcal{X}})$.

**Lemma A.6** (Junk trajectory does not learn). *Under the assumptions of Theorem A.4, for any $\epsilon > 0$,*

$$\mathbb{P}_{f\sim\mu_{\mathcal{F}},\boldsymbol{\theta}^k_\alpha}[\ell_{\mathcal{D}_f}(\boldsymbol{\theta}^k_\alpha) \leq \|f - \alpha\|^2 - \epsilon] \leq \mathcal{C}(\bar{\mu}_{\mathcal{F}}, \mu_{\mathcal{X}})/\epsilon.$$

Combining the above two lemmas, we prove the theorem.

*Proof of Theorem A.4.* By Lemma A.5, we can couple $\boldsymbol{\theta}^k_\alpha$ with $\boldsymbol{\theta}^k_f$ so that

$$\mathbb{P}_{f\sim\mu_{\mathcal{F}},\boldsymbol{\theta}^k_\alpha,\boldsymbol{\theta}^k_f}[\boldsymbol{\theta}^k_\alpha \neq \boldsymbol{\theta}^k_f] \leq \frac{\eta R}{2\tau}\sqrt{k\mathcal{C}(\bar{\mu}_{\mathcal{F}}, \mu_{\mathcal{X}})}.$$

So, for any $t > 0$,

$$\mathbb{P}_{f\sim\mu_{\mathcal{F}},\boldsymbol{\theta}^k_f}[\ell_{\mathcal{D}_f}(\boldsymbol{\theta}^k_f) \leq t] \leq \mathbb{P}_{f\sim\mu_{\mathcal{F}},\boldsymbol{\theta}^k_\alpha}[\ell_{\mathcal{D}_f}(\boldsymbol{\theta}^k_\alpha) \leq t] + \frac{\eta R}{2\tau}\sqrt{k\mathcal{C}(\bar{\mu}_{\mathcal{F}}, \mu_{\mathcal{X}})},$$

and the theorem follows by Lemma A.6. $\qquad\square$

### A.2.1 Proofs of auxiliary lemmas

*Proof of Lemma A.5.* For brevity, write $\boldsymbol{\theta}_\alpha^{\leq k} = (\boldsymbol{\theta}_\alpha^0, \ldots, \boldsymbol{\theta}_\alpha^k)$, and similarly for $\boldsymbol{\theta}_{\bar{f}}^{\leq k}$. We bound the KL-divergence between the junk trajectory $\boldsymbol{\theta}_\alpha^{\leq k}$ and the trajectory $\boldsymbol{\theta}_{\bar{f}}^{\leq k}$ by (a) using the chain rule for KL-divergence, (b) the fact that $\mathcal{L}(\boldsymbol{\theta}_\alpha^0) = \mathcal{L}(\boldsymbol{\theta}_f^0) = \mu_{\boldsymbol{\theta}}$, (c) the Markov property of GD training, (d) the definition of the update step (GD), and (e) the KL divergence between two Gaussians:

$$\mathrm{KL}(\mathcal{L}(\boldsymbol{\theta}_\alpha^{\leq k})||\mathcal{L}(\boldsymbol{\theta}_{\bar{f}}^{\leq k})) \tag{5}$$

$$\overset{(a)}{=} \mathrm{KL}(\mathcal{L}(\boldsymbol{\theta}_\alpha^0)||\mathcal{L}(\boldsymbol{\theta}_f^0)) + \sum_{k'=1}^k \mathrm{KL}(\mathcal{L}(\boldsymbol{\theta}_\alpha^{k'}|\boldsymbol{\theta}_\alpha^{\leq k'-1})||\mathcal{L}(\boldsymbol{\theta}_{\bar{f}}^{k'}|\boldsymbol{\theta}_{\bar{f}}^{\leq k'-1}))$$

$$\overset{(b)}{=} \sum_{k'=1}^k \mathrm{KL}(\mathcal{L}(\boldsymbol{\theta}_\alpha^{k'}|\boldsymbol{\theta}_\alpha^{\leq k'-1})||\mathcal{L}(\boldsymbol{\theta}_{\bar{f}}^{k'}|\boldsymbol{\theta}_{\bar{f}}^{\leq k'-1}))$$

$$\overset{(c)}{=} \sum_{k'=1}^k \mathrm{KL}(\mathcal{L}(\boldsymbol{\theta}_\alpha^{k'}|\boldsymbol{\theta}_\alpha^{k'-1})||\mathcal{L}(\boldsymbol{\theta}_{\bar{f}}^{k'}|\boldsymbol{\theta}_{\bar{f}}^{k'-1}))$$

$$\overset{(d)}{=} \sum_{k'=1}^k \mathbb{E}_{\boldsymbol{\theta}' \sim \mathcal{L}(\boldsymbol{\theta}_\alpha^{k'-1})}[\mathrm{KL}(\mathcal{N}(\boldsymbol{\theta}' - \eta \boldsymbol{g}_{\mathcal{D}_{\mathrm{junk}}}(\boldsymbol{\theta}'), \tau^2 \boldsymbol{I})||\mathcal{N}(\boldsymbol{\theta}' - \eta \boldsymbol{g}_{\mathcal{D}_f}(\boldsymbol{\theta}'), \tau^2 \boldsymbol{I}))]$$

$$\overset{(e)}{=} \sum_{k'=1}^k \frac{\eta^2}{2\tau^2} \mathbb{E}_{\boldsymbol{\theta}' \sim \mathcal{L}(\boldsymbol{\theta}_\alpha^{k'-1})}[\|\boldsymbol{g}_{\mathcal{D}_{\mathrm{junk}}}(\boldsymbol{\theta}') - \boldsymbol{g}_{\mathcal{D}_f}(\boldsymbol{\theta}')\|^2]. \tag{6}$$

In order to analyze this, let us simplify the following quantity. Throughout $\boldsymbol{x}, \boldsymbol{x}' \sim \mu_{\mathcal{X}}$ are i.i.d. We use that (a) $\mathcal{D}_{\mathrm{junk}}$ and $\mathcal{D}_f$ have the same marginal distribution of $\boldsymbol{x} \sim \mu_{\mathcal{X}}$, (b) the definitions of $\mathcal{D}_{\mathrm{junk}}$ and $\mathcal{D}_f$,

$$\boldsymbol{g}_{\mathcal{D}_{\mathrm{junk}}}(\boldsymbol{\theta}') - \boldsymbol{g}_{\mathcal{D}_f}(\boldsymbol{\theta}')$$

$$= -\mathbb{E}_{(\boldsymbol{x}, y) \sim \mathcal{D}_{\mathrm{junk}}}[(y - f_{\mathsf{NN}}(\boldsymbol{x}; \boldsymbol{\theta}'))\Pi_{B(0,R)}\nabla_{\boldsymbol{\theta}} f_{\mathsf{NN}}(\boldsymbol{x}; \boldsymbol{\theta}')]$$
$$\quad + \mathbb{E}_{(\boldsymbol{x}, y) \sim \mathcal{D}_f}[(y - f_{\mathsf{NN}}(\boldsymbol{x}; \boldsymbol{\theta}'))\Pi_{B(0,R)}\nabla_{\boldsymbol{\theta}} f_{\mathsf{NN}}(\boldsymbol{x}; \boldsymbol{\theta}')]$$

$$\overset{(a)}{=} -\mathbb{E}_{(\boldsymbol{x}, y) \sim \mathcal{D}_{\mathrm{junk}}}[y\Pi_{B(0,R)}\nabla_{\boldsymbol{\theta}} f_{\mathsf{NN}}(\boldsymbol{x}; \boldsymbol{\theta}')] + \mathbb{E}_{(\boldsymbol{x}, y) \sim \mathcal{D}_f}[y\Pi_{B(0,R)}\nabla_{\boldsymbol{\theta}} f_{\mathsf{NN}}(\boldsymbol{x}; \boldsymbol{\theta}')]$$

$$\overset{(b)}{=} -\mathbb{E}_{\boldsymbol{x}}[\alpha(\boldsymbol{x})\Pi_{B(0,R)}\nabla_{\boldsymbol{\theta}} f_{\mathsf{NN}}(\boldsymbol{x}; \boldsymbol{\theta}')] + \mathbb{E}_{\boldsymbol{x}}[f(\boldsymbol{x})\Pi_{B(0,R)}\nabla_{\boldsymbol{\theta}} f_{\mathsf{NN}}(\boldsymbol{x}; \boldsymbol{\theta}')]$$

$$= \mathbb{E}_{\boldsymbol{x}}[(f(\boldsymbol{x}) - \alpha(\boldsymbol{x}))\Pi_{B(0,R)}\nabla_{\boldsymbol{\theta}} f_{\mathsf{NN}}(\boldsymbol{x}; \boldsymbol{\theta}')].$$

Now let us draw $f \sim \mu_{\mathcal{F}}$, and bound the expected KL divergence of $\boldsymbol{\theta}_\alpha^{\leq k}$ with $\boldsymbol{\theta}_{\bar{f}}^{\leq k}$. By (a) plugging the above equation into (6), (b) using independence of $\boldsymbol{\theta}_\alpha^{k'-1}$ from $f$, (c) using the definition of $\mathcal{C}(\bar{\mu}_{\mathcal{F}}, \mu_{\mathcal{X}})$ to bound each coordinate, and (d) using the fact that $\Pi_{B(0,R)}$ is projection to the ball of radius $R$,

$$\mathbb{E}_f[\mathrm{KL}(\mathcal{L}(\boldsymbol{\theta}_\alpha^{\leq k})||\mathcal{L}(\boldsymbol{\theta}_{\bar{f}}^{\leq k}))] \overset{(a)}{=} \mathbb{E}_f[\sum_{k'=1}^k \frac{\eta^2}{2\tau^2} \mathbb{E}_{\boldsymbol{\theta}' \sim \mathcal{L}(\boldsymbol{\theta}_\alpha^{k'-1})}[\|\mathbb{E}_{\boldsymbol{x}}[(f(\boldsymbol{x}) - \alpha(\boldsymbol{x}))\Pi_{B(0,R)}\nabla_{\boldsymbol{\theta}} f_{\mathsf{NN}}(\boldsymbol{x}; \boldsymbol{\theta}')]\|^2]]$$

$$\overset{(b)}{=} \sum_{k'=1}^k \frac{\eta^2}{2\tau^2} \mathbb{E}_{\boldsymbol{\theta}' \sim \mathcal{L}(\boldsymbol{\theta}_\alpha^{k'-1})}[\mathbb{E}_f[\|\mathbb{E}_{\boldsymbol{x}}[(f(\boldsymbol{x}) - \alpha(\boldsymbol{x}))\Pi_{B(0,R)}\nabla_{\boldsymbol{\theta}} f_{\mathsf{NN}}(\boldsymbol{x}; \boldsymbol{\theta}')]\|^2]]$$

$$\overset{(c)}{\leq} \sum_{k'=1}^k \frac{\eta^2}{2\tau^2} \mathbb{E}_{\boldsymbol{\theta}' \sim \mathcal{L}(\boldsymbol{\theta}_\alpha^{k'-1})}[\mathcal{C}(\bar{\mu}_{\mathcal{F}}, \mu_{\mathcal{X}})\mathbb{E}_{\boldsymbol{x}}[\|\Pi_{B(0,R)}\nabla_{\boldsymbol{\theta}} f_{\mathsf{NN}}(\boldsymbol{x}; \boldsymbol{\theta}')\|^2]]$$

$$\overset{(d)}{\leq} \frac{k\eta^2 R^2}{2\tau^2}\mathcal{C}(\bar{\mu}_{\mathcal{F}}, \mu_{\mathcal{X}}). \tag{7}$$

Finally we apply (a) the data processing inequality for total variation distance, (b) Pinsker's inequality, (c) Jensen's inequality, and (d) the bound in (7):

$$\mathbb{E}_f[\text{TV}(\mathcal{L}(\boldsymbol{\theta}_\alpha^k), \mathcal{L}(\boldsymbol{\theta}_f^k))] \overset{(a)}{\leq} \mathbb{E}_f[\text{TV}(\mathcal{L}(\boldsymbol{\theta}_\alpha^{\leq k}), \mathcal{L}(\boldsymbol{\theta}_{\bar{f}}^{\leq k}))]$$

$$\overset{(b)}{\leq} \mathbb{E}_f[\sqrt{\frac{1}{2}\text{KL}(\mathcal{L}(\boldsymbol{\theta}_\alpha^{\leq k})\|\mathcal{L}(\boldsymbol{\theta}_{\bar{f}}^{\leq k}))}]$$

$$\overset{(c)}{\leq} \sqrt{\mathbb{E}_f[\frac{1}{2}\text{KL}(\mathcal{L}(\boldsymbol{\theta}_\alpha^{\leq k})\|\mathcal{L}(\boldsymbol{\theta}_{\bar{f}}^{\leq k}))]}$$

$$\overset{(d)}{\leq} \frac{\eta L}{2\tau}\sqrt{k\mathcal{C}(\bar{\mu}_\mathcal{F};\mu_\mathcal{X})}.$$

$\square$

Finally, we prove Lemma A.6, which is the last remaining lemma.

*Proof.* Define $\rho = \mathbb{E}_{\boldsymbol{x}}[(f(\boldsymbol{x}) - \alpha(\boldsymbol{x}))f_{\text{NN}}(\boldsymbol{x};\boldsymbol{\theta}_\alpha^k)]$. Recall the junk trajectory $\boldsymbol{\theta}_\alpha^k$ is drawn independently of $f \sim \mu_\mathcal{F}$, so, by definition of $\mathcal{C}(\bar{\mu}_\mathcal{F}, \mu_\mathcal{X})$,

$$\mathbb{E}_f[\rho^2] \leq \mathcal{C}(\bar{\mu}_\mathcal{F}, \mu_\mathcal{X})\, \mathbb{E}_{\boldsymbol{x}}[f_{\text{NN}}(\boldsymbol{x};\boldsymbol{\theta}_\alpha^k)^2].$$

Let $E$ be the event that $\rho^2 \leq \epsilon\, \mathbb{E}_{\boldsymbol{x}}[f_{\text{NN}}(\boldsymbol{x};\boldsymbol{\theta}_\alpha^k)^2]$. By a Markov bound, $\mathbb{P}[E] \geq 1 - \mathcal{C}(\bar{\mu}_\mathcal{F}, \mu_\mathcal{X})/\epsilon$. Finally, under event $E$ we have

$$\ell_{\mathcal{D}_f}(\boldsymbol{\theta}_\alpha^k) = \mathbb{E}_{\boldsymbol{x}}[(f(\boldsymbol{x}) - f_{\text{NN}}(\boldsymbol{x};\boldsymbol{\theta}_\alpha^k))^2]$$

$$= \mathbb{E}_{\boldsymbol{x}}[(f(\boldsymbol{x}) - \alpha(\boldsymbol{x}))^2] - 2\mathbb{E}_{\boldsymbol{x}}[(f(\boldsymbol{x}) - \alpha(\boldsymbol{x}))(f_{\text{NN}}(\boldsymbol{x};\boldsymbol{\theta}_\alpha^k) - \alpha(\boldsymbol{x}))] + \mathbb{E}_{\boldsymbol{x}}[(f_{\text{NN}}(\boldsymbol{x};\boldsymbol{\theta}_\alpha^k) - \alpha(\boldsymbol{x}))^2]$$

$$= \mathbb{E}_{\boldsymbol{x}}[(f(\boldsymbol{x}) - \alpha(\boldsymbol{x}))^2] - 2\rho + \mathbb{E}_{\boldsymbol{x}}[(f_{\text{NN}}(\boldsymbol{x};\boldsymbol{\theta}_\alpha^k) - \alpha(\boldsymbol{x}))^2]$$

$$\geq \mathbb{E}_{\boldsymbol{x}}[(f(\boldsymbol{x}) - \alpha(\boldsymbol{x}))^2] - 2\sqrt{\epsilon}\sqrt{\mathbb{E}_{\boldsymbol{x}}[(f_{\text{NN}}(\boldsymbol{x};\boldsymbol{\theta}_\alpha^k) - \alpha(\boldsymbol{x}))^2]} + \mathbb{E}_{\boldsymbol{x}}[(f_{\text{NN}}(\boldsymbol{x};\boldsymbol{\theta}_\alpha^k) - \alpha(\boldsymbol{x}))^2]$$

$$\geq \mathbb{E}_{\boldsymbol{x}}[(f(\boldsymbol{x}) - \alpha(\boldsymbol{x}))^2] - \epsilon$$

$$= \|f - \alpha\|_{L^2(\mu_\mathcal{X})}^2 - \epsilon$$

where in the last line we optimize over the quantity $\mathbb{E}_{\boldsymbol{x}}[(f_{\text{NN}}(\boldsymbol{x};\boldsymbol{\theta}_\alpha^k) - \alpha(\boldsymbol{x}))^2]$. $\square$

### A.3 Remark: relation to bound based on cross-predictability, and efficiently verifying $G$-alignment is small

In [AS20], a similar bound to Theorem A.4 was proved. The first main difference is that bound of [AS20] bound applied only to learning functions with binary output alphabet, $\{+1, -1\}$. The second difference is that [AS20] clips the gradient of the loss instead of clipping the gradient of the network. The third difference is that the bound of [AS20] was in terms of the cross-predictability, instead of the $G$-alignment of Definition A.1:

**Definition A.7** (Cross-predictability). *For any distribution over the inputs $\mu_\mathcal{X} \in \mathcal{P}(\mathcal{X})$ and any distribution over functions $\mu_\mathcal{F} \in \mathcal{P}(L^2(\mu_\mathcal{X}))$, the cross-predictability is*

$$\mathcal{CP}(\mu_\mathcal{F}, \mu_\mathcal{X}) = \mathbb{E}_{f,f'\sim\mu_\mathcal{F}}[\langle f, f'\rangle_{L^2(\mu_\mathcal{X})}^2].$$

Nevertheless, we show that the cross-predictability is an upper bound on the alignment.

**Lemma A.8.** *For any $\mu_\mathcal{X}$ and $\mu_\mathcal{F}$, we have $\mathcal{C}(\mu_\mathcal{F}, \mu_\mathcal{X}) \leq \sqrt{\mathcal{CP}(\mu_\mathcal{F}, \mu_\mathcal{X})}$.*

*Proof.* For any $h \in L^2(\mu_\mathcal{X})$, we show $\mathbb{E}_{f\sim\mu_\mathcal{F}}[\langle f, h\rangle^2] \leq \sqrt{\mathcal{CP}(\mu_\mathcal{F}, \mu_\mathcal{X})}\|h\|_{L^2(\mu_\mathcal{X})}^2$. In the following, let $f, f' \sim \mu_\mathcal{F}$ and $\boldsymbol{x}, \boldsymbol{x}' \sim \mu_\mathcal{X}$ be independent. By (a) a tensorization trick for $\boldsymbol{x}$, (b)

Cauchy-Schwarz, (c) the tensorization trick for $f$, and (d) reversing the tensorization trick for $\boldsymbol{x}$,

$$
\begin{aligned}
\mathbb{E}_{f \sim \mu_{\mathcal{F}}}[\langle f, h \rangle^2] &= \mathbb{E}_f[\mathbb{E}_{\boldsymbol{x}}[f(\boldsymbol{x})h(\boldsymbol{x})]^2] \\
&\overset{(a)}{=} \mathbb{E}_{f,\boldsymbol{x},\boldsymbol{x}'}[f(\boldsymbol{x})f(\boldsymbol{x}')h(\boldsymbol{x})h(\boldsymbol{x}')] \\
&= \mathbb{E}_{\boldsymbol{x},\boldsymbol{x}'}[\mathbb{E}_f[f(\boldsymbol{x})f(\boldsymbol{x}')]h(\boldsymbol{x})h(\boldsymbol{x}')] \\
&\overset{(b)}{\leq} \sqrt{\mathbb{E}_{\boldsymbol{x},\boldsymbol{x}'}[\mathbb{E}_f[f(\boldsymbol{x})f(\boldsymbol{x}')]^2]}\sqrt{\mathbb{E}_{\boldsymbol{x},\boldsymbol{x}'}[h(\boldsymbol{x})^2 h(\boldsymbol{x}')^2]} \\
&= \sqrt{\mathbb{E}_{\boldsymbol{x},\boldsymbol{x}'}[\mathbb{E}_f[f(\boldsymbol{x})f(\boldsymbol{x}')]^2]}\,\mathbb{E}_{\boldsymbol{x}}[h(\boldsymbol{x})^2] \\
&\overset{(c)}{=} \sqrt{\mathbb{E}_{\boldsymbol{x},\boldsymbol{x}',f,f'}[f(\boldsymbol{x})f(\boldsymbol{x}')f'(\boldsymbol{x})f'(\boldsymbol{x}')]}\,\mathbb{E}_{\boldsymbol{x}}[h(\boldsymbol{x})^2] \\
&\overset{(d)}{=} \sqrt{\mathbb{E}_{f,f'}[\mathbb{E}_{\boldsymbol{x}}[f(\boldsymbol{x})f'(\boldsymbol{x})]^2]}\,\mathbb{E}_{\boldsymbol{x}}[h(\boldsymbol{x})^2] \\
&= \sqrt{\mathcal{CP}(\mu_{\mathcal{F}}, \mu_{\mathcal{X}})}\|h\|^2_{L^2(\mu_{\mathcal{X}})}.
\end{aligned}
$$

$\square$

Interestingly, the above lemma provides an algorithmically-efficient way to verify that the $G$-alignment of a function is small. Namely, the "$G$-cross-predictability", is an upper bound on the $G$-alignment.

**Corollary A.9** (Efficiently verifying $G$-alignment is small)**.** *For any compact group $G$, any $G$-invariant distribution $\mu_{\mathcal{X}} \in \mathcal{P}(\mathcal{X})$, and any $\mu_{\mathcal{F}} \in \mathcal{P}(L^2(\mu_{\mathcal{X}}))$, we have*

$$
\mathcal{C}((\mu_{\mathcal{F}}, \mu_{\mathcal{X}}); G) \leq \sqrt{\mathcal{CP}((\mu_{\mathcal{F}}, \mu_X); G)} := \sqrt{\mathbb{E}_{f,f' \sim \mu_{\mathcal{F}}, g,g' \sim \mu_G}[\mathbb{E}_{\boldsymbol{x} \sim \mu_{\mathcal{X}}}[f(g(\boldsymbol{x}))f'(g'(\boldsymbol{x}))]^2]}
$$

The right-hand-side in this inequality can be approximated efficiently by taking a large enough empirical sample of $f$, $f'$ and $g$, $g'$ and $\boldsymbol{x}$. By McDiarmid's inequality, the plug-in empirical estimate will concentrate well for large enough number of samples assuming that the functions $f \sim \mu_{\mathcal{F}}$ are bounded. Thus, given a distribution of functions $\mu_{\mathcal{F}}$ and covariates $\mu_{\mathcal{X}}$, one can empirically show that no $G$-equivariant network can learn them.

However, Corollary A.9 is loose, which is why we prove our bounds in terms of the $G$-alignment instead of the $G$-cross-predictability, and means that we can obtain tighter bounds than [AS20] for learning sparse parities (although with a different scheme for clipping the gradients).

**Lemma A.10.** *Let $\mu_{\mathcal{X}} = \text{Unif}[\mathcal{H}_d]$. Let $k \in \{0, \ldots, d\}$, and let $\mu_{\mathcal{F}}$ be the uniform distribution on $\{\chi_S : S \subseteq [d], |S| = k\}$, where $\chi_S : \mathcal{H}_d \to \{+1, -1\}$ is given by $\chi_S(\boldsymbol{x}) = \prod_{i \in S} x_i$. Then*

$$
\binom{d}{k}^{-1} = \mathcal{C}(\mu_{\mathcal{F}}, \mu_{\mathcal{X}}) \ll \sqrt{\mathcal{CP}(\mu_{\mathcal{F}}, \mu_{\mathcal{X}})} = \binom{d}{k}^{-1/2}
$$

*Proof.* Let $S, S' \subseteq [d]$ be independent subsets of size $|S| = |S'| = k$. The cross-predictability is

$$
\mathcal{CP}(\mu_{\mathcal{F}}, \mu_{\mathcal{X}}) = \mathbb{E}_{S,S'}[\langle \chi_S, \chi_{S'} \rangle^2_{L^2(\mathcal{H}_d)}] = \mathbb{E}_{S,S'}[\delta_{S,S'}] = \binom{d}{k}^{-1}.
$$

On the other hand, the alignment of $\mu_{\mathcal{F}}$ is:

$$
\mathcal{C}(\mu_{\mathcal{F}}, \mu_{\mathcal{X}}) = \sup_h \mathbb{E}_S[\langle \chi_S, h \rangle^2] = \sup_h \binom{d}{k}^{-1} \sum_{S'' \subseteq [d], |S''| = k} \hat{h}(S'')^2 \leq \binom{d}{k}^{-1},
$$

where the last line is by Parseval's theorem. $\square$

## A.4 Remark: alternative proof using the statistical query lens

It is possible to prove the impossibility results for learning with GD by using arguments from the Statistical Query literature (see e.g., [Kea98, Rey20]). In particular, (GD) fits under the Correlational

Statistical Query (CSQ) model of computation because it accesses the data distribution $\mathcal{D}$ only through noisy evaluations of $\boldsymbol{g}_{\mathcal{D}}(\boldsymbol{\theta})$, where

$$\boldsymbol{g}_{\mathcal{D}}(\boldsymbol{\theta}) = \mathbb{E}_{(\boldsymbol{x},y)\sim\mathcal{D}}[y(\Pi_{B(0,R)}\nabla_{\boldsymbol{\theta}}f_{\mathsf{NN}}(\boldsymbol{x};\boldsymbol{\theta}))] - \mathbb{E}_{(\boldsymbol{x},y)\sim\mathcal{D}}[f_{\mathsf{NN}}(\boldsymbol{x};\boldsymbol{\theta})(\Pi_{B(0,R)}\nabla_{\boldsymbol{\theta}}f_{\mathsf{NN}}(\boldsymbol{x};\boldsymbol{\theta}))],$$

for different $\boldsymbol{\theta}^k$. The left-hand term is an expectation over the distribution $\mathcal{D}$ of a bounded function times $y$, since $\|\Pi_{B(0,R)}\nabla_{\boldsymbol{\theta}}f_{\mathsf{NN}}(\boldsymbol{x};\boldsymbol{\theta})\|^2 \leq R^2$. And the right-hand term is known since we may assume the distribution $\mu_{\mathcal{X}}$ is known. Therefore each noisy evaluation of $\boldsymbol{g}_{\mathcal{D}}(\boldsymbol{\theta})$ can be made with one CSQ [BIK90, BF02].[12]

We provide a sketch of the proof of the impossibility result using the CSQ formalism. By the $G$-equivariance of the GD algorithm, if GD can learn $f \in L^2(\mu_{\mathcal{X}})$ via statistical queries on the distribution $\mathcal{D}_f$, then for any $g \in G$ it can learn any $f \circ g$ via statistical queries on the distribution $\mathcal{D}_{f \circ g}$. But, if $\{\mathcal{D}_{f \circ g}\}_{g \in G}$ has high CSQ dimension then this is impossible, and therefore this proves that GD cannot efficiently learn the function $f$. The high CSQ dimension can ensured if the $G$-alignment of Definition 1.3 is small. For further comparison between lower bounds based on CSQ and lower bounds based on a junk-flow argument, see [AS20].

## B   Characterization of weak-learnability by GD

In this section, we give the deferred proofs from Section 3.1, characterizing weak-learnability for functions on the Boolean hypercube and functions on the unit sphere by FC networks with i.i.d. symmetric and i.i.d. initialization, respectively.

### B.1   Functions on Boolean hypercube: proof of Theorem 3.4

We first prove Theorem 3.4, which concerns functions on the hypercube, learned by FC networks with i.i.d. symmetric initialization. We show the impossibility and achievability results separately.

Recall that for functions $f : \mathcal{H}_d \to \mathbb{R}$ we may write

$$f(\boldsymbol{x}) = \sum_{S \subseteq [d]} \hat{f}(S)\chi_S(\boldsymbol{x}),$$

where for each $S \subseteq [d]$, we define the monomial $\chi_S : \mathcal{H}_d \to \{+1, -1\}$ by

$$\chi_S(\boldsymbol{x}) = \prod_{i \in S} x_i$$

and the Fourier coefficient

$$\hat{f}(S) = \mathbb{E}_{\boldsymbol{x}\sim\mathcal{H}_d}[f(\boldsymbol{x})\chi_S(\boldsymbol{x})].$$

These monomial functions form an orthogonal basis over the Boolean hypercube, which we will use throughout this section, i.e., for any $S, S' \subseteq [d]$,

$$\mathbb{E}_{\boldsymbol{x}\sim\mathcal{H}_d}[\chi_S(\boldsymbol{x})\chi_{S'}(\boldsymbol{x})] = \delta_{S,S'}.$$

#### B.1.1   Proof of impossibility result of Theorem 3.4

The impossibility result is proved using Theorem 3.1, and the fact from Proposition 2.5 that GD-training of FC networks with i.i.d. symmetric initialization is $G_{\mathrm{sign,perm}}$-equivariant. First, recall the computation of the $G_{\mathrm{sign,perm}}$-alignment.

**Lemma B.1** (Restatement of Lemma 3.5). *For any $f : \mathcal{H}_d \to \mathbb{R}$,*

$$\mathcal{C}((f, \mathcal{H}_d); G_{\mathrm{sign,perm}}) = \max_{k \in [d]} \binom{d}{k}^{-1} \sum_{\substack{S \subseteq [d] \\ |S|=k}} \hat{f}(S)^2.$$

---

[12]Although correlational statistical queries are typically defined with adversarially-chosen noise, analogous correlational statistical query lower bounds can be proved with random noise (see, e.g., Section 3.2 of [Boi20]).

**Remark B.2** (Alternative proof of Lemma B.1 using the Schur orthogonality theorem). *The same representation theory approach used to prove Lemma 3.7 for the $G_{\mathrm{rot}}$-alignment could have been used to prove Lemma B.1 for $G_{\mathrm{sign,perm}}$, instead of the ad hoc calculation. One would decompose the functions over the Boolean hypercube as $L^2(\mathcal{H}_d) = \oplus_{l=0}^{d}\mathcal{W}_{d,l}$, where $\mathcal{W}_{d,l}$ is the subspace of homogeneous degree-$l$ polynomials. The representations of $G_{\mathrm{sign,perm}}$ given by $\Phi_l(g) : \mathcal{W}_{d,l} \to \mathcal{W}_{d,l}$ mapping $\Phi_l(g)f = f \circ g^{-1}$ are all irreducible and inequivalent for distinct $l \neq l'$, and $\dim(\mathcal{W}_{d,l}) = \binom{d}{l}$.*

Using the previous lemma, we may relate a high $G_{\mathrm{sign,perm}}$-alignment to the existence of a non-negligible Fourier coefficient of $O(1)$ or $d - O(1)$ degree.

**Lemma B.3.** *Suppose $\{f_d\}_{d\in\mathbb{N}}$ is such that $\mathcal{C}((f_d, \mathcal{H}_d); G_{\mathrm{sign,perm}}) \geq \Omega(d^{-C})$ for some constant $C > 0$. Then there is a constant $C' > 0$ such that for each $d$ there is $S_d \subseteq [d]$ with $|S_d| \leq C'$ or $|S_d| \geq d - C'$ and $|\hat{f}_d(S_d)| \geq \Omega(d^{-C'})$.*

*Proof.* By Lemma B.1, we have

$$
\mathcal{C}_d((f_d, \mathcal{H}_d); G_{\mathrm{sign,perm}}) \leq \sum_{\substack{S \subseteq [d] \\ \min(|S|, d-|S|) \leq C}} \hat{f}_d(S)^2 \binom{d}{|S|}^{-1} + \sum_{\substack{S \subseteq [d] \\ \min(|S|, d-|S|) > C}} \hat{f}_d(S)^2 \binom{d}{|S|}^{-1}
$$

$$
\leq \sum_{\substack{S \subseteq [d] \\ \min(|S|, d-|S|) \leq C}} \hat{f}_d(S)^2 + \sum_{\substack{S \subseteq [d] \\ \min(|S|, d-|S|) > C}} \hat{f}_d(S)^2 \binom{d}{C+1}^{-1}
$$

$$
\leq \sum_{\substack{S \subseteq [d] \\ \min(|S|, d-|S|) \leq C}} \hat{f}_d(S)^2 + O(d^{-C-1})
$$

$$
\leq \max_{\substack{S \subseteq [d] \\ \min(|S|, d-|S|) \leq C}} d^C \hat{f}_d(S)^2 + O(d^{-C-1}).
$$

So if $\mathcal{C}((f_d, \mathcal{H}_d); G_{\mathrm{sign,perm}}) \geq \Omega(d^{-C})$, then

$$
\max_{\substack{S \subseteq [d] \\ \min(|S|, d-|S|) \leq C}} \hat{f}_d(S)^2 \geq \Omega(d^{-2C}).
$$

Letting $C' = 2C$ proves the lemma. $\qquad\square$

Now we are ready to prove the impossibility result of Theorem 3.4.

*Proof of impossibility result of Theorem 3.4.* Suppose that $\{f_d, \mathcal{H}_d\}$ is efficiently weak-learnable by GD-trained FC networks with symmetric initialization; let $\{f_{\mathsf{NN},d}, \mu_{\boldsymbol{\theta},d}, \eta_d, k_d, R_d\tau_d\}$ be the sequence of networks, initializations, and GD hyperparameters satisfying Definition 3.3, and in particular (2) for some constant $C > 0$. By Proposition 2.5, $(f_{\mathsf{NN},d}, \mu_{\boldsymbol{\theta},d})$-GD training is $G_{perm,sign}$-equivariant, so by Theorem 3.1,

$$
\mathbb{P}[\|f_d - f_{\mathsf{NN},d}(\cdot; \boldsymbol{\theta}_d)\|_{L^2(\mu_d)}^2 \leq \|f_d\|_{L^2(\mu_d)}^2 - d^{-C}] \leq \frac{\eta_d R_d \sqrt{k_d \mathcal{C}_d}}{2\tau_d} + \frac{\mathcal{C}_d}{d^{-C}} \leq O(d^{4c}\sqrt{\mathcal{C}_d} + d^C \mathcal{C}_d),
$$

where $\mathcal{C}_d = \mathcal{C}((f_d, \mathcal{H}_d); G_{\mathrm{sign,perm}})$, and we have used the bounds on the parameters guaranteed by Definition 3.3.

On the other hand, since the network weak-learns (2), the right-hand side must be greater than or equal to $9/10$, so $\mathcal{C}_d \geq \Omega(d^{-\max(8c,C)})$. By Lemma B.3, this implies there is a constant $C'$ and sequence $\{S_d\}$ such that $\min(|S_d|, d - |S_d|) \leq C'$ and $|\hat{f}_d(S_d)| \geq \Omega(d^{-C'})$. $\qquad\square$

### B.1.2 Example: a concrete function that is not efficiently weak-learnable

As a concrete example of a bounded function on the Boolean hypercube that is not efficiently weak-learnable by (GD), which was previously not known, consider the function $f \in L^2(\mathcal{H}_d)$:

$$f(\boldsymbol{x}) = \begin{cases} 0, & |\{j : x_j = 1\}| \equiv 0 \pmod 2 \\ 1, & |\{j : x_j = 1\}| \equiv 1 \pmod 4 \\ -1, & |\{j : x_j = 1\}| \equiv 3 \pmod 4 \end{cases}.$$

This function is invariant to permutations of the input, so its $G_{\mathrm{perm}}$-alignment is large. Therefore previous work that considers permutation symmetry of GD training of networks [SSS17, MS20, AS20, ACHM22] cannot imply that this is a hard function to learn. Nevertheless, we will show that its $G_{\mathrm{sign}}$-alignment is small. First, let us rewrite $f$ as:

$$f(\boldsymbol{x}) = \mathrm{Im}(\zeta(\boldsymbol{x})), \text{ where } \zeta(\boldsymbol{x}) = \exp((\pi i/2) \sum_{j=1}^d (x_j + 1)/2)$$

Let $\boldsymbol{s}, \boldsymbol{x} \sim \mathcal{H}_d$ and take the supremum over $h \in L^2(\mathcal{H}_d)$ such that $\|h\|^2 = 1$ in the calculations below. By (a) permutation invariance of $f$, (b) using that $\mathrm{Im}(z)^2 \leq |z|^2$ for any complex $z \in \mathbb{C}$, (c) a tensorization trick,

$$\mathcal{C}((f, \mathcal{H}_d); G_{\mathrm{sign,perm}})$$

$$\overset{(a)}{=} \mathcal{C}((f, \mathcal{H}_d); G_{\mathrm{sign}})$$

$$= \sup_h \mathbb{E}_{\boldsymbol{s}}[\mathbb{E}_{\boldsymbol{x}}[f(\boldsymbol{x} \odot \boldsymbol{s})h(\boldsymbol{x})]^2]$$

$$\overset{(b)}{\leq} \sup_h \mathbb{E}_{\boldsymbol{s}}[|\mathbb{E}_{\boldsymbol{x}}[\zeta(\boldsymbol{x} \odot \boldsymbol{s})h(\boldsymbol{x})]|^2]$$

$$\overset{(c)}{\leq} \sup_h \mathbb{E}_{\boldsymbol{s}}[\mathbb{E}_{\boldsymbol{x}, \boldsymbol{x}'}[\zeta(\boldsymbol{x} \odot \boldsymbol{s})\bar{\zeta}(\boldsymbol{x}' \odot \boldsymbol{s})h(\boldsymbol{x})h(\boldsymbol{x}')]]$$

$$\leq \sup_h \mathbb{E}_{\boldsymbol{x}, \boldsymbol{x}'}[h(\boldsymbol{x})h(\boldsymbol{x}') \mathbb{E}_{\boldsymbol{s}}[\zeta(\boldsymbol{x} \odot \boldsymbol{s})\bar{\zeta}(\boldsymbol{x}' \odot \boldsymbol{s})]]$$

$$= \sup_h \mathbb{E}_{\boldsymbol{x}, \boldsymbol{x}'}[h(\boldsymbol{x})h(\boldsymbol{x}') \mathbb{E}_{\boldsymbol{s}}[\exp((\pi i/2)(\sum_{j=1}^d (x_j s_j + 1)/2 - \sum_{j=1}^d (x_j' s_j + 1)/2)]]$$

$$= \sup_h \mathbb{E}_{\boldsymbol{x}, \boldsymbol{x}'}[h(\boldsymbol{x})h(\boldsymbol{x}') \prod_{j=1}^d \mathbb{E}_{s_j}[\exp((\pi i/2)(s_j(x_j - x_j')/2)]]$$

$$= \sup_h \mathbb{E}_{\boldsymbol{x}, \boldsymbol{x}'}[h(\boldsymbol{x})h(\boldsymbol{x}') \prod_{j=1}^d (\frac{1}{2}\exp((\pi i/2)((x_j - x_j')/2)) + \frac{1}{2}\exp((\pi i/2)((-x_j + x_j')/2)))]$$

$$= \sup_h \mathbb{E}_{\boldsymbol{x}, \boldsymbol{x}'}[h(\boldsymbol{x})h(\boldsymbol{x}') \prod_{j=1}^d \delta_{x_j x_j'}]$$

$$= \sup_h \mathbb{E}_{\boldsymbol{x}, \boldsymbol{x}'}[h(\boldsymbol{x})h(\boldsymbol{x}')\delta_{\boldsymbol{x}, \boldsymbol{x}'}]$$

$$= \sup_h \mathbb{E}_{\boldsymbol{x}}[h(\boldsymbol{x})^2]/2^d$$

$$= \frac{1}{2^d}.$$

So, $\mathcal{C}((f, \mathcal{H}_d); G_{\mathrm{sign,perm}}) \leq 1/2^d$, which is negligible. Plugging this into Theorem 3.1 implies that $f$ cannot be efficiently learned by FC networks with sign-flip symmetric initialization trained by (GD).

### B.1.3 Proof of achievability result of Theorem 3.4

We prove the converse achievability result with the following training setup. Consider a two-layer FC network architecture with parameters $\boldsymbol{\theta} = (\boldsymbol{W}, \boldsymbol{a})$ where $\boldsymbol{W} \in \mathbb{R}^{d \times m}$ and $\boldsymbol{a} \in \mathbb{R}^m$ and

$f_{\mathsf{NN}}(\boldsymbol{x}; \boldsymbol{\theta}) = \boldsymbol{a}^\top \sigma(\boldsymbol{W}\boldsymbol{x})$, where the activation function $\sigma$ is a bump function applied elementwise:

$$\sigma(z) = \begin{cases} 1, & -0.5 \le z \le 1.5 \\ 0, & z \notin (-0.5, 1.5) \end{cases}.$$

Set the initialization distribution $\mu_{\boldsymbol{\theta}} = \mu_{\boldsymbol{W}} \times \mu_{\boldsymbol{a}}$, where $\mu_{\boldsymbol{a}} = \delta_{\boldsymbol{0}}$, and $\boldsymbol{W} \sim \mu_w^{\otimes(m \times d)}$, where

$$\mu_w = (1-p)\delta_0 + \frac{p}{2}\delta_1 + \frac{p}{2}\delta_{-1},$$

for some parameter $p \in [0,1]$. Note that the above is a FC architecture with an i.i.d. sign-flip symmetric initialization.[13] We obtain the following guarantee for training the network with GD.

**Lemma B.4.** *Let $f : \mathcal{H}_d \to \mathbb{R}$, and let $\mathcal{D} \in \mathcal{P}(\mathcal{H}_d \times \mathbb{R})$ be the distribution of $(\boldsymbol{x}, f(\boldsymbol{x}))$ where $\boldsymbol{x} \sim \mathcal{H}_d$. Let $s \in \{0, \ldots, d\}$. Define $r_s = \max_{S, |S|=s} |\hat{f}(S)|/(d^2\binom{d}{s})$.*

*Consider training the above architecture and initialization with parameter $p = s/d$, width $m \ge 100/(r_s)^4$. If we run (GD) for one step, with learning rate $0 < \eta < 1/(4m)$ and noise level $\tau \le \eta(r_s)^4/(100m^2 d)$, then with probability at least $9/10$ we have*

$$\ell_{\mathcal{D}}(\boldsymbol{\theta}^1) \le \|f\|_{L^2(\mathcal{H}_d)}^2 - \eta(r_s)^4/4.$$

*Proof.* Let us prove that the loss is smooth in a small ball around initialization: For any $\boldsymbol{x} \in \mathcal{H}_d$ and $\tilde{\boldsymbol{W}}$ such that $\|\tilde{\boldsymbol{W}} - \boldsymbol{W}^0\|_\infty \le 1/(4d)$, we have $\tilde{\boldsymbol{W}}\boldsymbol{x} \in \mathbb{Z} + [-1/4, 1/4]$ almost surely. Therefore $\sigma'(\tilde{\boldsymbol{W}}\boldsymbol{x}) = \sigma''(\tilde{\boldsymbol{W}}\boldsymbol{x}) = \boldsymbol{0}$. So for any $\tilde{\boldsymbol{a}}$, one can show that the loss is smooth at $\tilde{\boldsymbol{\theta}} = (\tilde{\boldsymbol{W}}, \tilde{\boldsymbol{a}})$:

$$\nabla_{\boldsymbol{\theta}}^2 \ell_{\mathcal{D}}(\tilde{\boldsymbol{\theta}}) \lesssim m\|\sigma\|_\infty^2 \boldsymbol{I} \lesssim m\boldsymbol{I}. \tag{8}$$

Let us lower-bound the magnitude of the gradient. Since $\sigma'(\boldsymbol{W}^0\boldsymbol{x}) = 0$, we have

$$\|\nabla_{\boldsymbol{W}} \ell_{\mathcal{D}}(\boldsymbol{\theta}^0)\|_F^2 = \|\boldsymbol{0}\|_F^2 = 0.$$

However, the gradient with respect to $\boldsymbol{a}$ is nonzero. Writing $\boldsymbol{W}^0 = [\boldsymbol{w}_1^0, \ldots, \boldsymbol{w}_m^0]$, since $\boldsymbol{a}^0 = \boldsymbol{0}$,

$$\|\nabla_{\boldsymbol{a}} \ell_{\mathcal{D}}(\boldsymbol{\theta}^0)\|^2 = \|\mathbb{E}_{\boldsymbol{x}}[(f(\boldsymbol{x}) - (\boldsymbol{a}^0)^\top \sigma(\boldsymbol{W}^0\boldsymbol{x}))\sigma(\boldsymbol{W}^0\boldsymbol{x})]\|^2 = \sum_{i \in [m]} \mathbb{E}_{\boldsymbol{x}}[f(\boldsymbol{x})\sigma(\langle \boldsymbol{w}_i^0, \boldsymbol{x}\rangle)]^2. \tag{9}$$

Our main claim is that each term in this sum is lower-bounded in expectation.

**Claim B.5.** *For any $i \in [m]$, $\mathbb{E}_{\boldsymbol{w}_i^0}[\mathbb{E}_{\boldsymbol{x}}[f(\boldsymbol{x})\sigma(\langle \boldsymbol{w}_i^0, \boldsymbol{x}\rangle)]^2] \ge \frac{1}{d^4}\binom{d}{s}^{-2} \sum_{S, |S|=s} |\hat{f}(S)|^2 \ge (r_s)^2$.*

*Proof of claim.* For ease of notation, let $\boldsymbol{w} \sim \mathcal{L}(\boldsymbol{w}_i^0)$. Write $\boldsymbol{w} = \boldsymbol{a} \odot \boldsymbol{1}_S$ for $\boldsymbol{a} \sim \mathcal{H}_d$ and $S \subset [d]$ where each element of $[d]$ is in $S$ independently with probability $p$. Then

$$\mathbb{E}_{\boldsymbol{w}}[\mathbb{E}_{\boldsymbol{x}}[f(\boldsymbol{x})\sigma(\langle \boldsymbol{w}, \boldsymbol{x}\rangle)]^2] \tag{10}$$
$$= \mathbb{E}_{S, \boldsymbol{a}}[\mathbb{E}_{\boldsymbol{x}}[(-1)^{\lfloor |S|/2 \rfloor} \operatorname{sgn}(\hat{f}(S))\chi_S(\boldsymbol{a})f(\boldsymbol{x})\sigma(\langle \boldsymbol{a} \odot \boldsymbol{1}_S, \boldsymbol{x}\rangle)]^2]$$
$$\ge \mathbb{E}_{S, \boldsymbol{a}, \boldsymbol{x}}[(-1)^{\lfloor |S|/2 \rfloor} \operatorname{sgn}(\hat{f}(S))\chi_S(\boldsymbol{a})f(\boldsymbol{x})\sigma(\langle \boldsymbol{a} \odot \boldsymbol{1}_S, \boldsymbol{x}\rangle)]^2$$
$$= (\sum_{S' \subseteq [d]} \mathbb{E}_S[(-1)^{\lfloor |S|/2 \rfloor} \operatorname{sgn}(\hat{f}(S))\hat{f}(S') \mathbb{E}_{\boldsymbol{a}, \boldsymbol{x}}[\chi_S(\boldsymbol{a})\chi_{S'}(\boldsymbol{x})\sigma(\langle \boldsymbol{a} \odot \boldsymbol{1}_S, \boldsymbol{x}\rangle)]])^2$$
$$= (\sum_{S' \subseteq [d]} \mathbb{E}_S[(-1)^{\lfloor |S|/2 \rfloor} \operatorname{sgn}(\hat{f}(S))\hat{f}(S')c_{S,S'}])^2, \tag{11}$$

where in the last line we define $c_{S,S'} = \mathbb{E}_{\boldsymbol{a}, \boldsymbol{x}}[\chi_S(\boldsymbol{a})\chi_{S'}(\boldsymbol{x})\sigma(\langle \boldsymbol{a} \odot \boldsymbol{1}_S, \boldsymbol{x}\rangle)]$. If $S \not\subseteq S'$, then we have $c_{S,S'} = 0$, since $\mathbb{E}_{\boldsymbol{a}}[\chi_S(\boldsymbol{a})\sigma(\langle \boldsymbol{a} \odot \boldsymbol{1}_S, \boldsymbol{x}\rangle)]$ depends only on $\boldsymbol{x}_S$ and not on $\boldsymbol{x}_{[d]\backslash S}$. Furthermore, for any $S' \subseteq S \subseteq [d]$,

$$c_{S,S'} = \mathbb{E}_{\boldsymbol{a}, \boldsymbol{x}}[\chi_S(\boldsymbol{a})\chi_{S'}(\boldsymbol{x})\delta(\langle \boldsymbol{a} \odot \boldsymbol{1}_S, \boldsymbol{x}\rangle \in \{0, 1\})]$$

---

[13]To fully ensure i.i.d. initialization for both layers, we could take $\mu_{\boldsymbol{a}} = \mu_w^{\otimes m}$. However it is notationally simpler to take $\mu_{\boldsymbol{a}} = \delta_{\boldsymbol{0}}$, and furthermore any bounded initialization works for $\mu_{\boldsymbol{a}}$, because the same random features analysis goes through.

$$= \mathbb{E}_{\boldsymbol{a},\boldsymbol{x}}[\chi_S(\boldsymbol{a})\chi_{S'}(\boldsymbol{x}) \sum_{k=-d}^{d} \delta(\langle \boldsymbol{a} \odot \mathbf{1}_S, \boldsymbol{x} \rangle \in \{0,1\})\delta(\langle \boldsymbol{a} \odot \mathbf{1}_{S'}, \boldsymbol{x} \rangle = k)]$$

$$= \mathbb{E}_{\boldsymbol{a},\boldsymbol{x}}[\chi_S(\boldsymbol{a})\chi_{S'}(\boldsymbol{x}) \sum_{k=-d}^{d} \delta(\langle \boldsymbol{a} \odot \mathbf{1}_{S \setminus S'}, \boldsymbol{x} \rangle \in \{-k, -k+1\})\delta(\langle \boldsymbol{a} \odot \mathbf{1}_{S'}, \boldsymbol{x} \rangle = k)].$$

Notice that $\langle \boldsymbol{a} \odot \mathbf{1}_{S'}, \boldsymbol{x} \rangle = \sum_{i \in S'} a_i x_i = |S'| - 2|\{i \in S' : a_i x_i = -1\}|$. Also, $\chi_{S'}(\boldsymbol{a})\chi_{S'}(\boldsymbol{x}) = (-1)^{|\{i \in S' : a_i x_i = -1\}|}$. Therefore, $\chi_{S'}(\boldsymbol{a})\chi_{S'}(\boldsymbol{x}) = (-1)^{(|S'| - \langle \boldsymbol{a} \odot \mathbf{1}_{S'}, \boldsymbol{x} \rangle)/2}$, so plugging this into the above,

$$c_{S,S'} = \mathbb{E}_{\boldsymbol{a},\boldsymbol{x}}[\chi_{S \setminus S'}(\boldsymbol{a}) \sum_{k=-d}^{d} (-1)^{(|S'|-k)/2}\delta(\langle \boldsymbol{a} \odot \mathbf{1}_{S \setminus S'}, \boldsymbol{x} \rangle \in \{-k, -k+1\})\delta(\langle \boldsymbol{a} \odot \mathbf{1}_{S'}, \boldsymbol{x} \rangle = k)]$$

$$= \mathbb{E}_{\boldsymbol{a}}[\chi_{S \setminus S'}(\boldsymbol{a}) \sum_{k=-d}^{d} (-1)^{(|S'|-k)/2}\mathbb{P}_{\boldsymbol{x}_{S \setminus S'}}[\langle \boldsymbol{a} \odot \mathbf{1}_{S \setminus S'}, \boldsymbol{x} \rangle \in \{-k, -k+1\}]\mathbb{P}_{\boldsymbol{x}_{S'}}[\langle \boldsymbol{a} \odot \mathbf{1}_{S'}, \boldsymbol{x} \rangle = k]]$$

$$= \mathbb{E}_{\boldsymbol{a}}[\chi_{S \setminus S'}(\boldsymbol{a})] \sum_{k=-d}^{d} (-1)^{(|S'|-k)/2}\mathbb{P}_{\boldsymbol{x}_{S \setminus S'}}[\langle \mathbf{1}_{S \setminus S'}, \boldsymbol{x} \rangle \in \{-k, -k+1\}]\mathbb{P}_{\boldsymbol{x}_{S'}}[\langle \mathbf{1}_{S'}, \boldsymbol{x} \rangle = k],$$

where the last line uses the sign-flip symmetry of the distributions $\boldsymbol{a}, \boldsymbol{x} \sim \mathcal{H}_d$. Therefore if $S \neq S'$, we have $c_{S,S'} = 0$. Otherwise, if $S = S'$, we have

$$c_{S,S} = \sum_{k \in \{0,1\}} (-1)^{(|S|-k)/2}\mathbb{P}_{\boldsymbol{x}}[\langle \mathbf{1}_S, \boldsymbol{x} \rangle = k] = (-1)^{\lfloor |S|/2 \rfloor}\mathbb{P}_{\boldsymbol{x}}[\langle \mathbf{1}_S, \boldsymbol{x} \rangle \in \{0,1\}],$$

where we have used that $\langle \mathbf{1}_S, \boldsymbol{x} \rangle$ has the same parity as $|S|$ almost surely over $\boldsymbol{x}$, so either $\mathbb{P}_{\boldsymbol{x}}[\langle \mathbf{1}_S, \boldsymbol{x} \rangle = 0] = 0$ or $\mathbb{P}_{\boldsymbol{x}}[\langle \mathbf{1}_S, \boldsymbol{x} \rangle = 1] = 0$. Plugging this into (11),

$$\mathbb{E}_{\boldsymbol{w}}[\mathbb{E}_{\boldsymbol{x}}[f(\boldsymbol{x})\sigma(\langle \boldsymbol{w}, \boldsymbol{x} \rangle)]^2] = \mathbb{E}_S[|\hat{f}(S)|\mathbb{P}_{\boldsymbol{x}}[\langle \mathbf{1}_S, \boldsymbol{x} \rangle \in \{0,1\}]]^2$$

$$= \mathbb{E}_S[|\hat{f}(S)|\binom{|S|}{\lfloor |S|/2 \rfloor}/2^{|S|}]^2$$

$$\geq \left( \frac{1}{d} \sum_{S,|S|=s} \binom{d}{s}^{-1}|\hat{f}(S)|\binom{|S|}{\lfloor |S|/2 \rfloor}/2^{|S|} \right)^2$$

$$\geq \frac{1}{d^4}\binom{d}{s}^{-2} \sum_{S,|S|=s} |\hat{f}(S)|^2$$

$$\geq (r_s)^2,$$

where recall that $s = pd$ and we use that with probability at least $1/d$ the random set $S$ where each element is included with probability $p$ has size $|S| = s$. $\qquad \square$

Combining the above claim with a Hoeffding bound, we can bound the gradient with high probability.

**Claim B.6.** *Let $E_1$ denote the event that $(r_s)^2/2 \leq \|\nabla_{\boldsymbol{a}} \ell_{\mathcal{D}}(\boldsymbol{\theta}^0)\|^2/m \leq 1$. Then $\mathbb{P}_{\boldsymbol{\theta}^0}[E_1] \geq 19/20$.*

*Proof of claim.* In (9), $\|\nabla_{\boldsymbol{a}} \ell_{\mathcal{D}}(\boldsymbol{\theta}^0)\|^2$ is written as a sum of $m$ i.i.d. random variables of the form $\mathbb{E}_{\boldsymbol{x}}[f(\boldsymbol{x})\sigma(\langle \boldsymbol{w}_i^0, \boldsymbol{x} \rangle)]^2$. These are almost surely in the range $[0,1]$ since $\|f(\boldsymbol{x})\|_\infty \leq 1$ and $\|\sigma\|_\infty \leq 1$. This proves the upper bound almost surely.

For the lower bound, by Hoeffding's inequality,

$$\mathbb{P}[\|\nabla_{\boldsymbol{a}} \ell_{\mathcal{D}}(\boldsymbol{\theta}^0)\|^2/m \leq \mathbb{E}[\|\nabla_{\boldsymbol{a}} \ell_{\mathcal{D}}(\boldsymbol{\theta}^0)\|^2/m] - 5/\sqrt{m}] \leq \exp(-2 \cdot 25) < 1/20.$$

Finally, recall that

$$\mathbb{E}[\|\nabla_{\boldsymbol{a}} \ell_{\mathcal{D}}(\boldsymbol{\theta}^0)\|^2/m] = \mathbb{E}_{\boldsymbol{w}_i}[\mathbb{E}_{\boldsymbol{x}}[f(\boldsymbol{x})\sigma(\langle \boldsymbol{w}_i^0, \boldsymbol{x} \rangle)]^2] \geq (r_s)^2$$

by Claim B.5. Since we have taken $m \geq 100/(r_s)^4$, we have $(r_s)^2 - 5/\sqrt{m} \geq (r_s)^2/2$, which concludes the claim. $\qquad \square$

We also bound the magnitude of the added noise.

**Claim B.7.** *Let $E_2$ denote the event that $\|\boldsymbol{\xi}^0\|_\infty \leq \min(1/(4d), \frac{\eta(r_s)^4}{4m^2 d})$. Then $\mathbb{P}[E_2] \geq 19/20$.*

*Proof of claim.* By a union bound over the Gaussian tail bounds for all $md + m$ entries of $\boldsymbol{\xi}^0$. $\qquad\square$

Condition on events $E_1$ and $E_2$, which hold with probability at least $9/10$ by the above two claims. Since, $\boldsymbol{W}^1 = \boldsymbol{W}^0 + \boldsymbol{\xi}^0$, we have $\|\boldsymbol{W}^1 - \boldsymbol{W}^0\|_\infty \leq 1/(4d)$. By (a) the smoothness in (8), and (b) the learning rate choice $\eta \leq 1/(4m)$, (c) by the above bounds,

$$
\begin{aligned}
\ell_{\mathcal{D}}(\boldsymbol{\theta}^1) &\overset{(a)}{\leq} \ell_{\mathcal{D}}(\boldsymbol{\theta}^0) - \eta\|\nabla_{\boldsymbol{\theta}}\ell_{\mathcal{D}}(\boldsymbol{\theta}^0)\|^2 + |\langle\nabla_{\boldsymbol{\theta}}\ell_{\mathcal{D}}(\boldsymbol{\theta}^0), \boldsymbol{\xi}^0\rangle| + m\|\eta^2\nabla_{\boldsymbol{\theta}}\ell_{\mathcal{D}}(\boldsymbol{\theta}^0) + \boldsymbol{\xi}^0\|^2 \\
&\leq \|f\|_{L^2(\mathcal{H}_d)}^2 - \eta\|\nabla_{\boldsymbol{\theta}}\ell_{\mathcal{D}}(\boldsymbol{\theta}^0)\|^2(1 - 2m\eta) + \sqrt{m}\sqrt{(md+m)}\|\boldsymbol{\xi}^0\|_\infty + 2m(md+m)\|\boldsymbol{\xi}^0\|_\infty^2 \\
&\overset{(b)}{\leq} \|f\|_{L^2(\mathcal{H}_d)}^2 - \frac{\eta}{2}\|\nabla_{\boldsymbol{\theta}}\ell_{\mathcal{D}}(\boldsymbol{\theta}^0)\|^2 + \sqrt{m}\sqrt{(md+m)}\|\boldsymbol{\xi}^0\|_\infty + 2m(md+m)\|\boldsymbol{\xi}^0\|_\infty^2 \\
&\leq \|f\|_{L^2(\mathcal{H}_d)}^2 - \frac{\eta}{4}(r_s)^4,
\end{aligned}
$$

which proves the lemma. $\qquad\square$

Lemma B.4 implies the achievability result of Theorem 3.4:

*Proof of achievability result of Theorem 3.4.* Suppose that $\{f_d\}$ is a sequence of functions $f_d \in L^2(\mathcal{H}_d)$ such that $\|f_d\|_{L^2(\mathcal{H}_d)} \leq 1$ and there is a constant $C > 0$ such that

$$
\max_{\substack{S \subseteq [d] \\ \min(|S|, d-|S|) \leq C}} \hat{f}_d(S)^2 \geq \Omega(d^{-C}).
$$

Then for each $d \in \mathbb{N}$ there is $s_d \in \{0, \ldots, d\}$ such that $\max_{S \subseteq [d], |S| = s_d} |\hat{f}(S)|/(d^2\binom{d}{s_d}) \geq \Omega(d^{-2C-2})$. So by Lemma B.4 there is a FC architecture with width $m \leq O(d^{8C+8})$ such that initializing i.i.d. from a symmetric distribution and GD-training with learning rate $\eta \leq 1$ and noise level $\tau \geq \Omega(d^{-40C-41})$ reaches a loss $\leq \|f_d\|_{L^2(\mathcal{H}_d)}^2 - \Omega(d^{-12C-12})$ with probability at least $9/10$. One can take $R = \text{poly}(d)$, since under events $E_1$ and $E_2$ in the analysis of Lemma B.4 we only optimize in parameter regions with $\text{poly}(d)$-size gradients. $\qquad\square$

## B.2   Functions on unit sphere: proof of Theorem 3.6

We now prove the analogous result for functions on the unit sphere, again proving the impossibility and achievability results separately. We use that we can decompose $L^2(\mathbb{S}^{d-1})$ as

$$
L^2(\mathbb{S}^{d-1}) = \oplus_{l=0}^\infty \mathcal{V}_{d,l},
$$

where $\mathcal{V}_{d,l}$ is the space of degree-$l$ spherical harmonics, which has dimension $\dim(\mathcal{V}_{d,l}) = \frac{2l+d-2}{l}\binom{l+d-3}{l-1}$ (see, e.g., [Hoc12]). Let $\Pi_{\mathcal{V}_{d,l}} : L^2(\mathbb{S}^{d-1}) \to \mathcal{V}_{d,l}$ denote the projection onto the degree-$l$ spherical harmonics.

### B.2.1   Proof of impossibility result of Theorem 3.6

Again, we apply Theorem 3.1 to prove the impossibility result. Since the FC networks are i.i.d. Gaussian-initialized, GD training is $G_{\text{rot}}$-equivariant by Proposition 2.5. So recall the bound on the $G_{\text{rot}}$-alignment proved in the main text.

**Lemma B.8** (Restatement of Lemma 3.7)**.** *Let $f \in L^2(\mathbb{S}^{d-1})$. Then*

$$
\mathcal{C}((f, \mathbb{S}^{d-1}); G_{\text{rot}}) = \max_{l \in \mathbb{Z}_{\geq 0}} \|\Pi_{\mathcal{V}_{d,l}} f\|^2 / \dim(\mathcal{V}_{d,l}).
$$

We may now conclude the proof of impossibility of learning.

*Proof of impossibility result of Theorem 3.6.* Suppose $\{f_d, \mathbb{S}^{d-1}\}$ is efficiently weak-learnable by GD-trained FC networks with Gaussian initialization; let $\{f_{\mathsf{NN},d}, \mu_{\boldsymbol{\theta},d}, \eta_d, k_d, R_d, \tau_d\}$ be the sequence of networks, initializations, and GD hyperparameters achieving (2) for some constant $C$. On the other hand, by Proposition 2.5, GD training is $G_{\mathrm{rot}}$-equivariant, so by Theorem 3.1

$$\mathbb{P}[\|f_d - \tilde{f}_d\|^2_{L^2(\mathbb{S}^{d-1})} \leq \|f_d\|^2_{L^2(\mathbb{S}^{d-1})} - d^{-C}] \leq \frac{\eta_d R_d \sqrt{k_d \mathcal{C}_d}}{2\tau_d} + \frac{\mathcal{C}_d}{d^{-C}} \leq O(d^{4c}\sqrt{\mathcal{C}_d} + d^C \mathcal{C}_d),$$

where $\mathcal{C}_d = \mathcal{C}(f_d; \mathbb{S}^{d-1}, G_{\mathrm{rot}})$. By (2), the right-hand side must be greater than $9/10$, so $\mathcal{C}_d \geq \Omega(d^{-\max(8c,C)})$. By Lemma B.8, this implies that for each $d \in \mathbb{N}$, there is $l_d \in \mathbb{Z}_{\geq 0}$ such that $\|\Pi_{\mathcal{V}_{d,l_d}} f_d\|^2 / \dim(\mathcal{V}_{d,l_d}) \geq \Omega(d^{-\max(8c,C)})$. Since $\dim(\mathcal{V}_{d,l_d}) \geq 1$, this means $\|\Pi_{\mathcal{V}_{d,l_d}} f_d\|^2 \geq \Omega(d^{-\max(8c,C)})$. Now let us prove that $l_d$ is upper-bounded by a constant. Since $\|\Pi_{\mathcal{V}_{d,l_d}} f_d\|^2 \leq \|f_d\|^2 \leq 1$, we must have $\dim(\mathcal{V}_{d,l_d}) \leq O(d^{\max(8c,C)})$. Since $\dim(\mathcal{V}_{d,l}) \geq (d/l) \cdot ((l+d-3)/(l-1))^{l-1}$, it follows that $l_d \leq \min(8c,C)$ for all sufficiently large $d$. $\qquad\square$

## B.2.2 Proof of achievability result of Theorem 3.6

The positive weak learning result can again be achieved by two-layer FC networks. Indeed, in Theorem 1(b) of [GMMM21] it is proved that for any $l \in \mathbb{Z}_{\geq 1}$, a two-layer neural network random features model with $d^{\omega(l)}$ neurons will learn the projection of the target function to the degree-$l$ spherical harmonics. The 2-layer model is given by $f_{\mathsf{NN}}(\boldsymbol{x}; \boldsymbol{\theta}) = \boldsymbol{a}^\top \sigma(\boldsymbol{W}\boldsymbol{x})$, where only $\boldsymbol{a} \in \mathbb{R}^m$ is trained, and $\boldsymbol{W} = [\boldsymbol{w}_1, \ldots, \boldsymbol{w}_m] \in \mathbb{R}^{m \times d}$ is fixed to its initialization. Unfortunately, the first-layer initialization used in [GMMM21] is not Gaussian initialization but rather uniform over the unit sphere $\boldsymbol{w}_i^0 \sim \mathbb{S}^{d-1}$. Therefore, we must modify their analysis, and for simplicity we analyze only one step of GD training on the second-layer weights, when we have $\boldsymbol{w}_i^0 \overset{i.i.d.}{\sim} \mathcal{N}(0, I_d/\sqrt{d})$, and $\boldsymbol{a}^0 = \boldsymbol{0}$.

**Lemma B.9.** *Let $f \in L^2(\mathbb{S}^{d-1})$, and let $\mathcal{D} \in \mathcal{P}(\mathbb{S}^{d-1} \times \mathbb{R})$ be the distribution of $(\boldsymbol{x}, f(\boldsymbol{x}))$, where $\boldsymbol{x} \sim \mathbb{S}^{d-1}$.*

*Let $l \in \mathbb{Z}_{\geq 0}$ and $c > 0$. Suppose that $\|\Pi_{\mathcal{V}_{d,l}} f\|^2 \geq d^{-c}$. Consider training the above architecture and initialization with activation function $\sigma(t) = t^l$, width $m \geq d^{4l+4c+1}$. If we run (GD) for one step, with learning rate $0 < \eta < d^{-8l+8c-2}/m$ and noise level $\tau \leq \eta d^{-8l+8c-3}/(100m^2 d)$, then with probability at least $9/10$ for large enough $d$ we have*

$$\ell_{\mathcal{D}}(\boldsymbol{\theta}^1) \leq \|f\|^2_{L^2(\mathbb{S}^{d-1})} - \eta d^{-8l+8c+2}$$

*Proof.* Similarly to the proof of Lemma B.4, the proof relies on upper-bounding the smoothness of $f_{\mathsf{NN}}$ in a ball around initialization, as well as lower-bounding the gradient at initialization.

First, lower-bound the gradient:

**Claim B.10.** $\mathbb{P}[\|\nabla_{\boldsymbol{a}} \ell_{\mathcal{D}}(\boldsymbol{\theta}^0)\|^2 \geq m d^{-2l-2c-1/2}] \geq 49/50.$

*Proof.*

$$\nabla_{\boldsymbol{a}} \ell_{\mathcal{D}}(\boldsymbol{\theta}^0) = \mathbb{E}_{(\boldsymbol{x},y)\sim\mathcal{D}}[-y\nabla_{\boldsymbol{a}} f_{\mathsf{NN}}(\boldsymbol{x}; \boldsymbol{\theta}^0)] = \mathbb{E}_{\boldsymbol{x}\sim\mathbb{S}^{d-1}}[-f(\boldsymbol{x})\sigma(\boldsymbol{W}^0\boldsymbol{x})]$$

Let $C > 0$ be a constant. For any $i \in [m]$, let $E_i$ be the event that $\|\boldsymbol{w}_i^0\| \in [1 - d^{-C}, 1 + d^{-C}]$. Note $\mathbb{P}[E_i] \geq \Omega(d^{-C})$. And conditioned on the event $E_i$, we have

$$|\mathbb{E}_{\boldsymbol{x}\sim\mathbb{S}^{d-1}}[f(\boldsymbol{x})\sigma(\langle \boldsymbol{w}_i^0, \boldsymbol{x}\rangle) \mid E_i] - \mathbb{E}_{\boldsymbol{x}\sim\mathbb{S}^{d-1}}[f(\boldsymbol{x})\sigma(\langle \frac{\boldsymbol{w}_i^0}{\|\boldsymbol{w}_i^0\|}, \boldsymbol{x}\rangle) \mid E_i]| \leq \|f\| l 2^{l-1} d^{-C} = O(d^{-C}).$$

So letting $\boldsymbol{v}_i^0 = \boldsymbol{w}_i^0/\|\boldsymbol{w}_i^0\|$ we have

$$\mathbb{E}_{\boldsymbol{x}\sim\mathbb{S}^{d-1}}[f(\boldsymbol{x})\sigma(\langle \boldsymbol{w}_i^0, \boldsymbol{x}\rangle) \mid E_i]^2 \geq \mathbb{E}_{\boldsymbol{x}\sim\mathbb{S}^{d-1}}[f(\boldsymbol{x})\sigma(\langle \boldsymbol{v}_i^0, \boldsymbol{x}\rangle)]^2 - O(d^{-C})$$

Finally, by (a) a tensorization trick, (b) the rotational invariance of the distribution $\boldsymbol{v}_i^0$, (c) the Schur orthogonality relations for $G_{\mathrm{rot}}$,

$$\mathbb{E}_{\boldsymbol{v}_i^0}[\mathbb{E}_{\boldsymbol{x}\sim\mathbb{S}^{d-1}}[f(\boldsymbol{x})\sigma(\langle \boldsymbol{v}_i^0, \boldsymbol{x}\rangle)]^2]$$

$$\overset{(a)}{=} \mathbb{E}_{\boldsymbol{v}_i^0}[\mathbb{E}_{\boldsymbol{x}, \boldsymbol{x}' \sim \mathbb{S}^{d-1}}[f(\boldsymbol{x})f(\boldsymbol{x}')\sigma(\langle \boldsymbol{v}_i^0, \boldsymbol{x} \rangle)\sigma(\langle \boldsymbol{v}_i^0, \boldsymbol{x}' \rangle)]]$$

$$\overset{(b)}{=} \mathbb{E}_{g \sim G_{\mathrm{rot}}, \boldsymbol{v}_i^0}[\mathbb{E}_{\boldsymbol{x}, \boldsymbol{x}' \sim \mathbb{S}^{d-1}}[f(\boldsymbol{x})f(\boldsymbol{x}')\sigma(\langle g(\boldsymbol{v}_i^0), \boldsymbol{x} \rangle)\sigma(\langle g(\boldsymbol{v}_i^0), \boldsymbol{x}' \rangle)]]$$

$$= \mathbb{E}_{g \sim G_{\mathrm{rot}}, \boldsymbol{v}_i^0}[\mathbb{E}_{\boldsymbol{x}, \boldsymbol{x}' \sim \mathbb{S}^{d-1}}[f(\boldsymbol{x})f(\boldsymbol{x}')\sigma(\langle \boldsymbol{v}_i^0, g^{-1}(\boldsymbol{x}) \rangle)\sigma(\langle \boldsymbol{v}_i^0, g^{-1}(\boldsymbol{x}') \rangle)]]$$

$$= \mathbb{E}_{\boldsymbol{v}_i^0}[\mathbb{E}_{g \sim G_{\mathrm{rot}}}[\langle f, \sigma(\langle \boldsymbol{v}_i^0, g^{-1}(\cdot) \rangle) \rangle_{L^2(\mathbb{S}^{d-1})} \langle f, \sigma(\langle \boldsymbol{v}_i^0, g^{-1}(\cdot) \rangle) \rangle_{L^2(\mathbb{S}^{d-1})}]]$$

$$\overset{(c)}{=} \mathbb{E}_{\boldsymbol{v}_i^0}[\sum_{r=0}^{\infty} \frac{1}{\dim(\mathcal{V}_{d,r})} \|\Pi_{\mathcal{V}_{d,r}} f\|_{L^2(\mathbb{S}^{d-1})}^2 \|\Pi_{\mathcal{V}_{d,r}} \sigma(\langle \boldsymbol{v}_i^0, \cdot \rangle)\|_{L^2(\mathbb{S}^{d-1})}^2]$$

$$\geq \Omega(d^{-2l} \cdot \|\Pi_{\mathcal{V}_{d,l}} f\|^2)$$

So combined we have

$$\mathbb{E}_{\boldsymbol{v}_i^0}[\mathbb{E}_{\boldsymbol{x} \sim \mathbb{S}^{d-1}}[f(\boldsymbol{x})\sigma(\langle \boldsymbol{w}_i^0, \boldsymbol{x} \rangle) \mid E_i]^2] \geq \Omega(d^{-C-2l} \cdot \|\Pi_{\mathcal{V}_{d,l}} f\|^2) - O(d^{-2C}).$$

Taking $C = 2l + 2c$, we get

$$\mathbb{E}_{\boldsymbol{v}_i^0}[\mathbb{E}_{\boldsymbol{x} \sim \mathbb{S}^{d-1}}[f(\boldsymbol{x})\sigma(\langle \boldsymbol{w}_i^0, \boldsymbol{x} \rangle) \mid E_i]^2] \geq d^{-C}.$$

So since $m \geq 2C + 1$, by Hoeffding's inequality we get

$$\mathbb{P}_{\boldsymbol{W}^0}[\|\nabla_{\boldsymbol{a}} \ell_{\mathcal{D}}(\boldsymbol{\theta}^0)\|^2 \leq m d^{-C-1/2}] \ll 1,$$

proving the claim. $\square$

The rest of the proof proceeds similarly to the achievability proof of Theorem 3.4. We can upper-bound the smoothness under the event that $\boldsymbol{W}^0$ lies in a ball of radius 2, which occurs with high probability. And we can upper-bound the magnitude of the noise by Gaussian tail bounds. Therefore, the $1/\mathrm{poly}(d)$ lower bound on the gradient implies that a $\geq 1/\mathrm{poly}(d)$ amount of progress is made towards learning when training $\boldsymbol{a}$ for one step. Finally, we can again take $R = \mathrm{poly}(d)$ since we can ensure that the gradients remain $\mathrm{poly}(d)$-bounded.

$\square$

*Proof of achievability result of Theorem 3.6.* The proof is analogous to the achievability result of Theorem 3.4, but using Lemma B.9 instead of Lemma B.4. $\square$

## C  Extension of MSP necessity result, proof of Theorem 3.9

*Proof of Theorem 3.9.* For any set $T \subseteq [P]$, let $\mathcal{S}^{\not\subseteq T} = \{S \subseteq [P] : \hat{f}_*(S) \neq 0, S \not\subseteq T\}$. Since $h_*$ is not $\ell$-MSP, we can choose $T$ such that $\mathcal{S}^{\not\subseteq T} \neq \emptyset$, and for all $S \in \mathcal{S}^{\not\subseteq T}$, we have $|S \setminus T| \geq \ell + 1$.

Let $G' \subseteq G_{\mathrm{perm}}$ be the subgroup of permutations on indices $[d] \setminus T$. Furthermore, let $\alpha(\boldsymbol{x}) = \sum_{S \subseteq T} \hat{f}_*(S) \chi_S(\boldsymbol{x})$. Notice that $\alpha$ is invariant to the action of $G'$, since each permutation $\sigma \in G'$ keeps the indices $T$ fixed. Furthermore, $f_* - \alpha = \sum_{S \in \mathcal{S}^{\not\subseteq T}} \hat{f}_*(S) \chi_S(\boldsymbol{x})$, so

$$\|f_* - \alpha\|^2 > c_{h_*},$$

where $c_{h_*} = \min_{S \subseteq [P], \hat{h}_*(S) \neq 0} |\hat{h}_*(S)|^2 > 0$ is a constant depending only on $h_*$.

Let us now bound the $G'$-alignment of $f_* - \alpha$. For any $\beta \in L^2(\mathcal{H}_d)$, by Cauchy-Schwarz

$$\mathbb{E}_{\sigma \sim G'}[\langle f_* \circ \sigma - \alpha \circ \sigma, \beta \rangle_{L^2(\mathcal{H}_d)}^2] = \mathbb{E}_{\sigma \sim G'}[\langle \sum_{S \in \mathcal{S}^{\not\subseteq T}} \hat{f}_*(S) \chi_S \circ \sigma, \beta \rangle_{L^2(\mathcal{H}_d)}^2]$$

$$\leq 2^P \sum_{S \in \mathcal{S}^{\not\subseteq T}} \hat{f}_*(S)^2 \mathbb{E}_{\sigma \sim G'}[\langle \chi_S \circ \sigma, \beta \rangle_{L^2(\mathcal{H}_d)}^2]$$

And for any $S \in \mathcal{S}^{\not\subseteq T}$, define $G'(S) = \{\sigma(S) : \sigma \in G'\}$. Then (a) by Parseval's, using the orthogonality of the Fourier coefficients, and (b) since $|S \setminus T| \geq \ell + 1$ by construction for all $S \in \mathcal{S}^{\not\subseteq T}$, we have $|G'(S)| = (d - |T|)!/(d - |T| - |S \setminus T|)! \geq d^{\ell+1}/2^P$, for any $d \geq 2P$,

$$\mathbb{E}_{\sigma \sim G'}[\langle \chi_S \circ \sigma, \beta \rangle^2_{L^2(\mathcal{H}_d)}] = \frac{1}{|G'(S)|} \sum_{S' \in G'(S)} \langle \chi_{S'}, \beta \rangle^2_{L^2(\mathcal{H}_d)} \overset{(a)}{\leq} \frac{1}{|G'(S)|} \|\beta\|^2_{L^2(\mathcal{H}_d)} \overset{(b)}{\leq} \|\beta\|^2_{L^2(\mathcal{H}_d)} 2^P/d^{\ell+1}.$$

We conclude that

$$\mathbb{E}_{\sigma \sim G'}[\langle f_* \circ \sigma - \alpha \circ \sigma, \beta \rangle^2_{L^2(\mathcal{H}_d)}] \leq \|\beta\|^2_{L^2(\mathcal{H}_d)} 2^{2P} \sum_{S \in \mathcal{S}^{\not\subseteq T}} \hat{f}_*(S)^2/d^{\ell+1} \leq \|\beta\|^2_{L^2(\mathcal{H}_d)} C_{h_*}/d^{\ell+1},$$

where $C_{h_*}$ is a constant depending only on $h_*$. So

$$\mathcal{C}(f_* - \alpha; \mathcal{H}_d, G') \leq C_{h_*}/d^{\ell+1}.$$

Set $\epsilon_0 = c_{h_*}/2$. By Theorem 3.1 have

$$\mathbb{P}_{\boldsymbol{\theta}^k}[\ell_{\mathcal{D}}(\boldsymbol{\theta}^k) \leq \epsilon_0] \leq \frac{\eta L}{2\tau} \sqrt{\frac{C_{h_*} k}{d^{\ell+1}}} + \frac{2C_{h_*}}{d^{\ell+1} c_{h_*}} \leq \frac{C_{h_*} \eta L}{\tau} \sqrt{\frac{k}{d^{\ell+1}}} + \frac{2C_{h_*}}{d^{\ell+1}}.$$

$\square$

# D  Hardness results for SGD

In this section, for $\gamma > 0$, we let $\mathcal{D}(f, \mu_{\mathcal{X}}, \gamma) \in \mathcal{P}(\mathcal{X} \times \mathbb{R})$ denote the distribution of $(\boldsymbol{x}, f(\boldsymbol{x}) + \xi)$ where $\boldsymbol{x} \sim \mu_{\mathcal{X}}$ and $\xi \sim \mathcal{N}(0, \gamma^2)$ is independent noise.

## D.1  Warm-up: hardness from permutation equivariance

Before proving Theorem 4.4, we warm up with a simple observation that exploits permutation equivariance.

**Theorem D.1.** *For any $d$, define $f_d : \mathcal{H}_d \to \{+1, -1\}$ to be the parity of the first $\lfloor d/2 \rfloor$ bits, i.e.,*

$$f_d(\boldsymbol{x}) = \prod_{i=1}^{\lfloor d/2 \rfloor} x_i.$$

*Let $\{f_{\mathsf{NN},d}, \mu_{\boldsymbol{\theta},d}\}_{d \in \mathbb{N}}$ be a family of networks and initializations satisfying Assumption 2.4 (FC, i.i.d. initialization). Let $\gamma > 0$, and let $\{n_d\}$ be sample sizes such that $(f_{\mathsf{NN},d}, \mu_{\boldsymbol{\theta}_d})$-SGD training on $n_d$ samples from $\mathcal{D}(f_d, \mathcal{H}_d, \gamma)$ rounded to $\mathrm{poly}(d)$ bits yields parameters $\boldsymbol{\theta}_d$ with*

$$\mathbb{E}_{\boldsymbol{\theta}_d}[\|f_d - f_{\mathsf{NN}}(\cdot; \boldsymbol{\theta}_d)\|^2] \leq 0.01. \tag{12}$$

*Then, under $\gamma$-LPGN hardness, it is not possible to run $(f_{\mathsf{NN},d}, \mu_{\boldsymbol{\theta},d})$-SGD on $n_d$ samples and evaluate $f_{\mathsf{NN}}(\cdot; \boldsymbol{\theta}_d)$ in $\mathrm{poly}(d)$ time.*

This result follows straightforwardly from $G_{perm}$-equivariance of SGD in Proposition 2.5, which states that SGD on FC networks with i.i.d. initialization is blind to the order of the input vectors' coordinates, up to an unknown shared permutation. So if $(f_{\mathsf{NN},d}, \mu_{\boldsymbol{\theta},d})$-SGD can learn $f(\boldsymbol{x}) = \prod_{i=1}^{\lfloor d/2 \rfloor} x_i$ from $\gamma$-noisy samples, it can learn $f(\boldsymbol{x}) = \chi_S(\boldsymbol{x})$ from noisy samples, for any $S \subseteq [d]$ with $|S| = \lfloor d/2 \rfloor$. But that is a $\gamma$-LPGN-hard problem.

*Proof of Theorem D.1.* Formally, consider the algorithm $\mathcal{A}_d^{\mathrm{SGD}}$ that runs $(f_{\mathsf{NN},d}, \mu_{\boldsymbol{\theta},d})$-SGD on $n_d$ samples $(\boldsymbol{x}_i, y_i)_{i \in [n_d]}$ and outputs $\mathcal{A}_d^{\mathrm{SGD}}((\boldsymbol{x}_i, y_i)_{i \in [n_d]}) : \mathcal{H}_d \to \mathbb{R}$ given by $[\mathcal{A}_d^{\mathrm{SGD}}((\boldsymbol{x}_i, y_i)_{i \in [n_d]})](\cdot) = f_{\mathsf{NN},d}(\cdot; \boldsymbol{\theta}_d)$. Let us prove that $\mathcal{A}_d^{\mathrm{SGD}}$ achieves probability of error $\leq 0.01$ on the $\gamma$-LPGN problem.

Let $(S, \boldsymbol{q}, (x_i, y_i)_{i \in [n_d]})$ be a $(d, n_d, \gamma)$-LPGN instance, where $S \subseteq [d]$ is unknown, $\boldsymbol{q} \sim \mathcal{H}_d$ is random, and $(x_i, y_i)_{i \in [n_d]}$ is random with $\boldsymbol{x}_i \overset{i.i.d}{\sim} \mathcal{H}_d$ and $y_i = \prod_{i \in S} x_i + \xi_i$, for $\xi_i \sim \mathcal{N}(0, \gamma^2)$. Let $\sigma \in S_d$ be a permutation that sends $\{1, \ldots, \lfloor d/2 \rfloor\}$ to $S$. Then by (a) permutation equivariance

of $\mathcal{A}_d^{\mathrm{SGD}}$ from Proposition 2.5; (b) letting $\tilde{\boldsymbol{q}} = \sigma(\boldsymbol{q})$ and using that $\chi_S(\boldsymbol{q}) = \prod_{i=1}^{\lfloor d/2 \rfloor} \tilde{q}_i$; (c) letting $(\tilde{\boldsymbol{x}}_i, \tilde{y}_i)_{i \in [n_d]}$ be such that $\tilde{\boldsymbol{x}}_i = \sigma(\boldsymbol{x}_i)$ and $\tilde{y}_i = y_i$; and finally (d) using that $\tilde{\boldsymbol{q}} \sim \mathcal{H}_d$, and $(\tilde{\boldsymbol{x}}_i, \tilde{y}_i)_{i \in [n_d]} \overset{i.i.d.}{\sim} \mathcal{D}(f_*^{(d)}, \mu^{(d)}, \gamma)$ and a Markov bound on (12), ,

$$
\begin{aligned}
\mathbb{P}\{\mathrm{sgn}([\mathcal{A}_d^{\mathrm{SGD}}((\boldsymbol{x}_i, y_i)_{i \in [n_d]})](\boldsymbol{q})) = \chi_S(\boldsymbol{q})\} &\overset{(a)}{=} \mathbb{P}\{\mathrm{sgn}([\mathcal{A}_d^{\mathrm{SGD}}((\sigma(\boldsymbol{x}_i), y_i)_{i \in [n_d]})](\sigma(\boldsymbol{q}))) = \chi_S(\boldsymbol{q})\} \\
&\overset{(b)}{=} \mathbb{P}\{\mathrm{sgn}([\mathcal{A}_d^{\mathrm{SGD}}((\sigma(\boldsymbol{x}_i), y_i)_{i \in [n_d]})](\tilde{\boldsymbol{q}})) = \prod_{i=1}^{\lfloor d/2 \rfloor} \tilde{q}_i\} \\
&\overset{(c)}{=} \mathbb{P}\{\mathrm{sgn}([\mathcal{A}_d^{\mathrm{SGD}}((\tilde{\boldsymbol{x}}_i, \tilde{y}_i)_{i \in [n_d]})](\tilde{\boldsymbol{q}})) = \prod_{i=1}^{\lfloor d/2 \rfloor} \tilde{q}_i\} \\
&\overset{(d)}{\geq} 1 - 0.01 \geq 0.99.
\end{aligned}
$$

Therefore $\mathrm{sgn}(\mathcal{A}_d^{\mathrm{SGD}})$ solves the $(d, n_d, \gamma)$-LPGN problem. By the $\gamma$-LPGN-hardness assumption this algorithm cannot run in $\mathrm{poly}(d)$ time. So if the $(f_{\mathrm{NN},d}, \mu_{\boldsymbol{\theta},d})$-SGD training can be run in $\mathrm{poly}(d)$ time and $f_{\mathrm{NN}}(\cdot; \boldsymbol{\theta}_d)$ can be evaluated in $\mathrm{poly}(d)$ time, contradicting the $\gamma$-LPGN-hardness assumption. $\qquad\square$

## D.2 Hardness from sign-flip equivariance, proof of Theorem 4.4

Our second result is not as obvious, and exploits the sign-flip equivariance of training on FC architectures with sign-flip-symmetric initialization (such as Gaussian initialization). Let $f_{\mathrm{mod8},d} : \mathcal{H}_d \to \{0, \ldots, 7\}$ denote the function given by $f_{\mathrm{mod8},d}(\boldsymbol{x}) \equiv \sum_i x_i \pmod 8$. For brevity, we drop the subscript with $d$ and write $f_{\mathrm{mod8}}$, since $d$ is clear from context.

Restated, our main result is:

**Theorem D.2** (Theorem 4.4 restated). *Let $\{f_{\mathrm{NN},d}, \mu_{\boldsymbol{\theta},d}\}_{d \in \mathbb{N}}$ be a family of networks and initializations satisfying Assumption 2.4 (fully-connected) with i.i.d. symmetric initialization. Let $\gamma > 0$, and let $\{n_d\}$ be sample sizes such that $(f_{\mathrm{NN},d}, \mu_{\boldsymbol{\theta},d})$-SGD training on $n_d$ samples from $\mathcal{D}(f_{\mathrm{mod8}}, \mathcal{H}_d, \gamma)$ rounded to $\mathrm{poly}(d)$ bits yields parameters $\boldsymbol{\theta}_d$ with*

$$
\mathbb{E}_{\boldsymbol{\theta}_d}[\|f_{\mathrm{mod8}} - f_{\mathrm{NN}}(\cdot; \boldsymbol{\theta}_d)\|^2] \leq 0.0001. \tag{13}
$$

*Then, under $(\gamma/2)$-LPGN hardness, it is not possible to run $(f_{\mathrm{NN},d}, \mu_{\boldsymbol{\theta},d})$-SGD on $n_d$ samples and evaluate $f_{\mathrm{NN}}(\cdot; \boldsymbol{\theta}_d)$ in $\mathrm{poly}(d)$ time.*

The proof of Theorem 4.4 can be also proved via a reduction to LPGN. However, this reduction is much less straightforward.

### D.2.1 The secretly-flipped sum-mod-8 (SFSM8) problem

We prove the hardness of learning sum-mod-8 with SGD by using only the sign-flip equivariance, and no other properties of the SGD algorithm. The idea is that any sign-flip-equivariant algorithm that can learn $f_{\mathrm{mod8}}$, must also be capable of solving a much more difficult problem. First, define the problem of outputting sum-mod-8.

**Problem D.3.** *The $(d, n, \gamma)$-SM8 (sum-mod-8) problem is parametrized by $\gamma > 0$ and integers $d, n > 0$. It is as follows:*

- *Input: query vector $\boldsymbol{q} \sim \mathcal{H}_d$, samples $(\boldsymbol{x}_i, y_i)_{i \in [n]} \overset{i.i.d.}{\sim} \mathcal{D}(f_{\mathrm{mod8}}, \mathcal{H}_d, \gamma)$.*

- *Task: return $f_{\mathrm{mod8}}(\boldsymbol{q}) \in \{0, \ldots, 7\}$.*

Notice that SM8 is a trivial problem: an algorithm that was not sign-flip equivariant could ignore the samples $(\boldsymbol{x}_i, y_i)_{i \in [n]}$, and simply return $f_{\mathrm{mod8}}(\boldsymbol{q})$. However, sign-flip equivariance makes solving SM8 much more difficult: we prove that any sign-flip equivariant algorithm that can solve SM8, can also solve the problem SFSM8, defined below.

**Problem D.4.** *The $(d, n, \gamma)$-secretly-flipped sum-mod-8 (SFSM8) problem is parametrized by $\gamma > 0$, and integers $d, n > 0$. It is as follows:*

---

**Algorithm** REDUCE-LPGN-TO-SFSM8

*Inputs*: query $\boldsymbol{q} \in \mathcal{H}_d$, samples $(\boldsymbol{x}_i, y_i)_{i \in [n]}$ from an instance of $(d, n, \gamma_{\mathcal{A}}/2)$-LPGN, oracle $\mathcal{A}$ for $(d, n, \gamma_{\mathcal{A}})$-SFSM8.

1. For each $i \in [n]$, compute $t_i \in \{0, \ldots, 7\}$ by

$$t_i \equiv 2|S| - 2 - \sum_{j \in [d]} x_{i,j} \pmod{8},$$

   and let

$$\tilde{y}_i = \begin{cases} t_i + 2y_i, & t_i \in \{2, 3, 4, 5\} \\ t_i + 4 - 2y_i, & t_i \in \{0, 1\} \\ t_i - 4 - 2y_i, & t_i \in \{6, 7\} \end{cases}.$$

2. Let ans $= \mathcal{A}(\boldsymbol{q}, (\boldsymbol{x}_i, \tilde{\boldsymbol{y}}_i)_{i \in [n]}) - 2|S| + 2 + \sum_{j \in [d]} x_j$.

3. If ans $\equiv 2 \pmod{8}$, return 1. Else if ans $\equiv -2 \pmod{8}$, return $-1$. Else, return "error".

---

Figure 1: Reduction from LPGN to SFSM8 (Lemma D.6). Here, $\gamma_{\mathcal{B}} = \gamma_{\mathcal{A}}/2$. Note that $|S| = \lfloor d/2 \rfloor$ is known, so the reduction is efficient.

- *Unknown: sign-flip vector $\boldsymbol{s} \in \mathcal{H}_d$.*

- *Input: query vector $\boldsymbol{q} \sim \mathcal{H}_d$, modified samples $(\boldsymbol{x}_i \odot \boldsymbol{s}, y_i)_{i \in [n]}$, where $(\boldsymbol{x}_i, y_i)_{i \in [n]} \overset{i.i.d.}{\sim} \mathcal{D}(f_{\mathrm{mod}8}, \mathcal{H}_d, \gamma)$.*

- *Task: return $f_{\mathrm{mod}8}(\boldsymbol{q} \odot \boldsymbol{s}) \in \{0, \ldots, 7\}$.*

**Lemma D.5.** *Let $\mathcal{A}$ be a $G_{\mathrm{sign}}$-equivariant algorithm in the sense of Definition 2.2. Then $\mathcal{A}$'s error probability on SM8 equals $\mathcal{A}$'s error probability on SFSM8.*

*Proof.* By sign-flip equivariance, for any $\boldsymbol{s}, \boldsymbol{q}$, and any samples $(\boldsymbol{x}_i, y_i)_{i \in [n]}$,

$$[\mathcal{A}((\boldsymbol{x}_i \odot \boldsymbol{s}, y_i)_{i \in [n]})](\boldsymbol{q} \odot \boldsymbol{s}) \overset{d}{=} [\mathcal{A}((\boldsymbol{x}_i, y_i)_{i \in [n]})](\boldsymbol{q}).$$

So drawing $(\boldsymbol{x}_i, y_i)_{i \in [n]} \overset{i.i.d.}{\sim} \mathcal{D}(f_{\mathrm{mod}8}, \mathcal{H}_d, \gamma)$,

$$\mathbb{P}[[\mathcal{A}((\boldsymbol{x}_i, y_i)_{i \in [n]})](\boldsymbol{q}) = f_{\mathrm{mod}8}(\boldsymbol{q})] = \mathbb{P}[[\mathcal{A}((\boldsymbol{x}_i \odot \boldsymbol{s}, y_i)_{i \in [n]})](\boldsymbol{q} \odot \boldsymbol{s}) = f_{\mathrm{mod}8}(\boldsymbol{q} \odot \boldsymbol{s})].$$

$\square$

### D.2.2 Reduction from LPGN to SFSM8

Now we show that there is no algorithm that solves SFSM8 in polynomial time and samples, under the cryptographic assumption that learning parities with Gaussian noise (LPGN) is hard.

**Lemma D.6** (Reduction from LPGN to SFSM8)**.** *Given access to an oracle $\mathcal{A}$ for $(d, n, \gamma_{\mathcal{A}})$-SFSM8 that is correct with probability $> 9/10$, one can construct an algorithm $\mathcal{B}$ for $(d, n, \gamma_{\mathcal{A}}/2)$-LPGN that is correct with probability $> 9/10$ and runs in one call to $\mathcal{A}$ and $\mathrm{poly}(n, d)$ additional time. Furthermore, $\gamma_{\mathcal{B}} = \gamma_{\mathcal{A}}/2$.*

*Proof.* The pseudocode for the reduction is given in Figure 1. We prove correctness. Let $S \subseteq [d]$ be the unknown subset for the LPGN problem. Define $\boldsymbol{s} \in \mathcal{H}_d$ where $s_j = 1$ for all $j \in S$, and $s_j = -1$ for all $j \notin S$. Then, for any $\boldsymbol{x} \in \mathcal{H}_d$,

$$\sum_{j \in [d]} x_j s_j = \sum_{j \in [d]} x_j (1 + s_j) - \sum_{j \in [d]} x_j$$

$$= 2|\{j \in S : x_j = 1\}| - 2|\{j \in S : x_j = -1\}| - \sum_{j \in [d]} x_j$$

$$= 2(|S| - 2|\{j \in S : x_j = -1\}|) - \sum_{j \in [d]} x_j$$

$$\equiv 2 \prod_{j \in S} x_j + 2|S| - 2 - \sum_{j \in [d]} x_j \pmod{8}$$

where in the last line we use $|\{j \in S : x_j = -1\}| \equiv (\prod_{j \in S} x_j - 1)/2 \pmod{2}$.

This guarantees that after Step 1 of the algorithm, the samples $(\boldsymbol{x}_i, \tilde{y}_i)_{i \in n}$ are distributed as if they were drawn from an instance of $(d, n, \gamma_{\mathcal{A}})$-SFSM8 with secret sign-flip vector $\boldsymbol{s}$. I.e., $\boldsymbol{x}_i \overset{i.i.d.}{\sim} \mathcal{H}_d$, and $\tilde{y}_i = f_{\text{mod}8}(\boldsymbol{x}_i) + \xi_i$ for $\xi_i \overset{i.i.d}{\sim} \mathcal{N}(0, \gamma_{\mathcal{A}}^2)$.

By the correctness guarantee of $\mathcal{A}$, we know that with probability at least 9/10, the output $\mathcal{A}(\boldsymbol{q}, (\boldsymbol{x}_i, \tilde{y}_i)_{i \in [n]}) \in \{0, \ldots, 7\}$ satisfies $\mathcal{A}(\boldsymbol{q}, (\boldsymbol{x}_i, \tilde{y}_i)_{i \in [n]}) \equiv \sum_i q_i s_i \pmod{8}$. So by the above calculations, $\text{ans} \equiv 2 \prod_{j \in S} q_S$, proving correctness. $\qquad\square$

### D.2.3  Proof of Theorem 4.4

*Proof of Theorem 4.4.* Suppose by contradiction that $(f_{\text{NN},d}, \mu_{\boldsymbol{\theta},d})$-SGD can be run in $\text{poly}(d)$ time and the learned function can be evaluated in $\text{poly}(d)$ time. Let $\mathcal{A}_d = \text{round}(f_{\text{NN},d}(\cdot; \boldsymbol{\theta}_d))$ be the algorithm that runs $(f_{\text{NN},d}, \mu_{\boldsymbol{\theta},d})$-SGD on $n_d$ samples $(\boldsymbol{x}_i, y_i)_{i \in [n_d]}$ and returns the learned function, rounded to the nearest integer. By a Markov bound on (13), $\mathbb{P}\{[\mathcal{A}_d((\boldsymbol{x}_i, y_i)_{i \in [n_d]})](\boldsymbol{q}) = f_{\text{mod}8}(\boldsymbol{q})\} \geq 1 - 0.004 \geq 0.99$. Furthermore, $\mathcal{A}_d$ is $G_{\text{sign}}$-equivariant by Proposition 2.5. So by Lemma D.5, $\mathcal{A}_d$ gives a $\text{poly}(d)$-size circuit for the $(d, n_d, \gamma)$-SFSM8 problem. By the LPGN-to-SFSM8 reduction in Lemma D.6, we see that this contradicts the $(\gamma/2)$-LPGN-hardness assumption. $\qquad\square$

### D.3  On the cryptographic assumption that LPGN is hard

In our SGD hardness results, we assume hardness of the LPGN (Learning Parities with Gaussian Noise) problem from Definition 4.1. The standard hardness assumption in the literature is hardness of LPN (Learning Parities with Noise). The differences are that in LPN: (1) the noise is classification noise instead of Gaussian noise; (2) there is no promise that the unknown subset has size $|S| = \lfloor d/2 \rfloor$. In this appendix, we show that our LPGN assumption can be derived from the LPN assumption, with slightly different parameters.

**Definition D.7.** *The learning parities with noise, $(d, n, \rho)$-LPN, problem is parametrized by $d, n \in \mathbb{Z}_{>0}$ and $\rho \in (0, 1]$. An instance $(S, \boldsymbol{q}, (\boldsymbol{x}_i, y_i)_{i \in [n]})$ consists of (i) an unknown subset $S \subseteq [d]$, and (ii) a known query vector $\boldsymbol{q} \sim \mathcal{H}_d$, and i.i.d. samples $(\boldsymbol{x}_i, y_i)_{i \in [n]}$ such that $\boldsymbol{x}_i \sim \mathcal{H}_d$ and $y_i = \chi_S(\boldsymbol{x}_i)\zeta_i$, where*

$$\zeta_i \sim \begin{cases} 1, & \text{w.p. } (1 + \rho)/2 \\ -1, & \text{w.p. } (1 - \rho)/2 \end{cases}.$$

*The task is to return $\chi_S(\boldsymbol{q}) \in \{+1, -1\}$.*

In order to reduce from LPN to LPGN, let us define promise-LPN:

**Definition D.8.** *The promise-LPN problem is the LPN problem with the promise that $|S| = \lfloor d/2 \rfloor$.*

We prove the following theorem, where $\rho(d)$-LPN is the LPN problem where can take any number of samples $n$, and the value of $\rho$ depends on $d$.

**Theorem D.9.** *Suppose that for all constants $C > 0$, and any $\rho(d) \leq 1 - \exp(-C\sqrt{\log(d)})$, there are no $\text{poly}(d)$-time algorithms for $\rho(d)$-LPN. Then, for any constant $\gamma > 0$, there are no $\text{poly}(d)$-time algorithms for $\gamma$-LPGN.*

*Proof.* This follows by combining Lemmas D.10 and D.11, as outlined in the diagram below:

$$\text{LPN} \overset{\text{Lemma } D.10}{\Longrightarrow} \text{promise-LPN} \overset{\text{Lemma } D.11}{\Longrightarrow} \text{LPGN} .$$

$\qquad\square$

---

**Algorithm** LPN-TO-PROMISE-LPN

*Inputs*: query $\boldsymbol{q} \in \mathcal{H}_d$, samples $(\boldsymbol{x}_i, y_i)_{i \in [n_{\mathcal{B}}]}$, from an instance of $(d, \rho, n_{\mathcal{B}})$-LPN, oracle $\mathcal{A}$ for $(2d, \rho, n_{\mathcal{A}})$-promise-LPN.

1. Let $T = 10000 \log(d)/\rho^2$. Relabel the samples, splitting them into groups as

$$(\boldsymbol{x}_i, y_i)_{i \in [n_{\mathcal{B}}]} = \left( \bigsqcup_{r \in \{0,\ldots,d\}, t \in [T]} \{(\boldsymbol{x}_i^{(r,t)}, y_i^{(r,t)})\}_{i \in [n_{\mathcal{A}}+1]} \right) \sqcup (\boldsymbol{x}_i^{(*)}, y_i^{(*)})_{i \in [n_{\mathcal{A}}]}$$

2. For each $r \in \{0, \ldots, d\}$ and $t \in [T]$, let

$$\mathrm{ans}_{r,t} = \mathcal{A}([\boldsymbol{x}_{n_{\mathcal{A}}+1}^{(r,t)}, \boldsymbol{z}_{n_{\mathcal{A}}+1}^{(r,t)}], ([\boldsymbol{x}_i^{(r,t)}, \boldsymbol{z}_i^{(r,t)}], y_i^{(r,t)} \cdot \prod_{j=1}^{r} z_{ij}^{(r,t)})_{i \in [n_{\mathcal{A}}]}) \cdot \prod_{j=1}^{r} z_{n_{\mathcal{A}}+1,j}^{(r,t)},$$

   where $\boldsymbol{z}_i^{(r,t)} \sim \mathcal{H}_d$ is random padding and $[\boldsymbol{x}_i^{(r,t)}, \boldsymbol{z}_i^{(r,t)}] \in \mathcal{H}_{2d}$ is the concatenation.

3. For each $r \in \{0, \ldots, d\}$, let $\hat{p}_r = |\{t \in [T] : \mathrm{ans}_{r,t} = y_{n_{\mathcal{A}}+1}^{(r,t)}\}|/T$

4. Let $(\boldsymbol{z}_i^{(*)})_{i \in [n_{\mathcal{A}}]} \sim \mathcal{H}_d$. Let $\hat{r} = \arg\max_r \hat{p}_r$. Return

$$\mathcal{A}([\boldsymbol{q}, \boldsymbol{z}_{n_{\mathcal{A}}+1}^{(*)}], ([\boldsymbol{x}_i^{(*)}, \boldsymbol{z}_i^{(*)}], y_i^{(*)} \cdot \prod_{j=1}^{\hat{r}} z_{ij}^{(*)})_{i \in [n_{\mathcal{A}}]}) \cdot \prod_{j=1}^{\hat{r}} z_{n_{\mathcal{A}}+1,j}^{(*)}.$$

---

Figure 2: Reduction from LPN to promise-LPN (Lemma D.10). Here $n_{\mathcal{B}} = dT(1 + n_{\mathcal{A}}) + n_{\mathcal{A}}$.

### D.3.1 Reduction from LPN to promise-LPN

**Lemma D.10** (LPN to promise-LPN). *Given access to an oracle $\mathcal{A}$ for $(2d, n_{\mathcal{A}}, \rho)$-promise-LPGN that is correct with probability $> 19/20$, one can construct an algorithm $\mathcal{B}$ for $(d, n_{\mathcal{B}}, \rho)$-LPGN that is correct with probability $> 9/10$ and runs in $O(d \log(d)/\rho^2)$ calls to $\mathcal{A}$ and $O(n_{\mathcal{A}} d \log(d)/\rho^2)$ additional time. Furthermore, $n_{\mathcal{B}} \leq d \log(d) n_{\mathcal{A}}/\rho^2$.*

*Proof.* The pseudocode for the reduction is given in Figure 2. We prove correctness. Let $S \subseteq [d]$ be the unknown subset for the LPN problem. For any $r \in \{0, \ldots, d\}$, define the success probability of running $\mathcal{A}$ with $r$ as

$$s_r := \mathbb{P}[\mathrm{ans}_{r,1} = \chi_S(\boldsymbol{x}_{n_{\mathcal{A}}+1}^{(r,1)})].$$

Notice that if we take $r^* = d - |S|$, then the samples that we feed into $\mathcal{A}$ are those of the LPN problem with unknown subset $S' = S \cup \{d+1, \ldots, d+r\}$, which has size $|S'| = d = \lfloor 2d/2 \rfloor$. Therefore, the guarantee for $\mathcal{A}$ implies that if we take $r^* = d - |S|$, then

$$s_{r^*} > 19/20.$$

However, we are not given $r^*$ in the input of the LPGN problem, which is the main difficulty. Instead, in the first part of the algorithm we estimate the success probability for each $r \in \{0, \ldots, d\}$ using fresh samples, and at the end run $\mathcal{A}$ with the $r^*$ that we estimate gives the best success probability.

To prove correctness, let $E$ be the event that for all $r \in \{0, \ldots, d\}$ we have

$$|\hat{p}_r - \mathbb{P}[\mathrm{ans}_{r,1} = y_{n_{\mathcal{A}}+1}^{(r,1)}]| \leq 1/(50\rho).$$

By Hoeffding's inequality, $E$ holds with probability at least $\mathbb{P}[E] \geq 1 - 2\exp(-8\log(d)) \geq 1/100$. Now notice that $y_{n_{\mathcal{A}}+1}^{(r,1)}$ is just $\chi_S(\boldsymbol{x}_{n_{\mathcal{A}}+1}^{(r,1)})$ with classification noise so under the event $E$

$$|(\hat{p}_r + \rho/2 - 1/(2\rho))/\rho - s_r| \leq 1/(50).$$

Therefore, since $s_{r^*} \geq 19/20$, under event $E$ we must have $s_{\hat{r}} \geq 19/20 - 1/25 = 9/10 + 1/100$. Finally, since the probability of success of the algorithm is $s_{\hat{r}}$, and $\mathbb{P}[E] \geq 1/100$, we have that the algorithm's success probability is at least $9/10$. $\qquad\square$

---

**Algorithm** PROMISE-LPN-TO-LPGN

*Inputs*: query $\boldsymbol{q} \in \mathcal{H}_d$, samples $(\boldsymbol{x}_i, y_i)_{i \in [n]}$, from an instance of $(d, \rho, n)$-promise-LPN, oracle $\mathcal{A}$ for $(d, \gamma, n)$-LPGN.

    1. For each $i \in [n]$, let $\tilde{y}_i = \mathrm{RK}(y_i)$, where RK is the rejection kernel

$$\mathrm{RK} = \mathrm{RK}((1 + \rho)/2 \to \mathcal{N}(1, \gamma^2), (1 - \rho)/2 \to \mathcal{N}(-1, \gamma^2))$$

       from Lemma 5.1 of [BBH18].

    2. Return $\mathcal{A}(\boldsymbol{q}, (\boldsymbol{x}_i, \tilde{y}_i)_{i \in n})$

---

Figure 3: Reduction from promise-LPN to LPGN (Lemma D.11).

### D.3.2    Reduction from promise-LPN to LPGN

Here, we show that the LPGN hardness assumption follows from the standard LPN hardness assumption with classification noise. The technique is rejection kernels.

**Lemma D.11.** *Given access to an oracle $\mathcal{A}$ for $(2d, n, \gamma)$-LPGN that is correct with probability $> 49/50$, one can construct an algorithm $\mathcal{B}$ for $(d, n, \rho)$-promise-LPN that is correct with probability $> 19/20$ and runs in one call to $\mathcal{A}$ and $\mathrm{poly}(n, d)$ additional time. Furthermore, we can take $\rho \le 1 - \exp(-C\sqrt{\log(n)} \max(\gamma^2, 1/\gamma^2))$, for some universal constant $C > 0$.*

*Proof.* The pseudocode of the reduction is given in Figure 3. We use the rejection kernel technique developed in [BBH18] to convert the classification noise of LPN to the additive Gaussian noise of LPGN. Let $\rho = 1 - 2\delta$, where $\delta < 0.1$. Notice that if we define $p = (1 + \rho)/2 = 1 - \delta$ and $q = (1 - \rho)/2 = \delta$, Lemma 5.1 of [BBH18] provides a randomized map RK that runs in time and maps a $\mathrm{Rad}(p)$ random variable to $\mathcal{N}(1, \gamma^2)$ and a $\mathrm{Rad}(q)$ random variable to $\mathcal{N}(-1, \gamma^2)$. For some parameter $N > 0$, this map runs in $O(N)$ time and has the guarantee that

$$d_{\mathrm{TV}}(\mathrm{RK}(\mathrm{Rad}(p)), \mathcal{N}(1, \gamma^2)) \le \Delta \quad \text{and} \quad d_{\mathrm{TV}}(\mathrm{RK}(\mathrm{Rad}(q)), \mathcal{N}(-1, \gamma^2)) \le \Delta,$$

where

$$\Delta = \mathbb{P}_{X \sim \mathcal{N}(1, \gamma^2)}[X \notin \mathcal{S}] + (\mathbb{P}_{X \sim \mathcal{N}(-1, \gamma^2)}[X \notin \mathcal{S}] + \frac{\delta}{1 - \delta})^N,$$

and

$$\mathcal{S} = \{x : \frac{\delta}{1 - \delta} \le \frac{\exp(-(x-1)^2/(2\gamma^2))}{\exp(-(x+1)^2/(2\gamma^2))} \le \frac{1 - \delta}{\delta}\} = \{x : \frac{\delta}{1 - \delta} \le \exp(\frac{2x}{\gamma^2}) \le \frac{1 - \delta}{\delta}\}.$$

By standard Gaussian tail bounds,

$$\mathbb{P}_{X \sim \mathcal{N}(1, \gamma^2)}[X \notin \mathcal{S}] = \mathbb{P}_{X \sim \mathcal{N}(-1, \gamma^2)}[X \notin \mathcal{S}] \le \exp(-(\gamma^2 \log((1-\delta)/\delta)/2 - 1)^2/2\gamma^2).$$

So for any $\epsilon > 0$, as long as $\delta < \min(\exp(-10\sqrt{\log(1/\epsilon)}/\gamma^2), \exp(-10\sqrt{\log(1/\epsilon)}\gamma^2))$, we have

$$\mathbb{P}_{X \sim \mathcal{N}(1, \gamma^2)}[X \notin \mathcal{S}] \le \epsilon.$$

So letting $\epsilon \le 1/(2000n^2)$, and $N \ge \log(2000n^2)$, we have $\Delta \le n/100$.

Thus, $(\boldsymbol{q}, (\boldsymbol{x}_i, \tilde{y}_i)_{i \in [n]})$ is $\le 1/100$ total-variation distance from being drawn from LPGN with unknown set $S$. This means that the algorithm has success probability $49/50 - 1/100 \ge 19/20$. $\quad \square$

## E    On the equivariance of SGD and GD

Recall from Definition 1.1 that an algorithm $\mathcal{A}$ that takes in a distribution $\mathcal{D} \in \mathcal{P}(\mathcal{X} \times \mathbb{R})$ and outputs a function $\mathcal{A}(\mathcal{D}) : \mathcal{X} \to \mathbb{R}$ is said to be $G$-equivariant if

$$\mathcal{A}(\mathcal{D}) \overset{d}{=} \mathcal{A}(g(\mathcal{D})) \circ g,$$

for any $g \in G$. Here we view the group element as a function $g : \mathcal{X} \to \mathcal{X}$, since it acts on the space of inputs $\mathcal{X}$. We also define $g(\mathcal{D})$ to be the distribution of $(g(\boldsymbol{x}), y)$, where $(\boldsymbol{x}, y) \sim \mathcal{D}$. In

Definitions 2.1 and 2.2, we define the equivariance of GD and SGD, respectively. In Proposition 2.5, we claim that GD and SGD are $G_{perm}$-equivariant in the case of training FC networks with i.i.d. initialization. Furthermore, these algorithms are $G_{\text{sign,perm}}$-, and $G_{\text{rot}}$-equivariant when the initialization is symmetric and Gaussian, respectively. The equivariances of SGD in Proposition 2.5 are proved by [Ng04, LZA21]. They also show extension to other algorithms beyond SGD:

**Remark E.1.** *[LZA21] note that other popular optimizers on FC networks with i.i.d. symmetric initialization, such as AdaGrad and Adam, also yield $G_{\text{sign,perm}}$-equivariance, as well as SGD with a mini-batch, or gradient descent on the loss of the empirical distribution $\ell_{\hat{\mathcal{D}}}$, where $\hat{\mathcal{D}} = \frac{1}{n}\sum_{i=1}^{n}\delta_{(y_i, \boldsymbol{x}_i)}$, for $(\boldsymbol{x}_i, y_i)_{i\in[n]} \overset{i.i.d.}{\sim} \mathcal{D}$. Therefore our SGD hardness result in Theorem 4.4 applies to training with the above algorithms.*

We limit ourselves in this appendix to prove the equivariances claimed by Proposition 2.5 for GD. We provide the proofs only for completeness, since they are quite similar to the proofs of equivariance of SGD.

In the remainder of this section, suppose that we have an architecture $f_{\text{NN}}(\cdot; \boldsymbol{\theta})$ and an initialization $\mu_{\boldsymbol{\theta}}$ that satisfy Assumption 2.4, so that we can write the parameters $\boldsymbol{\theta} = (\boldsymbol{W}, \boldsymbol{\psi})$. Furthermore, let $\mathcal{D} \subseteq \mathcal{P}(\mathbb{R}^d \times \mathbb{R})$ be a data distribution.

**Definition E.2.** *For any invertible linear transformation $\boldsymbol{M} \in GL(d, \mathbb{R})$, let $\boldsymbol{M}$ act on $\boldsymbol{\theta}$ as $\boldsymbol{M}\square\boldsymbol{\theta} = (\boldsymbol{W}\boldsymbol{M}, \boldsymbol{\psi})$. Let $\boldsymbol{M}\square\mathcal{D}$ be the distribution of $(\boldsymbol{M}\boldsymbol{x}, y)$ for $(\boldsymbol{x}, y) \sim \mathcal{D}$.*

**Lemma E.3.** *Suppose that for some orthogonal transformation $\boldsymbol{M} \in O(d)$, if we draw $\boldsymbol{\theta}^0 \sim \mu_{\boldsymbol{\theta}}$, then $\boldsymbol{M}\square\boldsymbol{\theta}^0 \overset{d}{=} \boldsymbol{\theta}^0$. Let $\boldsymbol{\theta}^k$ be the weights from $(f_{\text{NN}}, \mu_{\boldsymbol{\theta}})$-GD on distribution $\mathcal{D}$, for any number of steps $k$, with any learning rate $\eta > 0$, and with any noise level $\tau > 0$. Similarly, let $\tilde{\boldsymbol{\theta}}^k$ be the weights from running $(f_{\text{NN}}, \mu_{\boldsymbol{\theta}})$-GD on distribution $\tilde{\mathcal{D}} = \boldsymbol{M}\square\mathcal{D}$.*

*Then, for any $\boldsymbol{x} \in \mathbb{R}^d$,*

$$f_{\text{NN}}(\boldsymbol{x}; \boldsymbol{\theta}^k) \overset{d}{=} f_{\text{NN}}(\boldsymbol{M}\boldsymbol{x}; \tilde{\boldsymbol{\theta}}^k)$$

*Proof.* We claim that we can couple $\boldsymbol{\theta}^k$ and $\tilde{\boldsymbol{\theta}}^k$ such that $\boldsymbol{\theta}^k = \boldsymbol{M}\square\tilde{\boldsymbol{\theta}}^k$ almost surely. Therefore, for any $\boldsymbol{x}$:

$$f_{\text{NN}}(\boldsymbol{x}; \boldsymbol{\theta}^k) = f_{\text{NN}}(\boldsymbol{x}; \boldsymbol{M}\square\tilde{\boldsymbol{\theta}}^k) = g_{\text{NN}}(\tilde{\boldsymbol{W}}^k \boldsymbol{M}\boldsymbol{x}; \tilde{\boldsymbol{\psi}}^k) = f_{\text{NN}}(\boldsymbol{M}\boldsymbol{x}; \tilde{\boldsymbol{\theta}}^k), \tag{14}$$

which proves the lemma. It remains to show our coupling inductively on $k$. The base case $k = 0$ is assumed. For the inductive step, assume that the coupling is true for $k \geq 0$, and prove it for $k + 1$. By (a) using (14),

$$\boldsymbol{g}_{\tilde{\mathcal{D}}}(\tilde{\boldsymbol{\theta}}^k) = \mathbb{E}_{(\boldsymbol{x},y)\sim\mathcal{D}}[(f_{\text{NN}}(\boldsymbol{M}\boldsymbol{x}; \tilde{\boldsymbol{\theta}}^k) - y)\Pi_{B(0,R)}\nabla_{\boldsymbol{\theta}} f_{\text{NN}}(\boldsymbol{M}\boldsymbol{x}; \tilde{\boldsymbol{\theta}}^k)]$$

$$\overset{(a)}{=} \mathbb{E}_{(\boldsymbol{x},y)\sim\mathcal{D}}[(f_{\text{NN}}(\boldsymbol{x}; \boldsymbol{\theta}^k) - y)\Pi_{B(0,R)}\nabla_{\boldsymbol{\theta}} f_{\text{NN}}(\boldsymbol{M}\boldsymbol{x}; \tilde{\boldsymbol{\theta}}^k)]$$

Furthermore, by (a) the chain rule for differentiation combined with (14), (b) using that $\boldsymbol{M}^\top \in O(d)$ preserves the $\|\cdot\|_2$ norm,

$$\Pi_{B(0,R)}\nabla_{\boldsymbol{\theta}} f_{\text{NN}}(\boldsymbol{M}\boldsymbol{x}; \tilde{\boldsymbol{\theta}}^k) = \Pi_{B(0,R)}[\nabla_{\boldsymbol{W}} g_{\text{NN}}(\tilde{\boldsymbol{W}}^k \boldsymbol{M}\boldsymbol{x}, \tilde{\boldsymbol{\psi}}^k), \nabla_{\boldsymbol{\psi}} g_{\text{NN}}(\tilde{\boldsymbol{W}}^k \boldsymbol{M}\boldsymbol{x}, \tilde{\boldsymbol{\psi}}^k)]$$

$$\overset{(a)}{=} \Pi_{B(0,R)}[\nabla_{\boldsymbol{W}} g_{\text{NN}}(\boldsymbol{W}^k \boldsymbol{x}, \boldsymbol{\psi}^k)\boldsymbol{M}^\top, \nabla_{\boldsymbol{\psi}} g_{\text{NN}}(\boldsymbol{W}^k \boldsymbol{x}, \boldsymbol{\psi}^k)]$$

$$= \boldsymbol{M}^\top \square (\Pi_{B(0,R)}[\nabla_{\boldsymbol{W}} g_{\text{NN}}(\boldsymbol{W}^k \boldsymbol{x}, \boldsymbol{\psi}^k), \nabla_{\boldsymbol{\psi}} g_{\text{NN}}(\boldsymbol{W}^k \boldsymbol{x}, \boldsymbol{\psi}^k)])$$

By linearity of the $\square$ operation, it follows that

$$\boldsymbol{g}_{\tilde{\mathcal{D}}}(\tilde{\boldsymbol{\theta}}^k) = \boldsymbol{M}^\top \square \boldsymbol{g}_{\mathcal{D}}(\boldsymbol{\theta}^k).$$

So since $\boldsymbol{M} \in O(d)$ we have $\boldsymbol{M}^{-1} = \boldsymbol{M}^\top$, implying

$$\boldsymbol{g}_{\mathcal{D}}(\boldsymbol{\theta}^k) = \boldsymbol{M}\square\boldsymbol{g}_{\tilde{\mathcal{D}}}(\tilde{\boldsymbol{\theta}}^k).$$

Also couple the added noises $\xi^k$ and $\tilde{\xi}^k$ so that $\xi^k = \boldsymbol{M}\square\tilde{\xi}^k$, which can be done since $\boldsymbol{M} \in O(d)$ and Gaussians are orthogonal-invariant. By linearity of the $\square$ operation, and the inductive hypothesis, it holds that $\boldsymbol{\theta}^{k+1} = \boldsymbol{M}\square\tilde{\boldsymbol{\theta}}^{k+1}$ almost surely. The proof of the claim follows by induction. $\square$

The above lemma immediately implies the GD equivariances claimed in Proposition 2.5.

*Proof of GD equivariance in Proposition 2.5.* Notice that $G_{perm}$, $G_{\mathrm{sign,perm}}$, and $G_{\mathrm{rot}}$ can be identified with subgroups of $O(d)$. For i.i.d. initialization $\mu_{\boldsymbol{W}} = \mu_w^{\otimes(m \times d)}$, for any permutation matrix $\boldsymbol{M}$ we have $\boldsymbol{M}\square\boldsymbol{\theta}^0 \stackrel{d}{=} \boldsymbol{\theta}^0$. Similarly, if $\mu_w$ is symmetric then for any signed permutation matrix $\boldsymbol{M}$ we have $\boldsymbol{M}\square\boldsymbol{\theta}^0 \stackrel{d}{=} \boldsymbol{\theta}^0$. Finally, if $\mu_w$ is Gaussian then for any rotation matrix $\boldsymbol{M} \in SO(d)$ we have $\boldsymbol{M}\square\boldsymbol{\theta}^0 \stackrel{d}{=} \boldsymbol{\theta}^0$. Lemma E.3 implies the GD equivariances claimed in Proposition 2.5. $\square$