# OpenReview forum: "On the non-universality of deep learning: quantifying the cost of symmetry"
_NeurIPS.cc/2022/Conference — NeurIPS 2022 Accept_

### Official Review · Reviewer_1FaY · 2022-07-07

**Rating:** 6
**Confidence:** 2
**Soundness:** 3 good
**Presentation:** 3 good
**Contribution:** 2 fair

**Summary:**

This paper tries to analyze which functions can and cannot be learned by a standard MLP with gradient descent. This is analyzed by looking at the symmetry of the initialization and GD so the intuition is that functions that do not follow this symmetry are hard to learn.

**Questions:**

Can you give an example (that wasn't shown previously) of something that isn't weak learnable?

**Limitations:**

Not discussed

**Strengths And Weaknesses:**

I would first want to state that I am not a theoretician so I do not think I am the right person to review this paper, but I did the best I could to evaluate it.

Strength:
- The paper is clearly written
- The proofs seem correct to me
- The question of what functions can be learned is an important one so the work is important and novel.

Weaknesses:
- I mainly question the significance of some of the results, especially since the definition of weak learnability is very weak (as the "non-negligible" edge can be quite negligible with a large value of C). For example in Trm. 3.6 the condition $\sum_{\ell=0}^C||P_{V_{d,
\ell}}f_d||^2\geq d^{-C}$, the r.h.s increases (or not decreases) with $C$ and the r.h.s decreases rapidly. It makes the condition very weak (as weak learnability is very weak) so I don't think it adds any value to our understanding.
- Missing related work Shai Shalev-Shwartz, Ohad Shamir and Shaked Shammah "Failures of Gradient-Based Deep Learning".
- You do not show results for GD/SGD but for GD/SGD with noise. This is equivalent to what is done in SGLD or differentially private learning which significantly hampers performance. It is not clear to me at all that the results with noise are relevant to the standard noiseless training due to the major impact the noise makes on the optimization process.

---

> ### Author Response · Authors · 2022-08-02
> **Response to Reviewer 1FaY**
>
> Thank you for your kind and helpful comments.
>
> 1. **Significance of the results.** The characterizations of weak learnability in Theorems 3.4 and 3.6 were not previously known. We argue that they add value to our understanding:
> * If you believe that Definition 3.3 of weak learnability is very weak (i.e., that it is very easy to weakly-learn a function), then it should be a surprising and strong result that there are functions that are not weakly-learnable by training a standard neural network.
> * Furthermore, in a field as open as deep learning, we believe that it is of great value to precisely map out the space of what neural networks can and cannot learn. Our results on weak learning in Section 3.1 and strong learning in Section 3.2 provide concrete steps in this direction.
> 2. **Citing Shalev-Shwartz et al.**
> * This work was cited in our original submission, and you are right that it is highly relevant. We will add more comparison with this work by giving an example of a function that we can show is hard to weakly-learn, but was not previously known to be hard to weakly-learn. [see also our answer to your question below]
> * We found another paper [Shamir18] to which you may be referring. This has a direct relation to our SGD hardness result in the sense that it also provides cryptographic hardness. The result of Shamir has some advantages in that it applies to SGD learning of one-layer networks satisfying certain mild conditions. However our reduction has the advantage of not requiring the data whitening step in SGD that is necessary in the argument of [Shamir18]. Hence our reduction is the first to imply the non-universality result for SGD training without preprocessing. We therefore maintain our contribution in this part, but remove the claim that this was the first reduction of its kind for SGD.
> 3. **On the noise in GD/SGD.**
> *  You are correct that the GD result assumes noise on the gradients. This is a limitation of our result that we did not hide and that we would love to improve on. In the revision we will note this as an open problem in the conclusion.
> * On the other hand, the hardness result for SGD only assumes noise on the data labels – not on the gradients. Therefore it is not equivalent to SGLD (and noise on the labels is a relatively realistic assumption).
> 4. **Specific example of something that is not weak learnable that was previously unknown.** Yes, we will add a specific example in the revision, for fully-connected networks with symmetric initialization. This function is $f : $ {$+1,-1$}$~^d \to \mathbb{R}$, where $f(x) = g(\sum_{i=1}^d x_i)$, where $g(t) = \begin{cases} 0, t \equiv 0 \pmod{2} \\ 1, t \equiv 1 \pmod{4}, -1, t \equiv 3 \pmod{4} \end{cases}$. It is a function that depends on the sum of the inputs. Therefore, permutation equivariance is not enough to prove that it is not efficiently weakly-learnable by GD, and previous works fall short. Instead, we will illustrate that the sign-flip equivariance means that this function cannot be efficiently learned.
>
> [Shamir18] = “Distribution-specific hardness of learning neural networks”

---

### Official Review · Reviewer_m7n1 · 2022-07-10

**Rating:** 6
**Confidence:** 2
**Soundness:** 3 good
**Presentation:** 2 fair
**Contribution:** 3 good

**Summary:**

This paper mainly focused on the universal property of deep learning. Specifically, the authors showed that when the training algorithm satisfies a certain symmetry/equivairance, deep learning is no longer universal. Then, they characterize two types of functions with gradient descent (GD): 1) the functions that fully-connected networks could weakly learn on the binary hypercube as well as the unit sphere, and 2) the functions that neural networks with latent low-d structure. Lastly the authors extend the results to the stochastic GD.

**Questions:**

1. In Line 237, the authors mentioned that if the $G-$orbit-alignment is very small, then the GD training cannot efficiently improve on the trivial loss, could the authors elaborate on this part a bit more, since the right-hand side in the equation between 235 and 236 also depends on other factors?

2. As I mentioned in the weakness part, some definitions and assumptions are abstract, could the authors give some examples for readers to follow easily?

3. Could the authors provide some numerical studies to demonstrate the applications mentioned in the paper indeed work in practice.

**Ethics Review Area:**

["I don’t know"]

**Limitations:**

N/A.

**Strengths And Weaknesses:**

Strength:

1. The paper has good novelty and defines a lot of new concepts, e.g., G-orbit-alignment, etc. It pushes the boundary of the research.

2. Use some decent mathematic tools.

Weakness:

1. The paper is very abstract, and some definition is not easy to think about. Could the authors give some examples to help readers better understand the newly defined concepts?

2. The authors always assume that the GD training algorithm is G-equivariant, but in practice, I was wondering if this condition could hold.

3. The paper lacks some empirical study to support the authors' findings.

---

> ### Author Response · Authors · 2022-08-02
> **Response to Reviewer m7n1**
>
> Thank you for your kind and helpful comments.
>
> 1. **Adding examples of GD training being equivariant.** We will add a paragraph entitled “Examples of G-equivariant algorithms in deep learning” that gives examples of when G-equivariance holds; including token-permutation-equivariance for transformers without positional embeddings, translation equivariance for circular convolutional networks where the convolution “wraps around”. In Assumption 2.4 we give conditions on networks that guarantee deep learning is permutation, sign-permutation, and orthogonal equivariant for different initializations (Proposition 2.5). These conditions are satisfied by fully-connected networks.
>
> 2. **Question about Line 237.** For efficient noisy gradient descent training the other factors (step size, Lipschitzness/clipping radius, inverse noise) are polynomially-bounded. This means that if the G-orbit-alignment is smaller than any polynomial, then the right-hand-side vanishes, implying that the algorithm cannot make progress. We will clarify this.
>
> 3. **Adding more concrete examples.**
> * We will add a paragraph called “Examples of G-equivariant algorithms in deep learning” (see our response above).
>
> * We will move some of the concrete calculations of the G-orbit-alignment for weak-learnability from the appendix to the main text, to better illustrate how to use our theorem.
> * We will also add a concrete example of a function which was not previously known to be hard to weakly-learn. It is $f(x) = g(\sum_{i=1}^d x_i)$, where $g(t) = \begin{cases} 0, t \equiv 0 \pmod{2} \\ 1, t \equiv 1 \pmod{4}, -1, t \equiv 3 \pmod{4} \end{cases}$. We will explain how to use our tools to derive this.
>
> 4. **Lack of empirical study.** This paper focuses on establishing rigorous results under the data assumptions made, and does not provide numerical experiments. Note that prior works, on which the current submission extends upon, namely [AS20], [ACHM22], [ABM22] (see discussions on these papers in the submission), did include simulations to show the limitations of learning under specific invariances. Here, instead, we obtain a more general framework for this type of arguments that leads to tighter lower-bounds.

---

### Official Review · Reviewer_EG9T · 2022-07-11

**Rating:** 7
**Confidence:** 3
**Soundness:** 4 excellent
**Presentation:** 3 good
**Contribution:** 4 excellent

**Summary:**

This paper studies lower bounds of Gradient Descent (with noise) (GD) and Stochastic Gradient Descent (SGD) training algorithms that are also $G$-equivariant. The paper introduces the concept of $G$-orbit alignment which is used towards proving the first main result which is a computational lower bound on $G$-equivariant GD which appears novel. Two applications of this lower bound are provided in terms of weak learnability of functions on the hypercube, hypersphere, and a necessity result on the merged staircase property. As a separate, but interesting in it's own right contribution, the authors provide a first-of-its-kind hardness result for SGD using fully-connected networks on the Learning Parities with Gaussian Noise problem.

**Questions:**

One of the central assumptions in much of the technical results is that $\mu_x$ is $G$-invariant. Often times real-world datasets, due to noise or other randomness in the data collection process exhibit soft equivariance rather than hard ones. Can you provide commentary on how this specifically affects the weak learnability results ---i.e. does depth now matter beyond 2-layer networks?

**Strengths And Weaknesses:**

**Disclaimer**
I did not check the veracity of the proofs beyond Appendix A.

This paper enjoys several strengths. In my opinion, the paper studies an extremely interesting problem and makes progress by providing several novel insights. I especially appreciated the idea of $G$-orbit alignment and its use in characterizing weak learnability. This is both intuitively clear and a great application. I also equally found the main results of the lower bounds on $G$-equivariant GD insightful and a significant step over previous work. The computational hardness results for SGD seemed interesting but this aspect was a bit beyond my area of expertise. The authors could do a bit more towards making this section more approachable given that the paper does not completely exhaust the 9-page limit. My main criticism of this work is that a lot of the interesting detail in the Appendix could have been brought forward to the main text. For example, the impossibility results for both applications that actually utilize Theorem 3.1 could be better served in the main paper. Overall, I found this paper to be a thoroughly enjoyable read as overall the clarity and high quality of writing made difficult ideas mostly approachable but at the same time did not water down their perceived impact.

**Minor**
- line 147 compact groups, not any groups

- minor notational issues. In definition 2.4 $\psi, \tilde{\theta}$ are not defined

- Definition 3.2 seems a bit contrived. Can you provide a bit more intuition for the constant 9/10?

---

> ### Author Response · Authors · 2022-08-02
> **Response to Reviewer EG9T**
>
> Thank you for your kind and helpful comments. We are glad that you enjoyed the paper.
>
> 1. **Adding details from the appendix.** We think that the representation theory arguments used to bound the G-orbit-alignment for the weak learnability results could be interesting to the reader, but these currently did not fit in the main body of the paper (we would need an extra 0.5-0.75 pages). If accepted, we could add these into the main text since we would get an extra page in the final version.
> Approachability of SGD section. We will update the SGD section to have more detail.
> 2. **Minor comments.** We will update these in the revision, thanks.
>
>     Line 147: will correct
>
>     Definition 2.4: will correct
>
>     9/10 is an arbitrary constant that is large enough. We will explain this in the revision.
>
> 3. **Question about real-world data.** This is a great question. It is likely that we could extend the result to when \mu_X is approximately G-invariant – but we are unsure whether such extensions would be applicable to real-world datasets. Beyond such soft relaxations, it seems difficult to extend our current arguments.  Because of this, we cannot comment on whether depth now matters for weak-learnability for data distributions that are not Gaussian or uniform on binary hypercube. We will update the wording of our abstract and paper to avoid giving any impression that the weak-learnability characterization holds beyond these settings.

---

> > ### Comment · Reviewer_EG9T · 2022-08-07
> > **Re:Rebuttal**
> >
> > Thank you for answering my main questions. I'm satisfied and happy to maintain my current score of 7.

---

### Official Review · Reviewer_bjbs · 2022-07-11

**Rating:** 5
**Confidence:** 3
**Soundness:** 2 fair
**Presentation:** 2 fair
**Contribution:** 3 good

**Summary:**

This paper studies the learnability of Lipschitz models with GD/SGD optimization. It assumes that the learning algorithm satisfies G-equivariant for some non-trivial symmetry group G (which holds for GD/SGD learning in neural networks), and shows two types of results:
- With GD learning, the paper characterizes the learnability of Lipschitz models on 1) functions of both the hypercube and the unit sphere; 2) functions with latent low-dimensional structure.
- For the function given in Eq. (4), with SGD learning, a fully-connected network cannot learn and evaluate the function (4) in polynomial time with respect to the input dimension d.


**Questions:**

Please refer to the [weakness] part.

**Limitations:**

Please refer to the [weakness] part.

**Strengths And Weaknesses:**

Strength:

The first result characterizes the learnability of a rather general class of learning models. The second result provides a nice example that neural networks, despite their universal approximation ability, cannot efficiently learn.

Weakness:

My major concern is that it requires the architecture to have bounded Lipschitz constant over parameters. However, almost all neural networks have unbounded Lipschitz constants. The authors argue (in the supplementary materials) that, in practice, we can clip the gradients to yield bounded derivatives. However, for a purely theoretical paper, this is not a sound justification. Clipping the gradient is equivalent to changing the learning algorithm, but the objective function remains unchanged, still having an unbounded Lipschitz constant.

In that sense, Section 3 only holds for models with Liptchiz constant, for example, one-hidden-layer networks with fixed output layers and bounded-gradient activations. This deviates from the major claims --- "regular architectures and initialization" of neural networks. Meanwhile, the authors criticize that the prior work [AS20] does not "reflect architectures used in practice", which is unfair.

To rigorously prove for neural network training, the authors can either show i) GD operates in a bounded region, within which we have bounded Lipschitz, or ii) the theorem holds for gradient-clipping GD. Please consider addressing this problem. Otherwise, it is not eligible to claim the results hold for general DNN. Note that prior works (such as [Ele22]) that require C-Lipschitz on the learning model did not claim to hold for NN in their theorems.

Another concern is that the second result does not seem to counter [AS20] directly. [AS20] shows there exists DNN that can emulate any efficient learning algorithm (that is able to learn in poly-time). It assumes that at least one poly-time efficient algorithm exists. However, the non-universality shown in Theorem 4.4 relies on the hardness of γ-LPGN hardness, which is not poly-time solvable by any algorithm. There is a gap between the "universality" discussed in [AS20] and this paper. It is still possible that a practical DNN with G-equivariance SGD learning can emulate any efficient algorithm.

---

> ### Author Response · Authors · 2022-08-02
> **Response to Reviewer bjbs**
>
> Thank you for your careful reading of our paper and for your helpful comments. Thank you also for your kind comments on the strengths of this paper. You have expressed two concerns.
>
> 1. **Lipschitzness assumption.** Your first concern is that Theorem 3.1 assumes the architecture is Lipschitz, which may be unrealistic. In a footnote, we claimed that one could avoid this assumption with gradient clipping. You want us to formalize this and it is understandable.
>
>      Accordingly, in our revision we will update the GD training algorithm to clip the gradients of the network. This means that in the revision we make no Lipschitzness assumption anymore. The proofs remain quite similar, since the Lipschitzness assumption was only used to bound the magnitude of each step. This matches the request (option ii) from your review.
>
>     Note: Clipping the gradients is done in [AS20], as well several related works [AKM+21], [ACHM22] that study noisy GD on neural networks. When we stated that the architectures of [AS20] did not "reflect architectures used in practice", we were instead referring to the emulation architectures used in [AS20] to simulate arbitrary computational circuits, that is quite far from the type of architectures/initialization used in practice that have iid weight initializations on more regular types of layers.
>
> 2. **SGD hardness result.** Your second concern is that the SGD hardness argument (our second result) may not imply the claimed non-universality. However, we believe that here you have misunderstood the result.
>
>     Theorem 4.4 shows that the function class $\mathcal{F} = $ {$f_{\ast}$} consisting of only the function $f_*(x) = \sum_{i}^d x_i \pmod{8}$ cannot be efficiently learned by an SGD-trained fully-connected neural network with symmetric initialization (such as Gaussian initialization). However, this function class $\mathcal{F}$ *is trivial to learn* for an arbitrary efficient algorithm. There is only one element in the function class $\mathcal{F}$, so the algorithm that ignores the samples and just outputs $f_*(\mathbf{x})$ has learned it. Hence, Theorem 4.4 does show the claimed non-universality: it exhibits a class $\mathcal{F}$ where SGD-trained FC-nets fail, but where there is an efficient algorithm that succeeds.
>
>     In order to prove this result, we combine $\gamma$-LPGN hardness and the $G$-equivariance of SGD, where $G$ is the group of sign flips. We show that SGD-trained FC-nets must work much harder than other algorithms to learn $\mathcal{F}$: if they could learn $\mathcal{F}$ efficiently, then there would be an efficient algorithm for solving $\gamma$-LPGN. On the other hand, this reduction to $\gamma$-LPGN does not hold for arbitrary efficient algorithms since they do not have the sign-flip-equivariance that SGD training of FC-nets has.
>
>     We will update the SGD section to be clearer and have more detail in the revision.
>
> Hopefully we have addressed your two concerns. Please let us know if there is anything else we can clarify.

---

> > ### Comment · Reviewer_bjbs · 2022-08-09
> > **I did misunderstand the SGD hardness result**
> >
> > Thanks so much for the authors' time and effort in the reply. My original review was incorrect about the hardness argument of SGD. Indeed, the second result of this paper provides a good insight into the limitation of SGD training. Even for a seemingly strong algorithm, as long as it has a certain symmetry, it may not be able to learn an easy ground truth within a reasonable time. I would like to raise the score.
> >
> > For the bounded Lipchitz issue, I need some time to check whether the proof only utilizes the bounded update at each iteration. I will update this comment later.

---

> > > ### Author Response · Authors · 2022-08-09
> > > **Re: SGD hardness**
> > >
> > > Thank you for your careful reading of the document and for raising your score.

---

### Author Response · Authors · 2022-08-02
**General response**

We thank the reviewers for their careful reading of the paper and for their thoughtful and helpful comments. We have been working on a revision, which we will upload soon. We summarize the main changes in our revision:
1. Improving the exposition of our results:
* Adding more examples of standard architectures for which training is G-equivariant.
* Adding more explanation and intuition for the SGD hardness result in the main text.
2. Moderately strengthening our results:
* Removing the Lipschitzness assumption and replacing it with gradient clipping as suggested by Reviewer bjbs. The result is strictly stronger now, and the proofs are mostly unchanged.
3. Improving the comparison to past work:
* Adding a concrete example of a function that is now known to be weakly-learnable, but was not previously known.
* Adding a reference to [Shamir18], see remark in response to reviewer 1FaY.

We summarize again the main contributions of the paper:

We prove a general computational lower bound for learning with neural networks trained by noisy gradient descent (GD). Our result applies whenever GD training is equivariant (true for many standard architectures), and quantifies the alignment needed between architectures and data in order for GD to learn. As applications: (i) we characterize the functions that fully-connected networks can  weakly-learn on the binary hypercube and unit sphere, demonstrating that depth-2 is as powerful as any other depth for weak-learning by fully-connected networks; (ii) we extend the merged-staircase necessity result for learning with latent low-dimensional structure [ABM22] beyond the mean-field regime.

Further, the G-orbit-alignment yields a tighter lower-bound that the cross-predictability bound from [AS20], [ACHM22] (see the example of k-parities). Surprisingly, our investigation also shows that full parities can be learned from regular NN with iid initializations, as opposed to the general beliefs emerging from [AS20], [ACHM22], [NY21] papers.

Finally, our techniques extend to stochastic gradient descent (SGD), for which we show nontrivial hardness results for learning with fully-connected networks, based on cryptographic assumptions, without any whitening processing steps.

[Shamir18] = “Distribution-specific hardness of learning neural networks”

[AS20] = “Poly-time universality and limitations of deep learning“

[ABM22] = “The merged-staircase property: a necessary and nearly sufficient condition for sgd learning of sparse functions on two-layer neural networks“

[ACHM22] = “An initial alignment between neural network and target is needed for gradient descent to learn”

[NY21] = “On symmetry and initialization for neural networks”

---

### Meta-Review · Area_Chair_rgYq · 2022-08-26

**Recommendation:** Accept
**Confidence:** Certain

**Metareview:**

This paper continues a line of work on the universality of deep learning and provide some natural assumptions under which previous results showing universality are not valid any more. This is an important contribution as indeed previous universality result seem to rely on unnatural constructions and architectures and this work highlight this.

There were some concerns about the practicality of the results, however this paper is mostly theoretical and I would like to judge it in the context of previous work such as Abe & Sandon. As far as I can see, in this context, the authors provide new insights and help to advance our understanding.

**Award:**

No

---

### Decision · Program_Chairs · 2022-09-14

Accept